1 **Contribution and uncertainty of sectorial and regional emissions to regional and global**
2 **PM$_{2.5}$ health impacts**

Monica Crippa[1], Greet Janssens-Maenhout[1], Diego Guizzardi[2], Rita Van Dingenen[1], Frank
Dentener[1]

[1]European Commission, Joint Research Centre (JRC), Via E. Fermi 2749, I-21027 Ispra (VA),
Italy

[2]Didesk Informatica, Verbania (VB), Italy

**Abstract**

In this work we couple the HTAP_v2.2 global air pollutant emission inventory with the global
source receptor model TM5-FASST to evaluate the relative contributions of the major
anthropogenic emission sources (power generation, industry, ground transport, residential,
agriculture and international shipping) to air quality and human health in 2010. We focus on
particulate matter (PM) concentrations because of the relative importance of PM$_{2.5}$ emissions in
populated areas and the well-documented cumulative negative effects on human health. We
estimate that in 2010, depending on the region, annual averaged anthropogenic PM$_{2.5}$
concentrations varied between ca 1 and 40 µg/m$^3$, with the highest concentrations observed in
China and India, and lower concentrations in Europe and North America. The relative
contribution of anthropogenic emission sources to PM$_{2.5}$ concentrations varies between the
regions. European PM pollution is mainly influenced by the agricultural and residential sectors,
while the major contributing sectors to PM pollution in Asia and the emerging economies are the
power generation, industrial and residential sectors. We also evaluate the emission sectors and
emission regions in which pollution reduction measures would lead to the largest improvement
on the overall air quality. We show that air quality improvements would require regional
policies, in addition to local and urban scale measures, due to the transboundary features of PM
pollution. We investigate emission inventory uncertainties and their propagation to PM$_{2.5}$
concentrations, in order to identify the most effective strategies to be implemented at sector and
regional level to improve emission inventories knowledge and air quality modeling. We show
that the uncertainty of PM concentrations depends not only on the uncertainty of local emission
inventories, but also on that of the surrounding regions. Countries having high emission
uncertainties are often impacted by the uncertainty of pollution coming from surrounding
regions, highlighting the need of effective efforts in improving emission not only within a region
but also from extra-regional sources. Finally, we propagate emission inventories uncertainty to
PM concentrations and health impacts. We estimate 2.1 million premature deaths/year with an
uncertainty of more than 1 million premature deaths/year due to the uncertainty associated only
with the emissions.

**1 Introduction**

Ambient particulate matter pollution ranks among the top five risk factors globally for loss of healthy life years and is the largest environmental risk factor (Lim et al., 2013;Anderson et al., 2012;Anenberg et al., 2012;Cohen et al., 2017). The world health organization (WHO, 2016) reported about 3 million premature deaths worldwide attributable to ambient air pollution in 2012. Health impacts of air pollution can be attributed to different anthropogenic emission sectors (power generation, industry, residential, transport, agriculture, etc.) and sector-specific policies could effectively reduce health impacts of air pollution. These policies are usually implemented under national legislation (Henneman et al., 2017; Morgan, 2012), while in Europe transboundary air pollution is also addressed by the regional protocol under the UNECE Convention on Long-Range Transport of Air Pollution (CLRTAP). At city/local level, several studies have been developed to assess the contribution of sector specific emissions to PM2.5 (particulate matter with a diameter less than 2.5 µm) concentrations with the aim of designing air quality plans at local and regional level (Karagulian et al., 2015; Thunis et al., 2016). Indeed, particulate matter can travel thousands of kilometers, crossing national borders, oceans and even continents (HTAP, part A, 2010). Local, regional and international coordination is therefore needed to define air pollution policies to improve global air quality and possibly human health. The CLRTAP's Task Force on Hemispheric Transport of Air Pollution looks at the long-range transport of air pollutants in the Northern Hemisphere aiming to identify promising mitigation measures to reduce background pollution levels and its contribution to pollution in rural as well as urban regions. Although primary $PM_{2.5}$ and intermediately lived (days-to-weeks) precursor gases can travel over long distances, the transboundary components of anthropogenic PM are mainly associated with secondary aerosols which are formed in the atmosphere through complex chemical reactions and gas-to-aerosol transformation, transport and removal processes, of gaseous precursors transported out of source regions (Maas and Grennfelt, 2016). However, the most extreme episodes of exposure often occur under extended periods of low wind speeds and atmospheric stability, favoring formation of secondary aerosols close to the source regions. Secondary aerosol from anthropogenic sources consists of both inorganic -mainly ammonium nitrate and ammonium sulfate and ammonium bisulfate and associated water, formed from emissions of sulphur dioxide ($SO_2$), nitrogen oxides (NOx) and ammonia ($NH_3$), and organic compounds involving thousands of compounds and often poorly known reactions (Hallquist et al., 2009). Exposure to and impact from aerosols on humans can be estimated by a variety of approaches, ranging from epidemiological studies to pure modelling approaches. The Burnett et al. (2014) risk-response methodology is often used in models to estimate premature deaths/mortality (PD) due to air pollution exposure, e.g. in Lelieveld et al. (2015) and Silva et al. (2016), who report a global mortality in 2010 due to air quality issues induced by anthropogenic emissions of 2.5 and 2.2 million people, respectively. A higher global mortality is found in a more recent work by Cohen et al. (2017) accounting for 3.9 million premature deaths/year due to different model assumptions. In Europe, Brant et al. (2013) estimate 680 thousand premature deaths, which is twice as high as the numbers reported for the CAFE (Clean Air for Europe) study (Watkiss et al., 2005). Recently, using the same emission database as in this study, Im et al. (2017) report a multi-model mean estimate of PD of 414 thousand (range 230-570 thousand) for Europe and 160 thousand PDs for the USA. At the global scale, models, in some cases using satellite information (Brauer et al., 2015;Van Donkelaar et al., 2016), are the most practical source of information of exposure to air pollution. However, model calculations are subject to a range of uncertainties related with incomplete understanding of transport, chemical transformation, removal processes, and not the least, emission information.

This work is developed in the context of the TF HTAP Phase 2 (Galmarini et al., 2017a), where a number of models are deployed to assess long-range sensitivities to extra-regional emissions, using the same HTAP_v2.2 anthropogenic emission inventory (Janssens-Maenhout et al., 2015). Differences in model results illustrate uncertainties in model formulations of transport, chemistry and removal processes, and are addressed in separate studies (Liang et al., 2018), but not of uncertainties in emission inventories. The objectives and novelties of this study are the evaluation of i) the relative contribution of anthropogenic emission sources to PM2.5 concentrations at global scale, ii) the emission sectors and emission regions in which pollution reduction measures would lead to the largest improvement on the overall air quality and iii) the relevance of uncertainties in regional sectorial emission inventories (power generation, industry, ground transport, residential, agriculture and international shipping), and their propagation in modelled $PM_{2.5}$ concentrations and associated impacts on health. This work applies the global source-receptor model TM5-FASST (TM5-FAst Scenario Screening Tool), which is extensively described and evaluated in this special issues (Van Dingenen et al., 2018), and couples it to the HTAP_v2.2 global emission inventory for the year 2010 to estimate global air quality and associated health impacts in terms of $PM_{2.5}$ concentrations. The regional and global scale, the focus on annual $PM_{2.5}$ and associated health metrics, warrants the use of the TM5-FASST model. However, the most extreme episodes of pollution may occur at more local-to-regional scales justifying the need for local. For instance, a recent study performed over hundreds of cities in Europe (Thunis et al., 2017) shows that in order to comply with the standards prescribed by the Air Quality Directives and the health guidelines by WHO, local actions at the city scale are needed.

Specifically, we show that the impact of emission inventory uncertainty on mortality estimates is comparable with the range of uncertainty induced by air quality models and population exposure functions. We also investigate the uncertainties in $PM_{2.5}$ from within the region to extra-regional contributions. Based on our analysis on the importance of emission uncertainties at sector and regional level on $PM_{2.5}$, we aim at informing local, regional and hemispheric air quality policy makers on the potential impacts of sectors with larger uncertainties (e.g. residential and agriculture) or regions (e.g. developing and emerging countries).

## 2 Methodology

### 2.1 TM5-FASST model and emission perturbations

This work is an application of the TM5-FASST model, which is extensively documented in a companion publication in this special issue. Van Dingenen et al., (2018) provide an extensive evaluation of the model, model assumptions, performance with regard to linearity and additivity of concentration response to different size of emission perturbations and future emission scenarios. Below we summarize the most important features of relevance for this work, and refer for more detail to Van Dingenen et al., (2018).

In order to calculate $PM_{2.5}$ concentrations corresponding to the HTAP_v2.2 emissions, we use the native 1°x1° resolution source-receptor gridmaps obtained for TM5-FASST_v0 (Van Dingenen et al., 2018). The TM5-FASST source-receptor model is based on a set of emission perturbation experiments (-20 %) of $SO_2$, NOx, CO, $NH_3$, and VOC and $CH_4$ using the global 1°x1° resolution TM5 model, the meteorological year 2001 (which was also used for the HTAP

Phase 1 experiments) and the community emission dataset prepared for the IPCC AR5 report (RCP, Representative Concentration Pathway) emissions for the year 2000 (Lamarque et al., 2010). TM5-FASST uses aggregated regional emissions (i.e. one annual emission value per pollutant or precursor for each of the 56 regions + shipping), with an implicit underlying 1°x1° resolution emission spatial distribution from RCP year 2000 which was partly based EDGAR methodology and gridmaps. The concentration of $PM_{2.5}$ contributing from and to each of 56 receptor regions is estimated as a linear function of the emissions of the source regions, including the aerosol components BC, primary organic matter (POM), $SO_4$, $NO_3$, and $NH_4$. While Secondary Organic Aerosol (SOA) from natural sources is included in the model calculations using the parameterisation described in Dentener et al. (2006), no explicit treatment of anthropogenic SOA is considered, since no reliable emission inventories of SOA precursor gases was available, and formation processes were not included in the parent TM5 model. A recent study by Farina et al. (2010) indicates a global source of 1.6 Tg, or ca 5.5 % of the overall SOA formation due to anthropogenic SOA. The relative importance of anthropogenic SOA ranges regionally widely, and is deemed higher in regions with less VOC emission controls. Inclusion of SOA would possibly lead to a somewhat larger role of the transboundary pollution transport (Farina et al., 2010;Peng et al., 2016;Shiraiwa et al., 2017), mainly for regions and sectors with large PM and VOC emissions (e.g. residential, and to some extent transport and industry).

Under the assumption that the individual sector contributions add up linearly to total $PM_{2.5}$ – this assumption is evaluated in Van Dingenen et al. (2018) to hold in most regions within 15 % error- the comparison of $PM_{2.5}$ concentrations calculated for the reference and scenario case yields an estimation of the contribution of each sector to total $PM_{2.5}$ concentrations.

Specifically, the reduced-form model TM5-FASST is computing the concentration resulting from an arbitrary precursor emission strength $E_i$ using a first order perturbation approach, i.e. for each PM component $j$, the change in concentration $dPM_j$ resulting from a change in emission strength $E_i(x)$ of precursor $i$ in source region $x$, relative to a reference emission $E_{i,ref}(x)$, is approximated by the first linear term of a Taylor expansion of PM as a function of emissions:

$$dPM_j(y) \cong A_{ij}[x,y]\big[E_i(x) - E_{i,ref}(x)\big] \tag{Eq. 1}$$

where

$$A_{ij}[x,y] = \frac{\Delta C_j(y)}{\Delta E_i(x)} \text{ with } \Delta E_i(x) = 0.2 E_{i,ref}(x) \tag{Eq. 2}$$

$A_{ij}[x,y]$ is a set of independently computed source-receptor matrices, expressing the linearized emission-concentration response between each relevant precursor ($i$) emission and PM component $j$ concentration, for each pair of source ($x$) and receptor ($y$) regions (Van Dingenen et al., 2018).

In Sect. S1.2 we explain in detail how Eq. 1 can be also applied for evaluating the attribution by sector as well as by source region, based on the work by Van Dingenen et al. (2018). Thus to calculate total $PM_{2.5}$ concentration in each receptor region, the 56 source region individual contributions must be summed. Using this approach, it is possible to evaluate the $PM_{2.5}$

concentrations from "within-region" and "extra-regional" $PM_{2.5}$ emissions. The extra-regional contribution represents the RERER metric (Response to Extra-Regional Emission Reduction) for a specific region used across the whole HTAP experiment (Galmarini et al., 2017b), in particular focusing on the $PM_{2.5}$ concentration reduction due to the contribution of the emissions of each anthropogenic sector (Eq. 3):

$$RERER = \frac{\sum R(foreign\ regions)}{\sum R(all\ regions)}$$
(Eq. 3)

where R represents the concentration response to each sector emission decrease.

As depicted in Fig. S1, the 56 TM5-FASST regions cover the entire globe, but their areal extent differs in terms of size, population, emission magnitude and presence of neighbouring countries (e.g. Europe comprises 18 TM5-FASST regions). In order to make the evaluation of external impacts on smaller regions (e.g. European countries) comparable to those of larger regions (like USA, China and India), in this work an aggregation procedure to 10 world regions (refer to Table S2) has been applied (China+, India+, SE Asia, North America, Europe, Oceania, Latin America, Africa, Russia and Middle East). In this work we focus on particulate matter due to its negative effects on human health (WHO, 2013;Pope and Dockery, 2006),Worldbank, 2016). The TM5-FASST model includes an assessment of the premature mortality due to ambient $PM_{2.5}$ concentrations on exposed population following the methodology developed by Burnett et al. (2014), as discussed in Sect. 4. Health impacts due to indoor air pollution or ozone are not evaluated in this work.

In the following, we will address the uncertainty of sector specific emissions from this inventory in a quantitative way as well as the differences we observe from one region to the other, based on the uncertainty of activity data and emission factors. As discussed in the next section, the reason to use HTAP_v2.2, and not e.g. the RCP2000 as the basis for our assessment of emission propagation, is that the TF HTAP aims at bringing policy relevant information, and to this end, it has compiled a policy relevant emission inventory (HTAP_v2.2) for the most recently available year. While the RCP2000 was at the basis of the FASST calculations, and presented the best community emissions effort at the time, the HTAP_v2.2 inventory is now much more accurate in particular given the focus on regional (and not so much gridded) emission analysis of our work.

**2.2 HTAP_v2.2 emissions**

The global anthropogenic emission inventory HTAP_v2.2 for the year 2010 (Janssens-Maenhout et al., 2015) is input to the global source-receptor model TM5-FASST to evaluate $PM_{2.5}$ concentrations for each world region/country with the corresponding health effects. The HTAP_v2.2 inventory includes for most countries official and semi-official annual anthropogenic emissions of $SO_2$, NOx, CO (carbon monoxide), NMVOC (non-methane volatile organic compounds), $PM_{10}$ (particulate matter with a diameter less than 10 μm) $PM_{2.5}$, BC (black carbon) and OC (organic carbon) by country and sector (Janssens-Maenhout et al., 2015). Here we focus on the 6 major anthropogenic emission sectors contributing to global $PM_{2.5}$

concentrations, namely the power generation ("power"), non-power industry, industrial processes and product use ("industry"), ground transportation ("transport"), residential combustion and waste disposal ("residential"), agriculture ("agriculture") and international shipping ("ship"). International and domestic aviation emissions are not considered in this study due to the lower contribution to air pollution compared to other anthropogenic sectors. It should be noted that agricultural emissions do not include agricultural waste burning and forest and savannah fires. Details on the emissions included in each aggregated sector can be found in Janssens-Maenhout et al. (2015). In addition to the reference HTAP_v2.2 emissions for the year 2010, a set of emission perturbation scenarios has been created by subtracting from the reference dataset the emissions of each sector.

## 2.3 Emission inventory uncertainties

In order to investigate how computed $PM_{2.5}$ concentrations are affected by the uncertainty of emission inventories, we perform a sensitivity analysis testing the upper and lower range of HTAP_v2.2 emissions including their uncertainties. Aggregated emissions of a certain pollutant $p$, from a sector $i$ and country $c$ are calculated as the product of activity data (AD) and emission factors (EF), therefore the corresponding uncertainty ($\sigma_{i,c,p}$) is calculated as following:

$$\sigma_{EMI\,i,c,p} = \sqrt{\sigma_{ADi,c}^2 + \sigma_{EF,i,p,c}^2} \qquad \text{(Eq.4)}$$

where $\sigma_{AD}$ and $\sigma_{EF}$ are the uncertainties (%) of the activity data and emission factors for a certain sector (i), country (c) and pollutant (p). Uncertainty values of the activity data by sector and country are obtained from Table 2 of Janssens-Maenhout et al. (2017) and Olivier et al. (2016). Using this approach, the uncertainty in the global total anthropogenic $CO_2$ emissions is estimated to range from -9% to +9% (95% confidence interval), with larger uncertainties of about ±15% for non-Annex I countries, and uncertainties of less than ±5% are obtained for the 1990 OECD countries[1] for the time series from 1990 (Olivier et al, 2016) reported to UNFCCC. Uncertainty values for the emission factors of gaseous pollutants are retrieved from the EMEP/EEA Guidebook (2013) and Bond et al. (2004) for particulate matter. In this work we assume that reported countries emissions are based on independent estimations of activity data and emission factors EFs, hence no cross-country correlation structure is assumed. This is in contrast to bottom-up gridded emission inventories like EDGAR, where the use of global activity datasets may lead to correlated errors between countries.

Therefore, we can calculate the overall uncertainty $\sigma_{EMI\,p,c}$ with the following equation (EMEP/EEA, 2013).

$$\sigma_{EMI\,p,c} = \sqrt{\sum_i \left( \sigma_{EMI\,i,c,p} * \frac{EMI_{i,c,p}}{EMI_{tot,c,p}} \right)^2} \qquad \text{(Eq. 5)}$$

---

[1] OECD countries in 1990: Australia, Austria, Belgium, Canada, Denmark, Finland, France, Germany, Greece, Iceland, Ireland, Italy, Japan, Luxembourg, Netherlands, New Zealand, Norway, Portugal, Spain, Sweden, Switzerland, Turkey, United Kingdom, United States.

where $EMI_{i,c,p}$ (in kton) represents the emission of a certain pollutant (p) in a certain country (c)
from a specific sector (i) and $EMI_{tot,c,p}$ (in kton) the corresponding emissions from all sectors for
that country and pollutant.

Table S3, reports the overall uncertainty calculated for each pollutant and for each TM5-FASST
region. Using an additional constraint that EFs and activities cannot be negative, a lognormal
distribution of the calculated uncertainties is assumed (Bond et al., 2004). Therefore we can
calculate the upper and lower range of emission estimates multiplying and dividing the reference
emissions by $(1+\sigma_{p,c})$, respectively. We do not account for the uncertainties of the atmospheric
transport model and the uncertainties due to aggregation, which are larger over smaller TM5-
FASST regions. Based on the upper and lower emission range per region, new TM5-FASST
model runs have been performed per source region to retrieve the corresponding range of
concentrations in receptor regions (therefore the total number of computations is 56*2 for the
uncertainty analysis).

**3 TM5-FASST modelling results**

In this section, we first provide 'central' estimates of regional (Sect. 3.1), sectorial (Sect. 3.2) and
gridded (Sect. 3.3) contributions, whereas the corresponding uncertainty estimates are discussed from
Sect. 3.4 onward.

**3.1 Regional contributions to PM$_{2.5}$ concentrations**

Figure 1 provides a global perspective on the fraction of within-region and extra-regional PM$_{2.5}$
concentrations for 10 aggregated world receptor regions using emissions of the year 2010, with
the extra-regional fraction (using the RERER metric) broken down into source region
contributions. Annual average population weighted anthropogenic PM$_{2.5}$ concentrations (refer to
Van Dingenen et al., (2018) for the calculation of this metric) ranged from few µg/m$^3$ (e.g. in
Oceania or Latin America), around 7-8 µg/m$^3$ for North America and Europe, and up to 33-39
29 µg/m$^3$ in China+ (including also Mongolia) and India+ (including also the rest of South Asia).
Anthropogenic PM$_{2.5}$ pollution in China+ and India+ is mainly affected by large emission
sources within the country (98 and 96%, respectively; RERER 2-4 %), although 4 % of the
Indian anthropogenic PM$_{2.5}$ pollution is mainly transported from the Gulf region and Middle East
, as was also observed by (Venkataraman et al., 2018). North America (98%) and Oceania (98%)
are mainly influenced by within-regional pollution due to their geographical isolation from other
regions. TM5-FASST computations attributed 11 % of the PM$_{2.5}$ in Europe to extra-regional
sources; for the Middle East and Gulf region extra-regional contributions amount to 18% (mainly
from Europe and Russia), for Africa 25% (mainly from Europe and Middle East), and Russia
28% (mainly from Europe, Middle East and Gulf region and China). Shipping emissions are not
considered in this figure due to their international origin, while inland waterways emissions are

still included in the ground transport sector. Transboundary air pollution is known to be an important issue in the rest of Asia, in particular for pollution transported from China to Korea and Japan (Park et al., 2014) and we estimate that the contribution of transported PM is up to 40% in South Eastern Asia (mainly from China and India). Within-region and extra-regional $PM_{2.5}$ concentrations for all the TM5-FASST regions are reported in Table S2.

Focusing on Europe, Fig. 2 shows within-region (in black) vs. extra-regional absolute population-weighted $PM_{2.5}$ concentrations (in µg/m$^3$) for 16 EU countries plus Norway and Switzerland, defined in TM5-FASST, as well as the source regions contributing to this pollution. Regional annual averages of population weighted $PM_{2.5}$ concentrations in Europe vary between 2-4 µg/m$^3$ in Northern European countries (like Finland, Norway and Sweden) up to 10-12 µg/m$^3$ for continental Europe. Although most of the computed annual average $PM_{2.5}$ concentrations for Europe are below the World Health Organization Air Quality Guideline of 10 µg/m$^3$ $PM_{2.5}$ (as annual average), these values represent only regional averages while several exceedances in urban areas are often observed in Europe. As further discussed in Sect. 3.2, an additional contribution to $PM_{2.5}$ concentrations comes from the shipping sector, mainly influencing Mediterranean countries (like Italy, Spain and France) and countries facing the North Sea, Baltic Sea and Atlantic Ocean (e.g. Benelux, Sweden, Great Britain, etc.). Transboundary air pollution from external regions contributes by 27% to 75 % and on average by 51 % to $PM_{2.5}$ pollution in European countries. Countries surrounded by oceans are mainly influenced by within-region pollution due to their geographical isolation from other source regions (e.g. Italy, Spain, Great Britain and Norway); therefore the fraction of extra-regional pollution ranges from 27% to 35%. The largest extra-regional contributions are calculated for Hungary (75%, mainly from Austria, Czech Republic, Rest of Central EU, Poland and Germany), Czech Republic (67%, mainly from Poland, Germany and Austria), Austria and Slovenia (66%, mainly from Czech Republic, Germany and Italy), Sweden+Denmark (65%, mainly from Germany, Norway and Poland), Bulgaria (63%, mainly from Romania), and Greece (61%).

The remaining EU countries are both affected by within-region and extra-regional pollution (the latter ranging from 40% to 59%), highlighting the importance of transboundary transport of $PM_{2.5}$ concentrations. For example Switzerland is influenced by the pollution coming from France, Italy and Germany; Rest of Central EU by Poland and Germany; Germany by France and Benelux; Poland by Czech Republic and Germany. Interestingly, Romania, Bulgaria, Greece and Hungary are also significantly affected by the pollution transported from Ukraine and Turkey, which is included in the "rest of the world" contribution of Fig. 2. Our results are consistent with the findings of the latest UNECE Scientific Assessment Report (Maas and Grennfelt, 2016), which highlights the importance of transboundary transport of organic and inorganic PM. As discussed in Sect. 3.4, insights on the uncertainty of within-region and extra-regional contributions to PM2.5 concentrations are provided in Fig. 5 for each TM5-FASST region.

**3.2 Sectorial contributions to $PM_{2.5}$ concentrations**

Figure 3 shows the relative sectorial contributions to anthropogenic $PM_{2.5}$ concentrations for the 56 TM5-FASST receptor regions, separating the fraction of extra-regional (RERER) (shaded colors) and within-region pollution, while Table 1 shows regional average values of sector-specific relative contributions. In most African regions (except Egypt) anthropogenic $PM_{2.5}$ concentrations are mainly produced by emissions in the residential sector. Agriculture is an important sector for Egypt, while Northern Africa is strongly influenced by shipping emissions

in the Mediterranean (30%). $PM_{2.5}$ in emerging economies in Asia, Latin America and Middle East are dominated by $PM_{2.5}$ concentrations from the residential sector, power generation and industrial. Asian countries, China, India, Indonesia and Philippines are mainly influenced by within-region pollution with the largest contributions coming from power, industry and residential sectors. PM2.5 pollution in Japan is characterised by the contribution of local sources like transport and agriculture, but it is also affected by transported pollution from China, especially from the industrial sector. Anthropogenic $PM_{2.5}$ in the remaining Asian countries is influenced by more than 50% by the pollution coming from China (e.g. Vietnam, Malaysia, Thailand, Mongolia, South Korea, Taiwan) or India (e.g. Rest of South Asia and South Eastern Asia) from the power, industry and residential activities. A different picture is seen for Europe where according to our calculations, annual PM concentrations stem mainly from the agricultural and residential sectors with a somewhat lower contribution from the transport sector. In Eastern European countries noticeable contributions are also found from the power and industrial sectors due to the relatively extensive use of polluting fuels like coal. $PM_{2.5}$ concentrations in USA and Canada are mostly from the power, industry and agricultural sectors. In Oceania industry and agriculture are the most important sectors. $PM_{2.5}$ from ship emissions mainly affect coastal areas of North Africa, SE Asia (e.g. in Japan, Taiwan, Malaysia, Indonesia and Philippines), Mediterranean countries (Spain 11%, Italy 5%, France 7% of their corresponding country totals), Northern EU regions (Great Britain 10%, Norway 6%, Sweden and Denmark 10% of their corresponding country totals) and Oceania (22% of the regional total). Over the international areas of sea and air no distinction between within-region and extra-regional concentrations is reported. Further details on within-region and extra-regional concentrations can be found in section S2 of the Supplementary Material.

### 3.3 Gridded $PM_{2.5}$ concentrations

Figure 4 shows the global 1°x1° gridmaps of anthropogenic $PM_{2.5}$ concentrations in 2010 for the reference case as well as the computed contributions from each of the major anthropogenic emission sectors. Anthropogenic $PM_{2.5}$ is ubiquitous globally and covers a range from a $\mu g/m^3$ or less over the oceans and seas to more than 50 $\mu g/m^3$ over Asia. As shown in Fig. 3, the most polluted countries in Asia are China, India and Rest of South Asia (which includes Afghanistan, Bangladesh, Bhutan, Nepal and Pakistan) with annual average anthropogenic $PM_{2.5}$ concentrations ranging from 29 to 40 $\mu g/m^3$; Mongolia and North Korea, Vietnam, South Korea, Rest of South Eastern Asia (including Cambodia, Lao People's Democratic Republic and Myanmar), Thailand, Japan and Taiwan are rather polluted areas with $PM_{2.5}$ concentration in the range of 6 to 14 $\mu g/m^3$. The highest annual $PM_{2.5}$ concentrations in Africa are computed in Egypt (11 $\mu g/m^3$ as annual average), Republic of South Africa (6.1 $\mu g/m^3$ as annual average) and Western Africa (4.0 $\mu g/m^3$ as annual average). The highest pollution in Europe is observed in the Benelux region, Italy and in some of the Eastern countries (e.g. Romania, Bulgaria and Czech Republic), while in Latin America the most polluted areas are Chile (13.7 $\mu g/m^3$ as annual average) and Mexico (4.2 $\mu g/m^3$ as annual average). Middle East, the Gulf region, Turkey, Ukraine and former USSR are also characterised by $PM_{2.5}$ concentrations ranging between 7.5 $\mu g/m^3$ and 9.2 $\mu g/m^3$ as annual averages. Table 2 reports annual average PM2.5 concentrations and the corresponding uncertainty range for each TM5-FASST region as discussed in Sect. 3.4.

The TM5-FASST model (Van Dingenen et al., 2018) has been also validated against concentration estimates derived from the WHO database (WHO, 2011, 2014, 2016) and satellite-

based measurements (van Donkelaar et al., 2010, 2014). The TM5-FASST modeled $PM_{2.5}$ concentrations have been compared to satellite products which are based on aerosol optical depth measurements together with chemical transport model information to retrieve from the total column the information of PM concentrations in the lowest layer of the atmosphere (Boys et al., 2014; van Donkelaar et al., 2010, 2014). The regional comparison of annul mean population weighted concentrations shows consistent results with the satellite based retrievals (e.g. rather good agreement for the globe as a whole, EU and USA within less than 15% deviation, while lower agreement for developing and emerging countries). Supplementary Material section S4 of the paper by van Dingenen et al. (2018) also reports the comparison between the $PM_{2.5}$ concentrations modeled by TM5-FASST and the measured ones reported in the WHO database, showing rather good agreement for Europe, North America and partly China due to the higher accuracy of the measurements. The comparison for Latin America and Africa is much less robust and the scatter possibly highlights a non-optimal modeling of specific sources relevant for these regions by TM5-FASST (e.g. large scale biomass burning) by the TM5-FASST model.

In our work, modelled $PM_{2.5}$ concentrations are in the range of the measurements and satellite-based estimates provided in several literature studies (Brauer et al., 2012;Brauer et al., 2015;Boys et al., 2014;Evans et al., 2013;Van Donkelaar et al., 2016), reporting for the whole Europe annual averaged $PM_{2.5}$ concentrations in the range between 11 and 17 µg/m$^3$, for Asia from 16 to 58 µg/m$^3$, Latin America 7-12 µg/m$^3$, Africa and Middle East 8-26 µg/m$^3$, Oceania 6 µg/m$^3$ and North America 13 µg/m$^3$ (note that measurements and satellite estimates would not separate anthropogenic and natural sources of PM, e.g. dust, large scale biomass burning, while the concentrations in this study consider anthropogenic emissions alone).

In order to understand the origin of global $PM_{2.5}$ concentrations, we look at sector specific maps (Fig. 4). The power and industrial sectors are mainly contributing to PM concentrations in countries having emerging economies and fast development (e.g. Middle East, China and India), while the ground transport sector is a more important source of PM concentrations in industrialised countries (e.g. North America and Europe) and in developing Asian countries. The residential sector is an important source of PM all over the world, also affecting indoor air quality (Ezzati, 2008;Lim et al., 2013;Chafe et al., 2014). PM concentrations in Africa and Asia are strongly influenced by this sector due to the incomplete combustion of rather dirty fuels and solid biomass deployed for domestic heating and cooking purposes. Interestingly, the agricultural sector is strongly affecting pollution in Asia as well as in Europe (Backes et al., 2016; Erisman et al., 2004) and North America, confirming the findings of the UNECE Scientific Assessment Report and several other scientific publications (Maas and Grennfelt, 2016;Pozzer et al., 2017;Tsimpidi et al., 2007;Zhang et al., 2008). The residential and agriculture sectors are less spatially confined, and emissions more difficult to be effectively regulated than point source emissions of the industrial and power sectors (e.g. in Europe the Large Combustion Plant Directive, the National Emission Ceilings or the Industrial Emissions, the Euro norms for road transport, etc.). Finally, shipping is mainly contributing to the pollution in countries and regions with substantial coastal areas, and with ship tracks on the Mediterranean Sea, the Atlantic, Pacific and Indian Oceans, as depicted in Fig. 4.

**3.4 Uncertainty from emissions**

**3.4.1 Propagation of emission uncertainties to anthropogenic $PM_{2.5}$ concentrations**

Table 2, as well as Fig. 5, report the annual average anthropogenic $PM_{2.5}$ concentrations ($\mu g/m^3$) estimated by TM5-FASST with the uncertainty bars representing the upper and lower range of concentrations due to emission inventories uncertainty. The extra-regional contribution to uncertainty is also addressed as well as the contribution of the uncertainty of primary particulate matter emissions to the upper range of $PM_{2.5}$ concentrations (Table 2). Primary PM emissions represent the dominant source of uncertainties, contributing from 45% to 97% to the total uncertainty in anthropogenic $PM_{2.5}$ concentrations for each country/region.

Figure 5 depicts the results of the propagation of the lowest and highest range of emissions including their uncertainty to $PM_{2.5}$ concentrations in Asia (Fig 5*a*) and - in more detail- Europe (Fig 5*b*), highlighting the contribution of within-region and extra-regional $PM_{2.5}$ concentrations and the corresponding uncertainties (error bars). Due to their large sizes, Indian and Chinese $PM_{2.5}$ concentrations and uncertainties are mainly affected by uncertainties from the residential, transport and agricultural sectors within these countries. Interestingly, in South Eastern and Eastern Asia uncertainties in $PM_{2.5}$ are strongly influenced by the Indian residential emissions. On the other hand, $PM_{2.5}$ in Thailand, Japan, Taiwan, South Korea, Mongolia and Vietnam are strongly affected by the uncertainty in the Chinese residential and industrial emissions. Consequently reducing the uncertainties in the Chinese and Indian emission inventories will help improving the understanding the long-range contribution of $PM_{2.5}$ pollution in most of Asian countries.

In Europe, the highest uncertainties in $PM_{2.5}$ concentrations are associated with the emissions from the residential, agriculture and transport sectors. In most of the Central and Eastern European countries modelled $PM_{2.5}$ is strongly affected by the uncertainty of transported extra-regional pollution, produced from the residential, agricultural and transport sectors. Conversely, uncertainties in Norway are dominated by national emissions, mainly from the residential and transport sectors, and in Italy from the residential and agriculture sectors. The remaining European countries are affected both by within-country and imported uncertainties. Fig. 5c represents the results of the propagation of the emissions range including their uncertainty to $PM_{2.5}$ concentrations for North America, Latin America, Oceania and Russia, while Fig 5*d* displays emission uncertainties for Africa, Middle East and the Gulf region. The uncertainty in the USA agricultural and residential emissions affect more than 50% of modelled Canadian $PM_{2.5}$ concentrations and the uncertainty in Mexico and Argentina is influenced by similar magnitudes (30-50%) by neighbouring countries. The uncertainty in within-region emissions, especially from the residential sector, dominates the overall levels of $PM_{2.5}$ uncertainties in Latin America. However, in addition, Chile's own agriculture and power sectors contribute significantly to the overall uncertainty levels. $PM_{2.5}$ levels in most of the African regions are strongly affected by the uncertainty in their own residential emissions, while in Egypt they are mostly influenced by the agricultural sector uncertainties (refer to Fig. 5d). Interestingly, anthropogenic $PM_{2.5}$ in Northern Africa is influenced by uncertainties in Italian emissions uncertainty as well as those from shipping emissions. Conversely, the Middle East and Turkey regions are influenced by a range of extra-regional emission uncertainties (e.g. Middle East is affected by the uncertainty of Turkey, Egypt and the Gulf region, while Turkey by Bulgaria, Gulf region and rest of Central EU).

**3.4.2 Ranking the sector specific contribution to emission uncertainties**

Figure 6 shows the average sector relative contribution to total emission inventory related uncertainty for the main $PM_{2.5}$ concentration precursors and world regions. These contributions can be interpreted as a ranking of the most effective improvements to be taken regionally to better constrain their inventories and reduce the final formation of $PM_{2.5}$ concentrations. The complete overview of all TM5-FASST regions contributions is provided in Fig. S2, where the share of each term of the sum of Eq. 5 $\left( \sigma_{EMI\ i,c,p} * \frac{EMI_{i,c,p}}{EMI_{tot,c,p}} \right)^2$, represents the sector contribution to the uncertainty of each pollutant in each region. $SO_2$ uncertainties mainly derive from the power generation sector especially countries with a dominant coal use; however, substantial contributions are also computed for the industrial sector in South Africa, Asia, Norway, some Latin American countries, Canada and Russian countries. Interestingly, for $SO_2$ some contributions are also observed from the residential sector in Africa and from the transport sector in some Asian countries (e.g. Korea, Vietnam, Indonesia, South Eastern Asia, etc.). Smith et al. (2011) report a range of regional uncertainty for $SO_2$ emissions up to 30%, while our estimates are slightly higher (up to 50%). NOx emissions uncertainty mainly stems from the transport sector, although some contributions are also seen from power generation in Russia, countries strongly relying on gas (e.g. Russia), the Middle East and the residential sector in Africa. Depending on the region, CO uncertainty (not shown) is dominated by either the transport or residential (particularly in Africa and Asia) sectors and for some regions by a similar contribution of these two sectors. NMVOC emission uncertainties mainly derive from poorly characterized industrial, transport and residential activities due to the complex mixture and reactivity of such pollutants. As expected, $NH_3$ emission uncertainty is dominated by the agricultural sector which appears to be less relevant for all other pollutants. Among all air pollutants, primary $PM_{2.5}$ represents one of the most uncertain pollutant due to very different combustion conditions, different fuel qualities and lack of control measures (Klimont et al., 2017).

Primary particulate matter emissions should be mainly improved for the residential, transport and in particular industrial sectors. Black carbon emission inventories should be better characterised in Europe, Japan, Korea, Malaysia etc. for the transport sector, where the higher share of diesel used as fuel for vehicles leads to higher BC emissions; in addition, BC emissions from the residential sector require further effort to better define EFs for the different type of fuels used under different combustion conditions. To constrain and improve particulate organic matter emissions, efforts should be dedicated to improve residential emissions estimates. Therefore, in the following section, we try to assess one of the major sources of uncertainty in the residential emissions in Europe which is the use of solid biofuel.

### 3.4.3 Assessing the uncertainty in household biofuel consumption with an independent inventory in Europe

The combustion of solid biomass (i.e. biofuel) for household heating and cooking purposes is one of the major sources of particulate matter emissions in the world. Wood products and residues are widely used in the residential sector, but national reporting often underestimates the emissions from this sector, due to the fact that often informal economic wood sales are not accurately reflected in the official statistics of wood consumption (AD) (Denier Van Der Gon et al., 2015). An additional uncertainty is related to the lack of information in the inventory regarding the emission factors (EF) variability, which depends on the combustion efficiency and type of wood (Weimer et al., 2008;Chen et al., 2012). In our work we estimate the uncertainty

attributable to wood combustion in the residential sector ($\sigma_{AD,RES\_bio}$) by comparing it to the recent TNO RWC (Netherlands Organization for Applied Scientific Research, Residential Wood Combustion) inventory of Denier van der Gon et al. (2015), which includes a revised biomass fuel consumption with the corresponding EDGARv4.3.2 activity data (Janssens-Maenhout et al., 2017), as shown in Table S4. In the TNO RWC inventory, wood use for each country has been updated comparing the officially reported per capita wood consumption data (from GAINS (Greenhouse Gas - Air Pollution Interactions and Synergies) and IEA (International Environmental Agency)) with the expected specific wood use for a country including the wood availability information (Visschedijk et al., 2009;Denier Van Der Gon et al., 2015). We can therefore assume that the TNO RWC inventory represents an independent estimate of wood consumption in the residential sector, allowing a more precise uncertainty estimation of the AD for this sector. Assuming that emissions are calculated as the product of AD and EF, the corresponding uncertainty can be calculated with Eq. 4, where $\sigma_{AD}$ ranges from 5 to 10% for European countries and Russia as reported for international statistics (Olivier et al., 2016). We can therefore calculate the residential emission factors uncertainty of each individual pollutant ($\sigma_{EF,p}$) from Eq. 4. In addition, based on the comparison of the recent estimates of wood consumption provided by TNO RWC AD, which should match better with observations and the EDGARv4.3.2 ones, we can evaluate the mean normalized absolute error (MNAE) considering all $N$ countries, following Eq. 6 (Yu et al., 2006), which represents our estimate of $\sigma_{AD,RES\_bio}$.

$$\text{MNAE} = \frac{1}{N} * \sum_j^N \frac{\left|TNO\ RWC_j - EDGARv4.3.2_j\right|}{TNO\ RWC_j} \qquad \text{(Eq.6)}$$

We estimate a value of $\sigma_{AD,RES\_bio}$ of 38.9% which is much larger compared to the 5-10% uncertainty reported for the fuel consumption of the international statistics ($\sigma_{AD}$). The issue of biofuel uncertainty mainly affects rural areas where wood is often used instead of fossil fuel.

Then, using Eq. 4 and the calculated $\sigma_{AD,RES\_bio}$ and $\sigma_{EF,p}$, we can evaluate a new $\sigma_{EMI,p,RES\_bio}$ for the residential sector including the uncertainty of the AD due to the use of wood as fuel for this sector, as reported in Table S5. Comparing the results shown in Table S5 with the factor of two uncertainty values expected for PM emissions from the residential sector (Janssens-Maenhout et al., 2015), we derive that the uncertainty associated with the emission factors for biomass combustion in the residential sector is the dominant source of uncertainty compared to the uncertainty in wood burning activity data. Large increases in reported biomass usage for domestic use has been noted in IEA energy statistics for some European countries (IEA, 2013,2014,2015,2016) and further increases are expected as countries are shifting their methodologies to estimate biofuel activity data away from fuel sales statistics to a modelling approach based on energy demand. In addition, several EU countries are increasing the use of biomass in order to accomplish the targets set in the context of the renewable energy directive (2009/28/EC) as reported in their national renewable energy action plans (http://ec.europa.eu/energy/node/71). When comparing the UNFCCC and the TNO RWC data, a higher value of $\sigma_{AD,RES\_bio}$ is obtained (59.5% instead of 38.9%), although its effect on the final residential emission uncertainty is less strong, as shown in Table S6. Table 3 shows the impact of biofuel combustion uncertainty in the residential sector on PM$_{2.5}$ concentrations. Upper-end

uncertainties indicate that $PM_{2.5}$ concentrations could be between 2.6 and 3.7 times larger than those derived from the HTAP_v2.2 inventory.

**4 Health impact assessment**

Annual population weighted $PM_{2.5}$ concentrations represent the most robust and widely used metric to analyse the long-term impacts of particulate matter air pollution on human mortality (Pope and Dockery, 2006;Dockery, 2009). As described in Sect 2.5 and S5 of the paper by Van Dingenen et al. (2018), the mortality estimation in TM5-FASST is based on the integrated exposure-response functions defined by Burnett et al. (2014). The increased risk from exposure to air pollution is estimated using exposure-response functions for five relevant deaths causes, namely Ischemic heart disease (IHD), Cerebrovascular Disease (CD, stroke), Chronic Obstructive Pulmonary Disease (COPD), Lung Cancer (LC), Acute Lower Respiratory Infections (ALRI). The relative risk (RR) represents the proportional increase in the assessed health outcome due to a given increase in $PM_{2.5}$ concentrations (Burnett, 2014).

In this section, we investigate the impact of total and sector-specific anthropogenic population weighted $PM_{2.5}$ concentrations on health and we show comparisons with mortality estimates provided by WHO and recent scientific publications (Silva et al., 2016). Figure 7 represents the premature deaths (PD) distribution due to air pollution, using population weighted $PM_{2.5}$ concentrations and representative for anthropogenic emissions in the year 2010. The most affected areas are China and India, but also some countries of Western Africa and urban areas in Europe (in particular in the Benelux region and Eastern Europe). Our computations indicate that annual global outdoor premature mortality due to anthropogenic $PM_{2.5}$ amounts to 2.1 million premature deaths, with an uncertainty range related to emission uncertainty of 1-3.3 million deaths/year. In our work we only evaluate how the uncertainty of emission inventories influences the health impact estimates focusing on the interregional aspects (i.e. we do not evaluate effects of misallocation of sources within regions) and not all the other sources of uncertainties, such as the uncertainty of concentration-response estimates, of air quality models used to estimate particulate matter concentrations, etc. An overview of the propagation of the uncertainty associated with an ensemble of air quality models to health and crop impacts is provided by Solazzo et al. (2018). Solazzo et al. find in their analysis over the European countries a mean number of PDs due to exposure to $PM_{2.5}$ and ozone of approximately 370 thousands (inter-quantile range between 260 and 415 thousand). Moreover, they estimate that a reduction in the uncertainty of the modelled ozone concentration by 61% - 80% (depending on the aggregation metric used) and by 46% for $PM_{2.5}$, produces a reduction in the uncertainty in premature mortality and crop loss of more than 60%. However, we show here that the often neglected emission inventories' uncertainty provides a range of premature deaths of ±1.1 million at the global scale, which is in the same order of magnitude of the uncertainty of air quality models and concentration-response functions (Cohen et al., 2017). In 2010, using our central estimate, 82% of the PDs occur in fast growing economies and developing countries, especially in China with 670 thousand and India with an almost equal amount of 610 thousand PD/year. Table 4 summarizes our estimates of premature mortality for aggregated world regions, with Europe accounting for 210 thousand PD/year and North America 100 thousand PD/year.

Our results are comparable with Lelieveld et al. (2015) and Silva et al. (2016) who, using the same Burnett et al. (2014) methodology, estimate a global premature mortality of 2.5 and 2.2 million people, respectively, due to air quality in 2010 for the same anthropogenic sectors.

However, a recent work published by Cohen et al. (2017) estimates a higher value of global mortality (3.9 million PD/year) mainly due to a lower minimum risk exposure level set in the exposure response function, the inclusion of the urban increment calculation and the contribution of natural sources. When comparing mortality estimates we need to take into account that several elements affect the results, like the resolution of the model, the urban increment subgrid adjustment (including information on urban and rural population, refer to Van Dingenen et al. 2018), the inclusion or not of natural components, the impact threshold value used, and RR functions. In this study, we use the population weighted $PM_{2.5}$ concentration (excluding natural components) at 1x1 degree resolution as metric for estimating health effects due to air, with a threshold value of 5.8 µg/m$^3$, no urban increment adjustment, and relative risk functions accordingly with Burnett et al. (2014). We also estimate that 7 % of the global non accidental mortalities from the Global Burden of Disease (http://vizhub.healthdata.org/gbd-compare; Forouzanfar et al. (2015)) are attributable to air pollution in 2010; 8.6% of total mortality in Europe is due to air pollution, ranging from less than 1% up to 17% depending on the country; similarly, Asian premature mortality due to air quality is equal to 8.7% of total Asian mortality, with 10.6% contribution in China and 8.5% in India. Lower values are found for African countries and Latin America where other causes of mortalities are still dominant compared to developed countries.

Table 5 shows the number of premature deaths for each receptor region, highlighting the premature mortality induced within the country itself and outside the receptor region. The PD induced by Chinese and Indian emissions are mainly found within these two countries; however, the annual PDs caused by China and India in external regions contribute an additional 700 thousand and ca 500 thousand PD/year, respectively, representing more than 50% of the global mortality. Clearly, reducing emissions and emission uncertainties in these two regions will have therefore the largest over-all benefit on global air quality improvement as well as on global human health. As explained in Sect. 3.1, PDs attributed to internal/external emissions are directly linked (proportional) to the internal/external PM2.5 contributions. For most of the TM5-FASST regions, PDs due to anthropogenic emissions within the source region are higher than the extra-regional contributions. However, there are marked exceptions, such as Hungary, Czech Republic, Mongolia, etc., where the extra-regional and within-region contributions to mortality are at least comparable. For instance, Hungary and Czech Republic are strongly influenced by polluted regions in Poland (mainly); likewise Mongolia is affected by the vicinity of sources in China. The Gulf region produces a lot of its own pollution, but is also influenced by transport from Africa and Eurasia as reported by Lelieveld et al. (2009).

Detailed information on the premature deaths for each TM5-FASST region and the contributing anthropogenic emission sectors is shown in Figs. 8a and 8b. Health effects induced by air quality in industrialized countries are mainly related with agriculture (32.4% of total mortality or 68 thousand PD/year), residential combustion (17.8% or 37 thousand PD/year) and road transport (18.7% or 39 thousand PD/year) for Europe and with power generation (26.4% or 26 thousand PD/year), industry (19% or 19 thousand PD/year), residential (17% or 17 thousand PD/year) and agriculture (24.0% or 24 thousand PD/year) for North America. The health impacts observed in most Western EU countries is due both to within-regions and extra-regional pollution, while in several Eastern EU countries the impact of neighbouring countries is even larger compared to within-region pollution. The premature deaths induced by international shipping emissions represent 5.5% of total EU PD, in the range the results of Brandt et al. (2013a) (ca 50 thousand

PDs). PM related mortality in developing countries and fast growing economies is mostly affected by industrial (up to 42% in China or 279 thousand PD/year) and residential activities (ranging from 27% in China and 76% in Western Africa), and also by power generation (up to 24% in India or 113000 PD/year). Chinese emissions have a strong impact on China, Japan, Vietnam, Mongolia+Korea, Thailand while the Indian emissions impact the rest of South and South Eastern Asia. Reducing Chinese and Indian emissions will reduce the PM related mortality in almost all countries in Asia. Our results are in agreement with the study of Oh et al. (2015) where they highlight the role of transported pollution from China in affecting Korean and other South Eastern Asian countries $PM_{2.5}$ concentrations and health effects, as well as the need of international measures to improve air quality.

**Conclusions**

We coupled the global anthropogenic emission estimates provided by the HTAP_v2.2 inventory for 2010 (merging national and regional inventories) to the global source receptor model TM5-FASST, to study $PM_{2.5}$ concentrations and the corresponding health impacts, including an evaluation of the impacts of uncertainties in national emission inventories. Annual and regionally averaged anthropogenic $PM_{2.5}$ concentrations, corresponding to the 2010 emissions, vary between ca 1 and 40 $\mu g/m^3$, with the highest annual concentrations computed in China (40 $\mu g/m^3$, range: 22.4 - 76.6 $\mu g/m^3$) India (35 $\mu g/m^3$, range: 16.6 - 73.4 $\mu g/m^3$), North America (8 $\mu g/m^3$, range: 4.4 - 14.4 $\mu g/m^3$) and Europe (on average ca 8 $\mu g/m^3$, range: 5 - 18 $\mu g/m^3$). Anthropogenic $PM_{2.5}$ concentrations are mainly due to emissions within the source region, but extra-regional transported air pollution can contribute by up to 40%, e.g. from China to SE Asia, from EU to Russia, etc.). Moreover, due to the transport of PM between European countries, EU wide directives can help improving the air quality across Europe.

For our analysis we aggregate our results derived from 56 TM5-FASST source regions, into 10 global regions to facilitate the comparison of results in regions of more equal size. The relative contribution of anthropogenic sectors to $PM_{2.5}$ concentrations varies in different regions. In Europe in 2010, the agriculture and residential combustion sectors contribute strongest to $PM_{2.5}$ concentrations and these sectors are also associated with relatively large emission uncertainties. $PM_{2.5}$ concentrations in China and other emerging economies are predominantly associated with the power generation, industry and residential activities.

Using the HTAP_v2.2 emission inventory and TM5-FASST, we also evaluate how the uncertainty in sectors and regions propagates into $PM_{2.5}$. The aim of our analysis is to provide insights on where improvement of country emission inventories would give largest benefits, because of their highest uncertainty and highest contribution to the formation of $PM_{2.5}$ concentrations. The uncertainty of PM concentrations depends in variable proportions to the uncertainties of the emissions within receptor regions, and surrounding regions. We show that reducing the uncertainties in the Chinese and Indian emission inventories (e.g. from industry and residential sectors) will be highly relevant for more accurate quantification of the contribution of the long-range sources to $PM_{2.5}$ pollution in most of Asian countries. Here we demonstrate how analysis of uncertainties in national/regional sectorial emission inventories can further inform coordinated transboundary and sector-specific policies to significantly improve global air quality. Among all anthropogenic emission sectors, the combustion of biomass for household purposes represents one of the major sources of uncertainties in emission inventories both in

terms of wood consumption and emission factor estimates. Further effort is therefore required at national level to better characterize this source.

Finally, we analyse the air quality effects on health. Global health effects due to $PM_{2.5}$ concentrations calculated with TM5-FASST and anthropogenic emissions in 2010 are estimated to be ca 2.1 million premature deaths/year, but the uncertainty associated with emission ranges between 1-3.4 million deaths/year, with the largest fraction of PD (82%) in developing countries.

Acknowledgements:

Role of authors. The authors acknowledge financial support by the Administrative Arrangement AMITO2 with DG ENV. This analysis was inspired by HTAP2 joint studies on regional contributions to global air pollution. This publication is an application of the companion paper 'TM5-FASST: a global atmospheric source-receptor model for rapid impact analysis of emission changes on air quality and short-lived climate pollutants' by Van Dingenen et al. (2018).

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

| | POWER | INDUSTRY | TRANSPORT | RESIDENTIAL | AGRICULTURE | SHIPPING |
|---|---|---|---|---|---|---|
| **Africa** | 26.7 | 16.1 | 3.6 | 37.9 | 8.2 | 7.4 |
| **China+** | 18.3 | 42 | 7.5 | 23.1 | 8.8 | 0.3 |
| **India+** | 20.8 | 19.4 | 11.4 | 45.2 | 3 | 0.2 |
| **SE Asia** | 17.1 | 35.9 | 9 | 27.2 | 7.4 | 3.4 |
| **Europe** | 15.1 | 14.3 | 18.7 | 19.7 | 27.7 | 4.4 |
| **Latin America** | 25.6 | 33.7 | 6.6 | 18.9 | 12.6 | 2.6 |
| **Middle East** | 37.9 | 25.2 | 9.7 | 11.7 | 13.7 | 1.8 |
| **Russia** | 23.5 | 30.9 | 8.6 | 13 | 23.1 | 0.8 |
| **North America** | 20.4 | 23.5 | 10.8 | 15.5 | 25.6 | 4.2 |
| **Oceania** | 13.9 | 30.7 | 5.1 | 9.8 | 18.6 | 21.8 |

1  **Table 2 - Annual average PM$_{2.5}$ concentrations (μg/m$^3$) with upper and lower range in brackets due to**
2  **emission inventories uncertainty (1 standard deviation, σ). The upper and lower range of PM$_{2.5}$**
3  **concentrations are calculated as the reference concentrations multiplied and divided by (1+σ) respectively.**
4  **The third column reflects the fractional uncertainty due to the contribution of primary PM$_{2.5}$ emissions.**

| World region | TM5-FASST region | PM$_{2.5}$ concentration (μg/m$^3$) | Fraction of uncertainty due to primary PM emissions (%) |
|---|---|---|---|
| Asia | South Korea | 13.8 (8.3 - 24.9) | 71% |
| | Japan | 6.9 (3.8 - 13.3) | 84% |
| | Mongolia+ North Korea | 14.6 (9.0 - 25.9) | 75% |
| | China | 39.9 (22.4 - 76.6) | 78% |
| | Taiwan | 6.4 (3.7 - 10.9) | 77% |
| | Rest of South Asia | 29.3 (13.9 - 64.9) | 87% |
| | India | 34.7 (16.6 - 73.4) | 86% |
| | Indonesia | 2.4 (1.3 - 4.6) | 86% |
| | Thailand | 8.0 (5.1 - 12.6) | 88% |
| | Malaysia | 3.1 (1.8 - 5.2) | 85% |
| | Philippines | 2.0 (1.1 - 3.8) | 80% |
| | Vietnam | 14.2 (7.0 - 30.4) | 92% |
| | Rest of South Eastern Asia | 8.6 (4.6 - 17.6) | 89% |
| Europe | Austria+Slovenia | 8.4 (4.0 - 19.6) | 59% |
| | Switzerland | 10.1 (4.9 - 23.3) | 52% |
| | Benelux | 10.1 (5.2 - 22.7) | 59% |
| | Spain+Portugal | 5.4 (3.4 -9.4) | 77% |
| | Finland | 2.6 (1.3 - 5.8) | 66% |
| | France | 9.3 (5.0 - 19.0) | 69% |
| | Great Britain+Ireland | 6.1 (3.2 - 13.0) | 66% |
| | Greece+Cyprus | 7.6 (4.8 - 12.7) | 74% |
| | Italy+Malta | 11.8 (6.2 - 25.2) | 64% |
| | Germany | 9.3 (5.0 - 20.0) | 54% |
| | Sweden+Denmark | 4.1 (2.2 - 8.4) | 65% |
| | Norway | 2.4 (1.2 - 5.4) | 89% |
| | Bulgaria | 10.6 (5.4 - 21.6) | 66% |
| | Hungary | 9.2 (4.4 - 21.6) | 60% |
| | Poland+Baltic | 7.9 (3.6 - 20.2) | 54% |
| | Rest of Central EU | 9.3 (4.7 – 20.4) | 63% |
| | Czech Republic | 10.3 (4.8 - 25.1) | 58% |
| | Romania | 10.9 (5.5 - 24.1) | 67% |

| World region | TM5-FASST region | PM₂.₅ concentration (μg/m³) | Fraction of uncertainty due to primary PM emissions (%) |
|---|---|---|---|
| Africa | Northern Africa | 4.2 (2.3 - 4.3) | 80% |
| | Egypt | 11.0 (5.0 - 27.8) | 46% |
| | Western Africa | 4.0 (1.7 - 10.2) | 96% |
| | Eastern Africa | 2.7 (1.4 - 5.7) | 89% |
| | Southern Africa | 1.0 (0.5 - 2.2) | 90% |
| | Rep. of South Africa | 6.1 (3.1 - 12.5) | 84% |
| Gulf/ Middle East | Middle East | 9.2 (5.4 - 17.8) | 58% |
| | Turkey | 8.7 (4.9 - 17.1) | 67% |
| | Gulf region | 7.8 (4.7 - 14.5) | 57% |
| Latin America | Brazil | 1.6 (1.1 - 2.6) | 85% |
| | Mexico | 4.2 (2.1 - 9.2) | 62% |
| | Rest of Central America | 2.0 (1.0 - 4.0) | 78% |
| | Chile | 13.7 (7.3 - 29) | 70% |
| | Argentina+Uruguay | 1.1 (0.7 - 1.9) | 77% |
| | Rest of South America | 2.4 (1.6 - 3.9) | 69% |
| NA | Canada | 4.3 (2.4 - 8.3) | 66% |
| | USA | 7.8 (4.4 - 14.4) | 71% |
| Russia | Kazakhstan | 4.9 (3.2 - 8.9) | 62% |
| | Former USSR Asia | 7.5 (4.0 - 17.6) | 49% |
| | Russia (EU) | 3.3 (1.9 - 6.7) | 57% |
| | Russia (Asia) | 2.7 (1.7 - 5.1) | 64% |
| | Ukraine | 7.8 (4.2 - 15.9) | 65% |
| Oceania | Australia | 1.1 (0.8 - 1.4) | 84% |
| | New Zealand | 0.3 (0.1 - 0.5) | 60% |
| | Pacific Islands | 0.2 (0.1 - 0.4) | 75% |

**Table 3 - PM$_{2.5}$ concentrations due to the residential sector emissions in Europe, European part of Russia, Ukraine and Turkey and uncertainty range including the uncertainty in the biomass consumption for the same sector.**

| | PM$_{2.5}$ (µg/m$^3$) - RESIDENTIAL | PM$_{2.5}$ (µg/m$^3$)- RESIDENTIAL including biomass uncertainty |
|---|---|---|
| **Romania** | 3.1 | 11.4 |
| **Czech Republic** | 2.9 | 10.7 |
| **Italy+Malta** | 3.6 | 10.6 |
| **Rest of Central EU** | 2.5 | 9.2 |
| **Hungary** | 2.5 | 9.1 |
| **Bulgaria** | 2.3 | 8.6 |
| **Poland+Baltic** | 2.2 | 8.3 |
| **Austria+Slovenia** | 2.2 | 7.1 |
| **Ukraine** | 1.7 | 6.1 |
| **France** | 2.1 | 6.0 |
| **Turkey** | 1.7 | 5.9 |
| **Norway** | 1.3 | 4.1 |
| **Switzerland** | 1.4 | 3.9 |
| **Greece+Cyprus** | 1.2 | 3.8 |
| **Germany** | 1.1 | 3.0 |
| **Spain+Portugal** | 1.0 | 2.7 |
| **Benelux** | 0.9 | 2.5 |
| **Sweden+Denmark** | 0.8 | 2.4 |
| **Finland** | 0.7 | 2.1 |
| **Great Britain+Ireland** | 0.7 | 1.8 |
| **Russia (EU)** | 0.4 | 1.3 |

**Table 4 – Absolute and population size normalized number of premature deaths/year due to anthropogenic PM$_{2.5}$ air pollution in world regions and corresponding uncertainty range.**

| | PD (deaths/year) |
|---|---|
| China+ | $6.7 \cdot 10^5$ ($3.5 \cdot 10^5$ - $1.0 \cdot 10^6$) |
| India+ | $6.1 \cdot 10^5$ ($2.7 \cdot 10^5$ - $9.6 \cdot 10^5$) |
| Europe | $2.6 \cdot 10^5$ ($1.4 \cdot 10^5$ - $4.8 \cdot 10^5$) |
| SE Asia | $1.5 \cdot 10^5$ ($8.3E \cdot 10^4$ - $2.5 \cdot 10^5$) |
| Russia | $1.1 \cdot 10^5$ ($6.7 \cdot 10^4$ - $2.4 \cdot 10^5$) |
| North America | $1.0 \cdot 10^5$ ($5.5 \cdot 10^4$ - $1.7 \cdot 10^5$) |
| Africa | $7.4 \cdot 10^4$ ($3.4 \cdot 10^4$- $1.6 \cdot 10^5$) |
| Middle East | $5.6 \cdot 10^4$ ($3.2 \cdot 10^4$ - $9.7 \cdot 10^4$) |
| Latin America | $2.6 \cdot 10^4$ ($1.4 \cdot 10^4$ - $5.3 \cdot 10^4$) |
| Oceania | $5.5 \cdot 10^1$ ($3.4 \cdot 10^1$ - $1.2 \cdot 10^2$) |

**Table 5 – Number of premature deaths for each receptor region including the within-region and extra-regional attribution based on PM2.5 'central' estimates, which do not consider uncertainty. For the RERER metric refer also to Table S2.**

| world regions | TM5-FASST region name | PDs in receptor region (deaths/year) | Within-region PDs (deaths/year) | Extra-regional PDs (deaths/year) |
|---|---|---|---|---|
| Africa | Eastern Africa | 16705 | 8218 | 8487 |
| Africa | Egypt | 17282 | 11380 | 5902 |
| Africa | Northern Africa | 5424 | 3427 | 1997 |
| Africa | Rep. of South Africa | 9065 | 8797 | 268 |
| Africa | Southern Africa | 345 | 322 | 23 |
| Africa | Western Africa | 25081 | 19785 | 5296 |
| Asia | China | 655870 | 643129 | 12741 |
| Asia | Indonesia | 17780 | 14803 | 2977 |
| Asia | India | 474660 | 412298 | 62362 |
| Asia | Japan | 25636 | 15181 | 10455 |
| Asia | South Korea | 25295 | 7510 | 17784 |
| Asia | Mongolia+North Korea | 12657 | 4076 | 8581 |
| Asia | Malaysia | 2014 | 1058 | 957 |
| Asia | Philippines | 121 | 94 | 27 |
| Asia | Rest of South Asia | 134280 | 67170 | 67110 |
| Asia | Rest of South Eastern Asia | 23316 | 3814 | 19502 |
| Asia | Thailand | 21231 | 10495 | 10736 |
| Asia | Taiwan | 3443 | 1028 | 2415 |
| Asia | Vietnam | 30750 | 20286 | 10464 |
| Europe | Austria+Slovenia | 6073 | 1806 | 4267 |
| Europe | Bulgaria | 4739 | 1709 | 3030 |
| Europe | Benelux | 9090 | 4201 | 4889 |
| Europe | Switzerland | 3200 | 1568 | 1632 |
| Europe | Czech Republic | 7936 | 2696 | 5240 |
| Europe | Germany | 36256 | 18595 | 17661 |
| Europe | Spain+Portugal | 11291 | 8487 | 2804 |
| Europe | Finland | 0 | 0 | 0 |
| Europe | France | 22046 | 13320 | 8727 |
| Europe | Great Britain+Ireland | 13949 | 9459 | 4490 |
| Europe | Greece+Cyprus | 3117 | 1133 | 1984 |
| Europe | Hungary | 14211 | 3820 | 10391 |
| Europe | Italy+Malta | 24417 | 16312 | 8105 |
| Europe | Norway | 674 | 516 | 158 |
| Europe | Poland+Baltic | 28686 | 15877 | 12809 |

| | | | | |
|---|---|---|---|---|
| **Europe** | **Rest of Central EU** | 6764 | 3418 | 3346 |
| **Europe** | **Romania** | 14155 | 6979 | 7176 |
| **Europe** | **Sweden+Denmark** | 2650 | 1021 | 1629 |
| **Latin America** | **Argentina+Uruguay** | 133 | 75 | 58 |
| **Latin America** | **Brazil** | 4261 | 3968 | 293 |
| **Latin America** | **Chile** | 3332 | 3283 | 49 |
| **Latin America** | **Mexico** | 10478 | 8447 | 2031 |
| **Latin America** | **Rest of Central America** | 3413 | 2772 | 640 |
| **Latin America** | **Rest of South America** | 4489 | 4164 | 325 |
| **Middle East** | **Gulf region** | 15176 | 11225 | 3951 |
| **Middle East** | **Middle East** | 6784 | 2804 | 3980 |
| **Middle East** | **Turkey** | 34151 | 24191 | 9960 |
| **North America** | **Canada** | 3262 | 1491 | 1771 |
| **North America** | **USA** | 97877 | 90176 | 7701 |
| **Oceania** | **Australia** | 28 | 25 | 3 |
| **Oceania** | **New Zealand** | 24 | 15 | 9 |
| **Oceania** | **Pacific Islands** | 3 | 1 | 2 |
| **Russia** | **Kazakhstan** | 3389 | 1100 | 2290 |
| **Russia** | **Former USSR Asia** | 10757 | 6420 | 4337 |
| **Russia** | **Russia (Asia)** | 1348 | 601 | 746 |
| **Russia** | **Russia (EU)** | 25149 | 12704 | 12445 |
| **Russia** | **Ukraine** | 71724 | 44604 | 27120 |

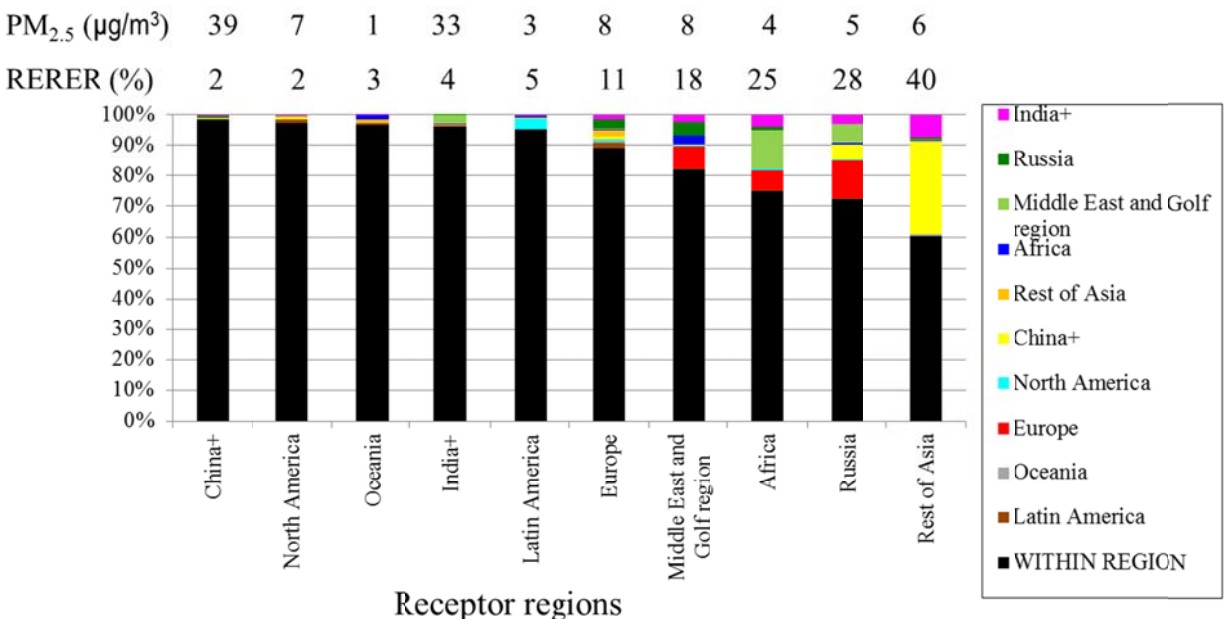

**Figure 1 – Within-region vs. imported extra-regional anthropogenic population weighted PM$_{2.5}$ concentrations [%] for aggregated world regions based on 'central' estimates. Annual average population weighted anthropogenic concentrations (in µg/m$^3$) are reported on top of each bar together with the RERER metric (%). Shipping emissions were not included.**

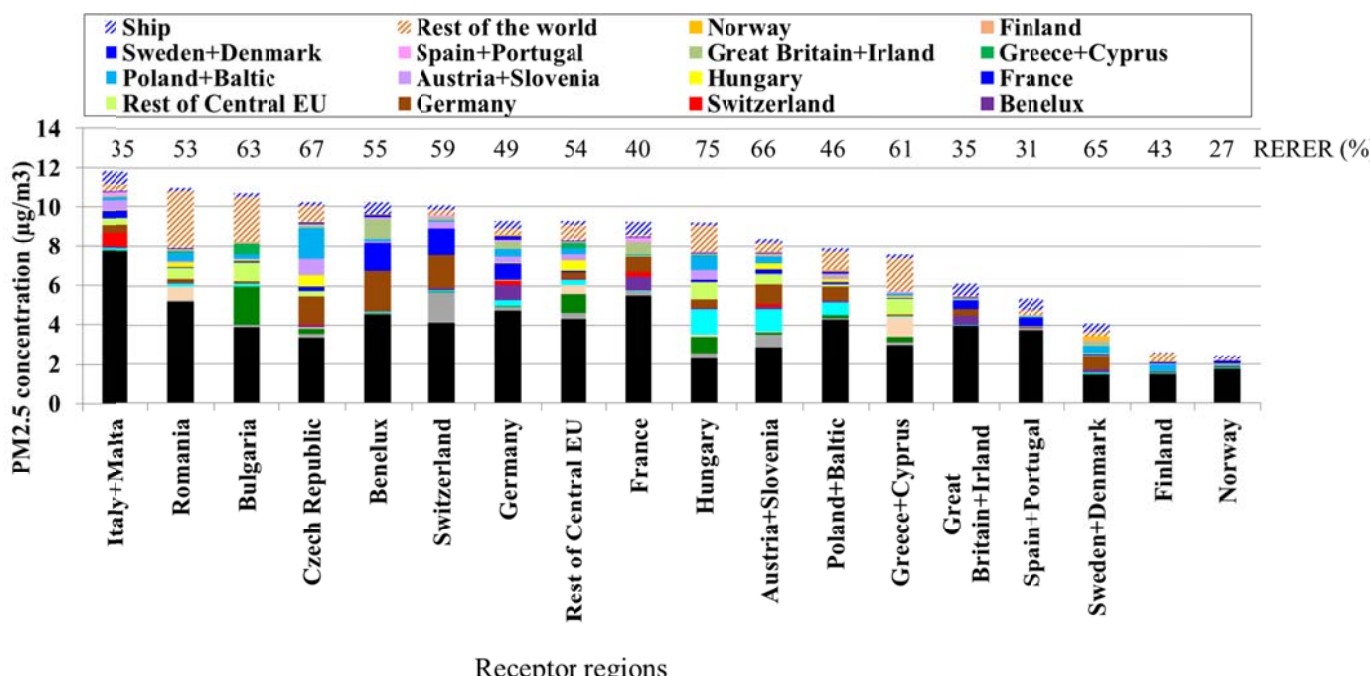

**Figure 2 – Anthropogenic PM$_{2.5}$ concentrations in 18 countries and sub-regions in Europe separated in within-region and extra-regional contributions. The RERER metric (%) is reported on top of each bar.**

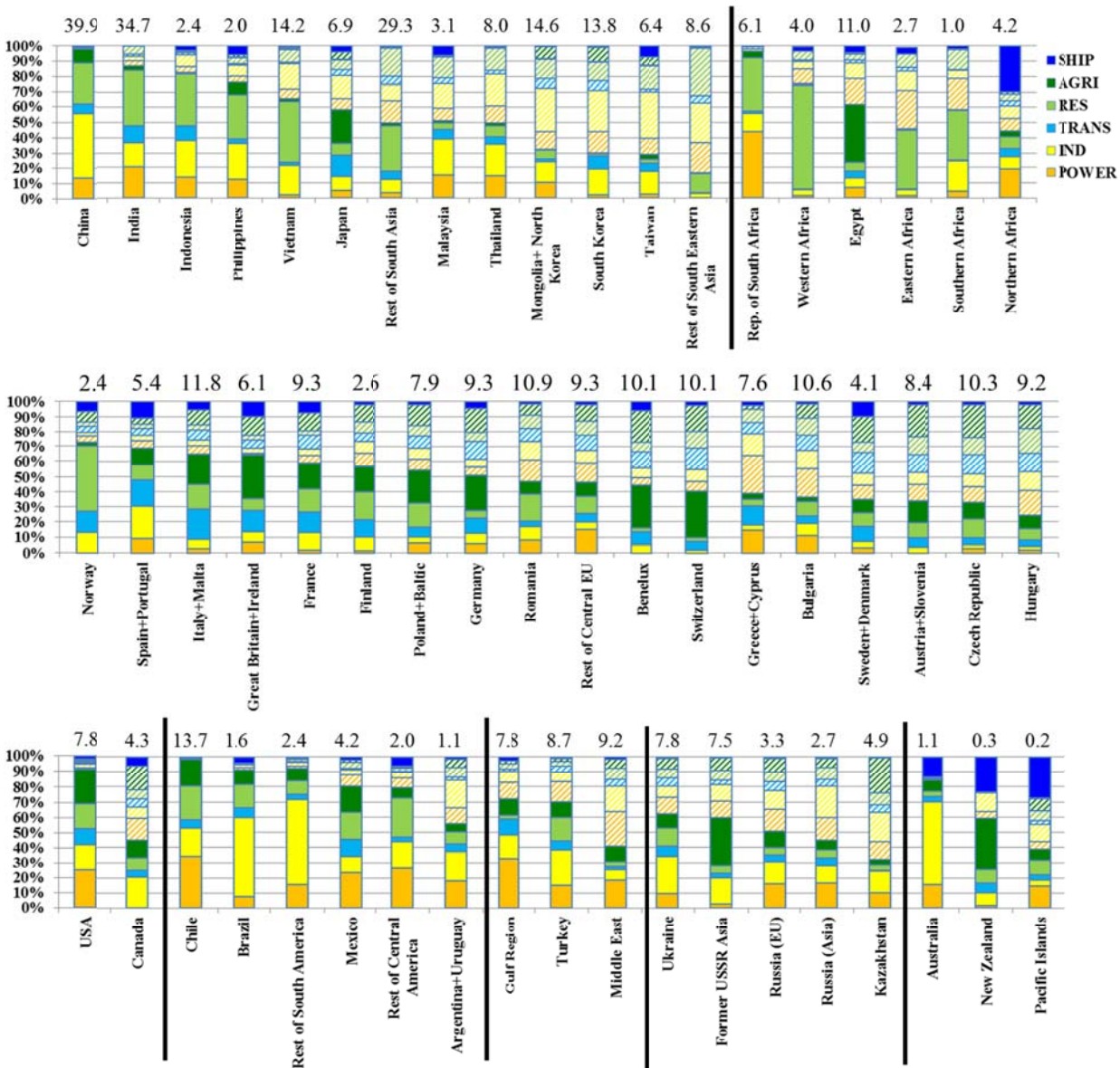

**Figure 3 - Fraction of within-region and extra-regional (shaded areas) anthropogenic PM$_{2.5}$ concentrations separate by sector for receptor region within the macro-regions: Asia and Africa (upper panel), Europe (middle panel), North America, Latin America, Middle East, Russia and Oceania (lower panel). Annual averaged anthropogenic concentrations (in μg/m$^3$) are reported on top of each bar. The RERER metric (%) for the 56 TM5-FASST regions is also reported in Table S2.**

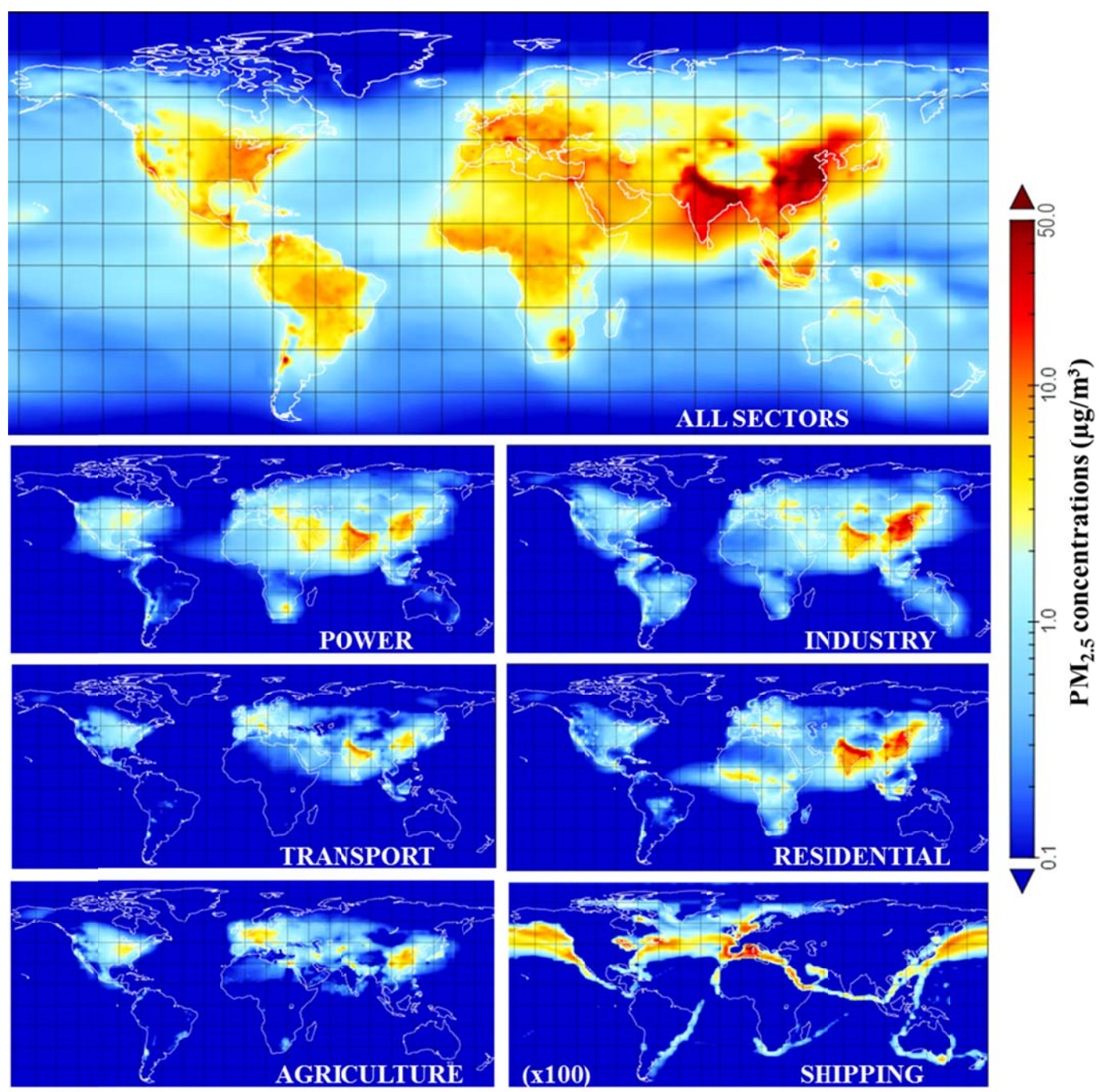

**Figure 4 – Total anthropogenic PM$_{2.5}$ concentrations (µg/m$^3$) and sectorial contributions using 2010 emissions.**

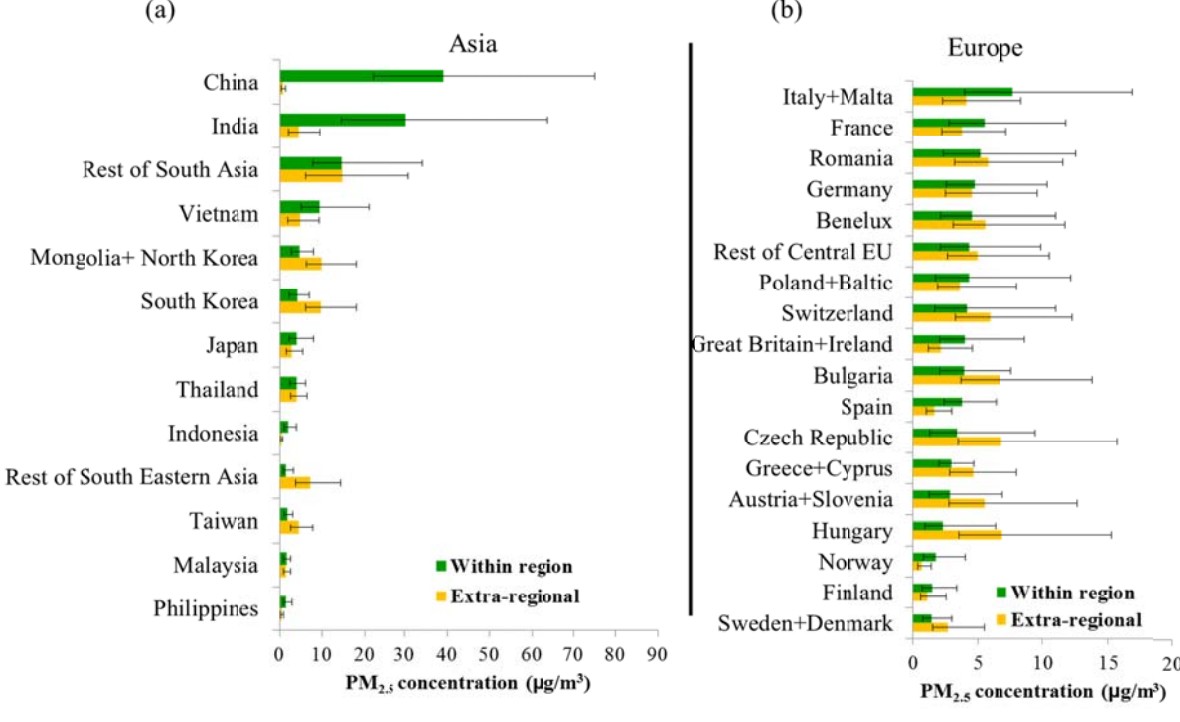

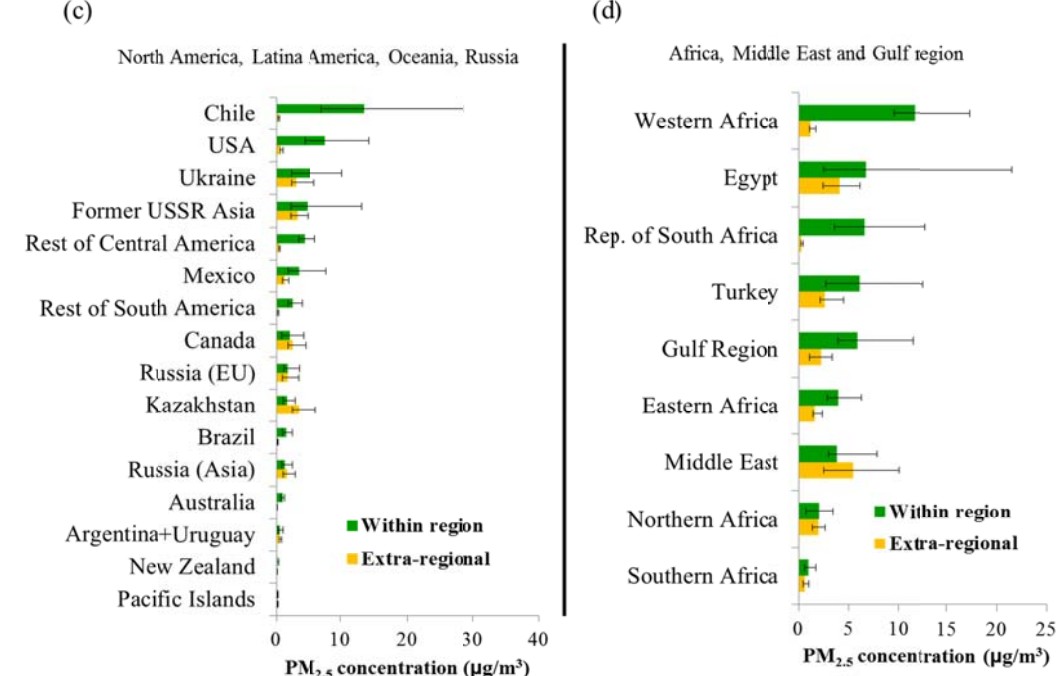

**Figure 5 - Within-region and extra-regional anthropogenic PM$_{2.5}$ concentrations and emission related uncertainties for Asia (panel *a*), Europe (panel *b*), North America, Latin America, Oceania and Russia (panel *c*) and Africa, Gulf region and Middle East (panel *d*). The error bars are calculated multiplying and dividing the reference emissions by (1+σ) as discussed in Sect. 2.3.**

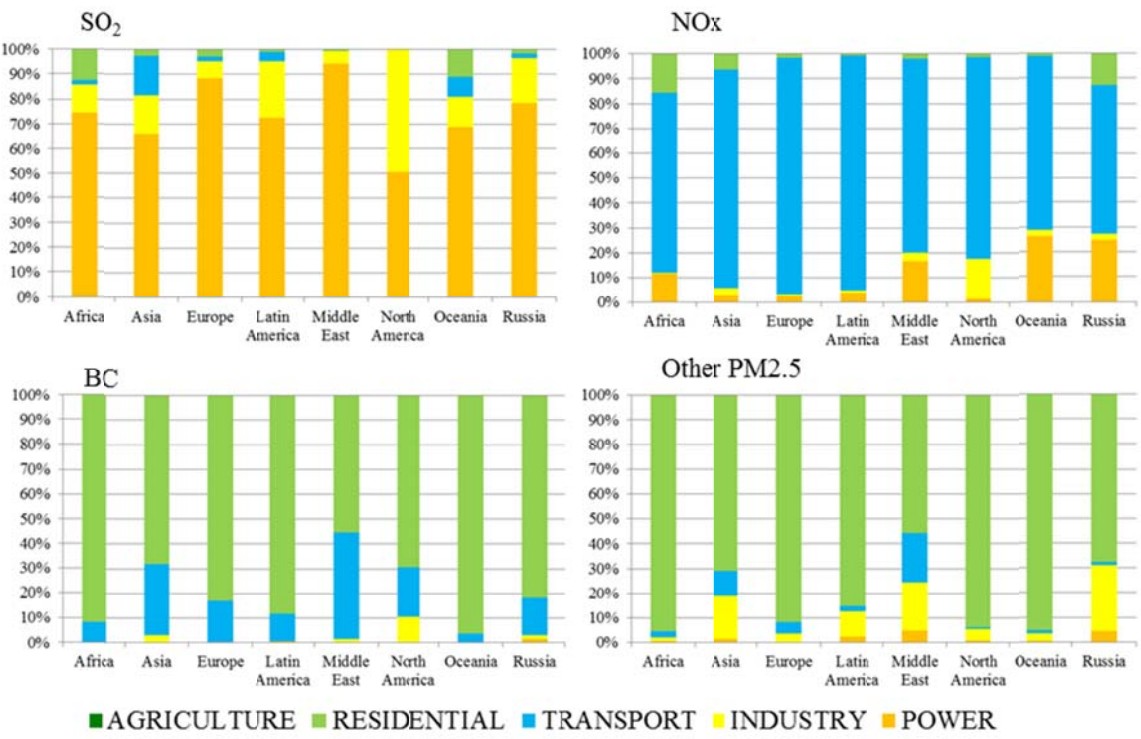

**Figure 6 – Contribution of anthropogenic sectors to the emission uncertainty of various pollutants for different world regions.**

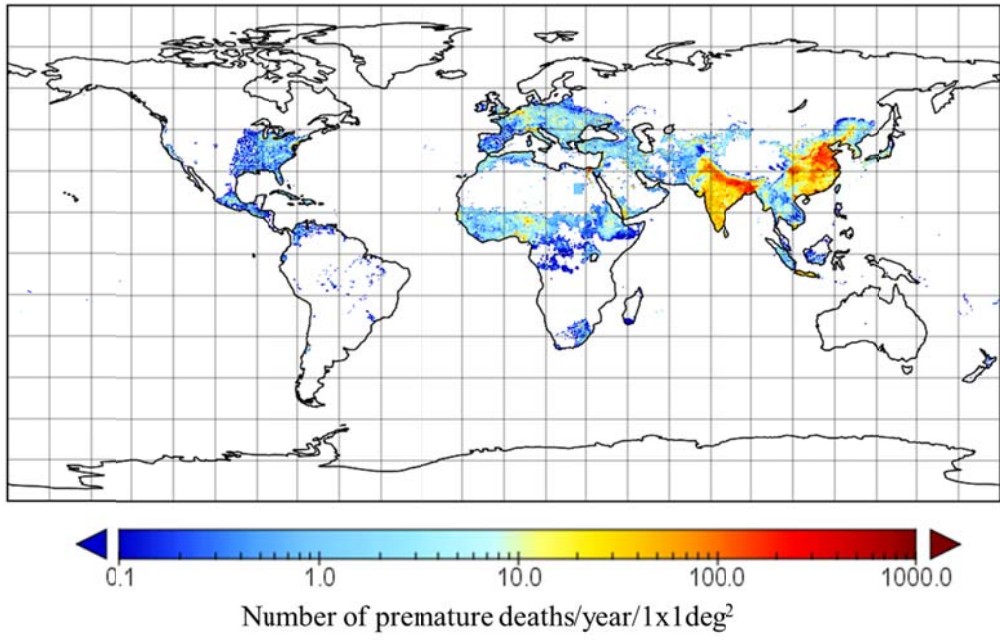

Number of premature deaths/year/1x1deg²

**Figure 7 – Global distribution of premature deaths in 2010 caused by anthropogenic particulate matter pollution estimated using the methodology described in Burnett et al. (2014). A threshold value of 5.8 µg/m3 is assumed and no urban increment adjustments are considered. The relative risk functions of Burnett et al. (2014) are used for the premature death dose-response estimates.**

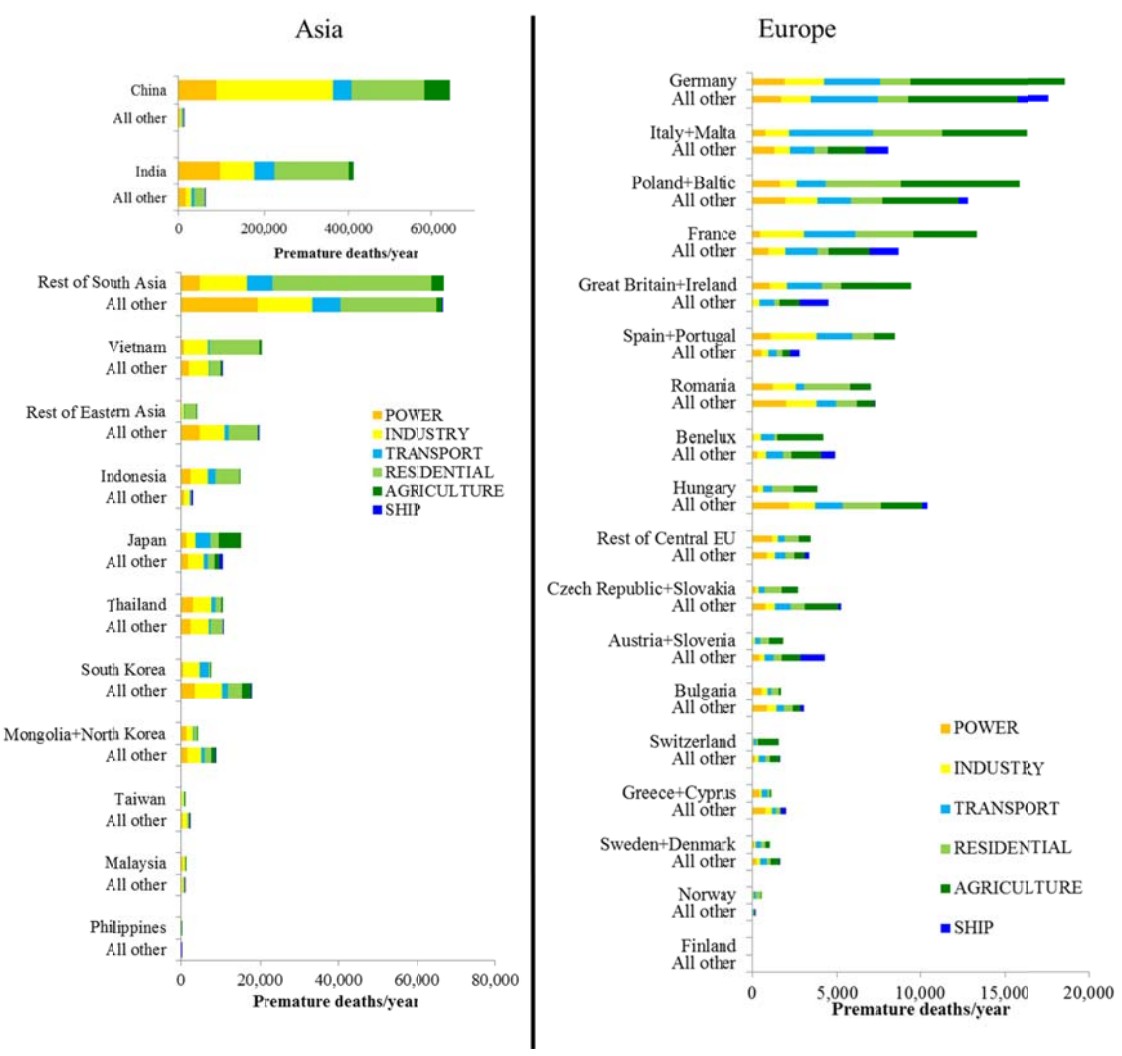

**Figure 8a – Anthropogenic emission sector contributions to premature mortality (deaths/year) due to PM$_{2.5}$ population weighted concentrations in the TM5-FASST receptor regions of Asia (left) and Europe (right). Sector and region contributions pertain to the 'central' emission estimates.**

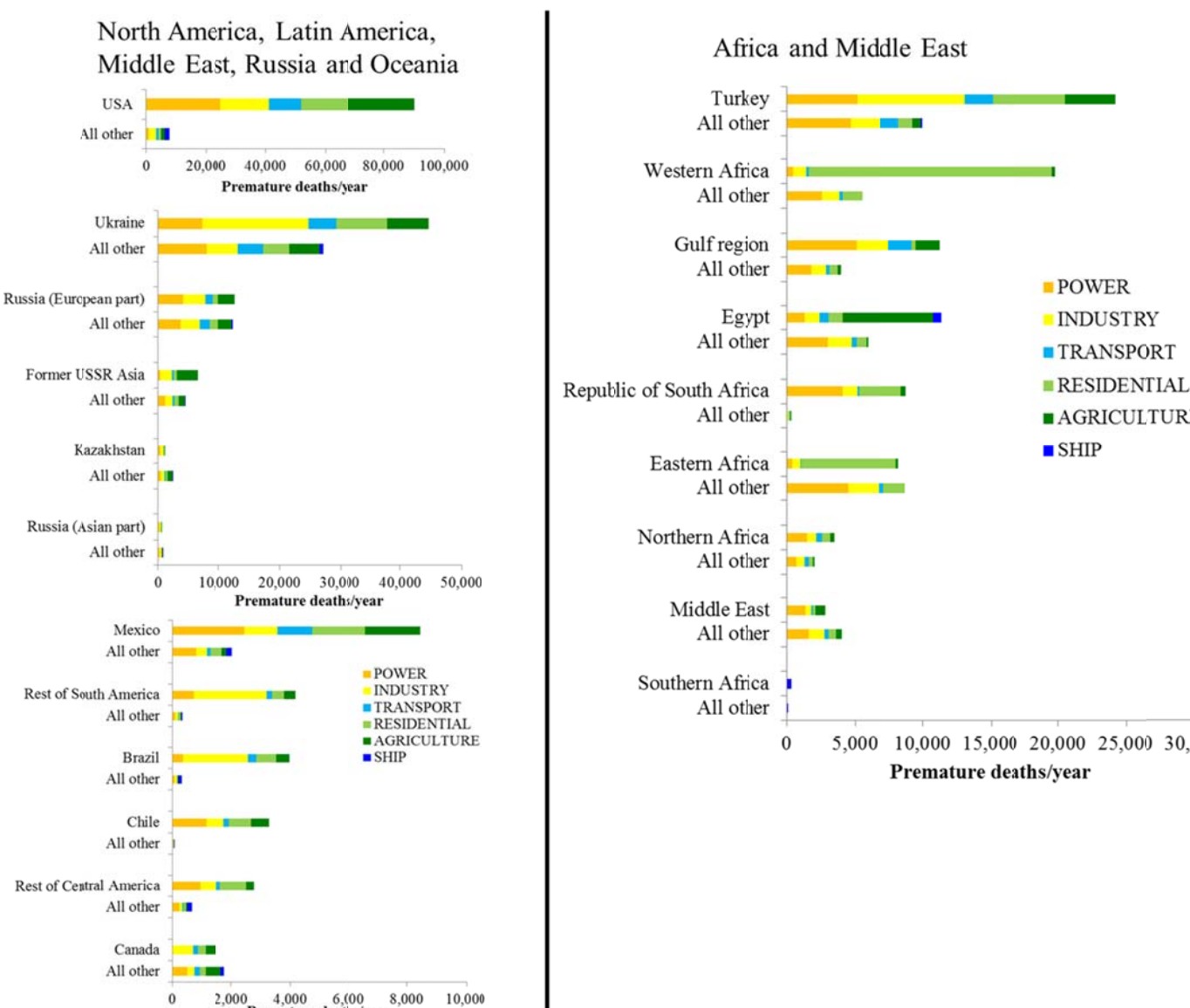

**Figure 8b – Anthropogenic emission sector contributions to premature mortality (deaths/year) due to PM$_{2.5}$ population weighted concentrations in the TM5-FASST receptor regions of North America, Latin America, Russia, Middle East and Oceania (left hand side) and Africa (right hand side). Note that mortality estimates for Argentina+Uruguay, Australia, New Zealand and Pacific Islands are not reported being several orders of magnitude lower than other countries estimates. Sector and region contributions pertain to the 'central' emission estimates.**