# Peer review of "Contribution and uncertainty of sectorial and regional emissions to regional and global"

_Atmospheric Chemistry and Physics, 2017_

## Short Comment (SC1) · 13 Sep 2017

References to the following papers would be relevant:

Brandt, J., J. D. Silver, J. H. Christensen, M. S. Andersen, J. Bønløkke, T. Sigsgaard, C. Geels, A. Gross, A. B. Hansen, K. M. Hansen, G. B. Hedegaard, E. Kaas and L. M. Frohn, 2013. Contribution from the ten major emission sectors in Europe to the Health-Cost Externalities of Air Pollution using the EVA Model System – an integrated modelling approach. Atmospheric Chemistry and Physics, Vol. 13, pp. 7725-7746, 2013. www.atmos-chem-phys.net/13/7725/2013/, doi:10.5194/acp-13-7725-2013.

[Figure]

Brandt, J., J. D. Silver, J. H. Christensen, M. S. Andersen, J. Bønløkke, T. Sigsgaard, C. Geels, A. Gross, A. B. Hansen, K. M. Hansen, G. B. Hedegaard, E. Kaas and L. M. Frohn, 2013. Assessment of Past, Present and Future Health-Cost Externalities of Air Pollution in Europe and the contribution from international ship traffic using the EVA Model System. Atmospheric Chemistry and Physics. Vol. 13, pp. 7747-7764, 2013. www.atmos-chem-phys.net/13/7747/2013/. doi:10.5194/acp-13-7747-2013.

———————————————————

**[ACPD](...)**

---

## Short Comment (SC2) · 4 Oct 2017

This is an interesting study investigating sectoral and regional impacts of anthropogenic emission to regional and global air quality and health. It is a very exciting study, and the topic is very important. I only have a short comment.

There is a previous comprehensive study assessing the sectoral and regional impacts of both anthropogenic and natural emissions to PM2.5 over Central Asia. It would be helpful if the authors include this reference and add some discussions on how your results over this region compare with previous studies.

[Figure]

Reference:

Kulkarni, S., Sobhani, N., Miller-Schulze, J. P., Shafer, M. M., Schauer, J. J., Solomon, P. A., Saide, P. E., Spak, S. N., Cheng, Y. F., Denier van der Gon, H. A. C., Lu, Z., Streets, D. G., Janssens-Maenhout, G., Wiedinmyer, C., Lantz, J., Artamonova, M., Chen, B., Imashev, S., Sverdlik, L., Deminter, J. T., Adhikary, B., D'Allura, A., Wei, C., and Carmichael, G. R.: Source sector and region contributions to BC and PM2.5 in Central Asia, Atmos. Chem. Phys., 15, 1683-1705, https://doi.org/10.5194/acp-15-1683-2015, 2015.

---

## Referee Comment (RC1) · Anonymous Referee #1 · 26 Dec 2017

The manuscript by Crippa et al. investigates the regional and sectoral contributions to PM2.5 and associated health impacts throughout the world. This is accomplished through application of the TM5-FASST response tool. This topic is useful and their results are new, and also appropriate for the scope of this journal. They also provide a much needed estimation of how uncertainty in the emissions estimates propagate into uncertainties in PM2.5 estimates, which is a source of error not often well quantified in health impact studies. That being said, the manuscript good use more attention to previous works, especially in the introduction. These and some additional comments are highlighted below, which include requests for more information about the fidelity of the modeling estimates used here, and the impact of a few assumptions in its applica-

tion that are made but not evaluated either through their own work presented here or references to literature (i.e. assuming PM2.5 responds linearly to emissions changes, or that anthropogenic SOA is negligible). Addressing these concerns constitutes major revisions, after which point this manuscript will be suitable for publication in ACP.

Major:

1.35: I wonder if the authors considered including some more recent estimates e.g. from the Global Burden of Disease project on estimated numbers of premature mortalities from ambient PM2.5 exposure, such as Cohen et al., The Lancet, 2017.

Ok, I see that relevant works be e.g. Lelieveld (2015), Silva (2016) or Cohen (2017) are finally discussed on page 11. Such works however should be discussed as part of the introduction and background information, in order to more clearly articulate the role of the present work.

In general the introduction was laking in some detail with regards to previous works that have considered sector-specific health impacts, the role of model uncertainty vs emissions uncertainties or uncertainties in concentration-response parameterizations on estimates of PM2.5 health impacts.

2.1: Suggest adding references to any number of studies that have estimated the human health impacts of sector-specific policies for PM2.5 reduction.

Equation 1: This equation is an approximation, not an equals sign. This should be clearly indicated, and the error associated with ignoring second-order terms should be discussed, either using evidence from the own authors work or from reference to many previous studies in the literature that have explored the nonlinear response of PM2.5 to emissions perturbations.

Section 2.2: Some essential details of the TM5-FASST model are missing. What is the accuracy of the baseline PM2.5 (total, and speciated) concentrations estimated by TM5 fast compared to in situ measurements in different parts of the world? In

locations where such data is not available, how do the model estimates compare to those from other models, or from remote-sensing derived products? How much error is expected owing to the coarse model resolution when estimating population-weighted concentrations, given the relatively high-resolution variability in population densities?

Ok - while I do see that there is a single paragraph addressing this in very broad strokes, on page 7 (21-28), this evaluation is incommensurate with the scales of the paper. Given the regional, sectoral and species specificity of the source attribution results, the authors need to examine model fidelity on the same scales.

11.31: Cohen et al. (2017) also report a range for the total estimated global premature deaths from ambient PM2.5 - which should be repeated here. This is interesting to consider, as the source of the uncertainty in the Cohen paper is from uncertainty in the concentration-response relationships (IERs), not from uncertainties in the exposure estimates that may be owing to uncertainties (in part) from emissions. However, the range of values cited here (+/- 1.1 million) indicates that this uncertainty associated with emissions estimates is an factor, which hasn't been much considered previously. This is an import results of the present work which I believe could be highlighted more (i.e. by comparing the magnitude of the emissions-driven uncertainties to the magnitude of other types of uncertainties considered in different studies). Quantitative summary of this (similar to the final sentence of the manuscript) would be nice to see in the abstract as well.

Minor:

2.10-2.14: What fraction of secondary PM2.5 long-range transport is owing to transport of the gas-phase precursors vs the transport of the secondarily formed PM2.5 itself?

2.27: Clarify here that this inventory, and the prescribed emissions for these experiments, pertain only to anthropogenic emissions.

3.14: Can the authors comment on the validity of this assumption, as backed up by

their own investigations or those in previous studies in the literature?

3.23: The source-receptor modeling was based around a single year that didn't alight with the year of the emissions considered. To what extent does this misalignment potentially impact results? Or to what extent is the meteorology in this particular year representative of a climatological average? I guess I'm just wondering if the authors have checked if 2001 was for any reason particularly extreme with regards to temperature, precipitation, transport, or sources of natural PM2.5 such as biomass burning?

3.27: To what extent does not including anthropogenic SOA influence conclusions about the role of different sectors?

4.4: It seems that rather than aggregation the authors could consider some metrics that are normalized with regards to the country size or population.

4.20: Here and elsewhere the Janssens-Maenhout (2017, submitted) paper is cited, although it's hard to evaluate what information is contained therein about

Figure 1: It's not clear – are the % contributions to the average PM2.5 in each region, or to the population-weighted average PM2.5 in each region?

7.34: I think the impacts of the residential sector on indoor air quality are well known and have been documented in many previous studies, that could be cited.

7.39: Similarly, the role of the agricultural sector or NH3 in particularly has been noted in several previous and recent studies. The authors continue to cite only Maas and Grennfelt, 2016, despite the broader literature available for comparison.

8.9-11: Can the authors explain why primary emissions play such a large role in the uncertainty analysis, compared to their contribution to absolute PM2.5 concentration?

9.20: Given that this work doesn't include anthropogenic SOA, what is the role of NMVOCs in PM2.5 formation? I guess I was just surprised to see these mentioned here.

11.34: What is the "urban increment subgrid adjustment"?

11.33 - 35: I strongly agree that these factors are critical towards making these comparisons, as are sources of information such as populations densities and baseline mortality rates. For those precise reasons, the authors should provide details on these aspects as used in their study, as have been provided in the cited works, in order to make such comparisons possible and meaningful.

12.10-12: What it is about these regions that given them such relatively large era-regional contributions to PM2.5 health impacts?

Editorial:

2.23: "not to the least" change to "not the least"

2.35: "at sector" change to "at the sector"

2.36: "on the potential" change to "of the potential"

3.19: Some of this sentence seems to be missing.

4.16: "as following" change to "as follows"

6.16: "across" change to "an across"

8.23: "Europe the" change to "Europe, the"

---

## Referee Comment (RC2) · Anonymous Referee #2 · 10 Jan 2018

The authors coupled the HTAPv2.2 global air pollutant emission inventory with the global source receptor model TM5-FASST to evaluate the relative contribution of the major anthropogenic emission sources to air quality and health in 2010. They focused on $PM_{2.5}$ due to its negative impacts on human health. The objective of this paper is to evaluate the emissions uncertainties at sector and regional levels, and their propagation in modeled $PM_{2.5}$ concentrations and associated impacts on health. Although the authors state that they have two objectives, I do not quite understand the difference between the two. I find that what the paper is trying to do is important but there are some major problems that need to be addressed before this can be published in ACP.

First, if the objective is to understand the health impacts of $PM_{2.5}$, I believe that the authors need to make sure that their model simulations match with the observations. I do not find the existing comparison in the paper (p. 7, l. 21-28) very convincing. The authors could have at least compared with the recent WHO database of annual $PM_{2.5}$ concentrations at various cities (http://www.who.int/phe/health_topics/outdoorair/databases/cities/en/). For the US, there is much better database that could be used (https://www.epa.gov/outdoor-air-quality-data/interactive-map-air-quality-monitors). The authors seem to allude that it is ok to not include the natural emissions but I disagree and think that the natural emissions need to be included in the model.

Second, I find that the emissions uncertainty estimate seems a little simplistic to only assess within the HTAP inventory, considering the existing differences among various inventories. Also, if the RCP emissions for the year 2000 are used as a baseline, to me it makes more sense to use RCP 2010 in their analysis, rather than switching to HTAP v2.2. Or if the HTAP is to be used, the uncertainty analysis should include the differences in estimates between RCP 2010 and HTAP v2.2. Also, it might be a good idea to compare with some other estimates in existing studies that have estimated emissions uncertainties for certain countries.

Third, I also find it problematic that important details and assumptions of TM5-FASST methodology are described in the paper that is still under preparation. I am assuming that the ΔPMref and ΔEref in Eq. 1 refer to the difference between the TM5-FASST simulation results for $PM_{2.5}$ (and also $PM_{10}$ as well?) using the RCP baseline and the perturbation (-20%) and the emissions themselves, respectively. However, I find it troublesome that these stay constant when the emissions change for all regions and sectors. We know that $PM_{2.5}$ formation is a non-linear process and I do not believe it would work in a linear form for every region for every sector. If it does, maybe that is because simulation uses too coarse of a resolution and the result does not seem realistic. Also, it seems problematic that no explicit treatment of anthropogenic SOA is considered.

Is it correct that TM5-FASST simulations were run for each sector separately and also for all the sectors combined? That is how it looks like from Figure 4. If so, can the authors confirm that the sum of concentrations from each of the sectors run separately are similar to the values when the simulation was done including all the sector emissions together? It would be a nice test to check the linearity in the model. If the simulations were done in this way, then what was the reason equation 1 had to be used? The authors could have easily calculated the impact of each sector using these simulations instead?

I have a hard time understanding the sentence on p. 9 l. 5-8. How do the authors determine the relative contribution to total emission inventory uncertainty? Are the authors using the uncertainty

for a specific sector over the total uncertainty for a specific pollutant as the "average sector relative contribution to total emission inventory"? If so, this does not necessarily take the magnitude of emissions into account and so maybe just looking at this value and deciding which sector to focus on might be a little too simplistic?

Are the upper and the lower boundaries of $PM_{2.5}$ concentrations (Table 2 and Figure 5) calculated based on the linear relationship between emissions per region? In other words, are they simply calculated from emissions, rather than running the simulation again in a chemical transport model?

Minor comments:
1. I would like to see a figure that shows the 10 aggregated receptor regions, as it is unclear, for example, what China+ region includes. Does it just include Mongolia? Or also Korea and Japan?
2. Why are some European countries lumped together in Figure 2 (Austria and Slovenia, for example), whereas others are not?
3. Why are there more countries in Figure 3 than in Figure 1?

p. 2. l. 30-34: The sentence is too long and difficult to understand. Please rephrase the sentence.

p. 2 l. 36-37: The authors state that a second objective of the analysis is to "inform local, regional hemispheric air quality policy makers on the potential impacts of less known emission sectors or regions" but they are focusing on the "6 major anthropogenic emission sectors (l. 6-7, p. 3)." What do they mean by "less known emission sectors" then?

p. 3. l. 19-20. This sentence is not finished.

p. 3. L. 22. Why was such a coarse resolution used, when HTAPv2.2 is much finer?

p. 3 l. 30 relativey → relatively

p. 7 l. 37-39 Perhaps a reference to Bauer et al. (2016) would be appropriate here.

p. 11 l. 36. It is unclear to me where this value (7% for the global non accidental mortalities) is coming from. Can you clarify or cite the source?

p. 12 l. 10 such the Gulf → such as the Gulf

p. 19 Table 2. How do you quantify the uncertainty for a certain pollutant for a region?

Reference:
Bauer, S.E., K. Tsigaridis, and R.L. Miller, 2016: Significant atmospheric aerosol pollution caused by world food cultivation. *Geophys. Res. Lett.*, 43, no. 10, 5394-5400, doi:10.1002/2016GL068354.

---

## Author Comment (AC1) · 16 Feb 2018

We acknowledge the comment posted by J. B. Brandt and we will take into consideration his suggestions as following:

- We cited the paper by Brandt et al. 2013a in the introduction: The total number of premature deaths in the whole model domain is 680000 while we estimate 210000. As Brandt et al. report in their work, their premature deaths estimates are twice as high as the numbers reported for the CAFE study due to the different domain considered for Europe. For this reason a consistent comparison of estimated premature deaths is not feasible.

- At page 12, line 21 we will also add the following sentence: "The premature deaths induced by international shipping emissions represent 5.5% of total EU PD, in the range the results of Brandt et al. (2013a)."

References Brandt, J., J. D. Silver, J. H. Christensen, M. S. Andersen, J. Bønløkke, T. Sigsgaard, C. Geels, A. Gross, A. B. Hansen, K. M. Hansen, G. B. Hedegaard, E. Kaas and L. M. Frohn, 2013b. Assessment of Past, Present and Future Health-Cost Externalities of Air Pollution in Europe and the contribution from international ship traffic using the EVA Model System. Atmospheric Chemistry and Physics. Vol. 13, pp. 7747-7764, 2013a.

---

## Author Comment (AC2) · 16 Feb 2018

We acknowledge the comment posted by N. Sobhani however the type of analysis of his study is a bit different from the purpose of our work. Kulkarni, S. et al. mainly look at the intercontinental transport of pollution from Europe and Russia to Central Asia and try to identify the sources of pollution. In addition they focus on large scale biomass burning emissions while this source is not considered in our study. Similarly to Kulkarni, S. et al. we identify the residential, industry and partly the transport sector as the most important ones contributing to PM formation in Asia, together with the power sector for the secondary inorganic fraction of PM. Although we recognize the relevance

of the work of Kulkarni, S. et al., we think the citation of their study is not fully pertinent with the purpose of our work.

References Kulkarni, S., Sobhani, N., Miller-Schulze, J. P., Shafer, M. M., Schauer, J. J., Solomon, P. A., Saide, P. E., Spak, S. N., Cheng, Y. F., Denier van der Gon, H. A. C., Lu, Z., Streets, D. G., Janssens-Maenhout, G., Wiedinmyer, C., Lantz, J., Artamonova, M., Chen, B., Imashev, S., Sverdlik, L., Deminter, J. T., Adhikary, B., D'Allura, A., Wei, C., and Carmichael, G. R.: Source sector and region contributions to BC and PM2.5 in Central Asia, Atmos. Chem. Phys., 15, 1683-1705, https://doi.org/10.5194/acp-15-683-2015, 2015.

---

## Author Comment (AC3) · 16 Feb 2018

The authors are grateful to Referee#1 for the helpful comments that helped improve the manuscript. Due to the strict link between this publication and the work recently submitted by van Dingenen et al. (submitted, 2018) about the TM5-FASST methodology, we offered the possibility to the Editor and the Reviewer to access the work of van Dingenen et al. (submitted, 2018) although not yet published in ACPD. Thanks to the Reviewer's comments, we also realized that some methodological aspects of the TM5-FASST tool could have been further developed also in the publication of van Dingenen et al. (submitted, 2018). Therefore, discussions on the comparison between PM2.5 modeled concentrations vs. the measured ones, as well as further details about the extension of the "perturbation approach" to the attribution of sectors and sources will be included in the review phase of the paper by van Dingenen et al. (submitted, 2018). We feel that we have been able to address all concerns, as outlined below.

**Anonymous Referee #1**

The manuscript by Crippa et al. investigates the regional and sectoral contributions to PM2.5 and associated health impacts throughout the world. This is accomplished through application of the TM5-FASST response tool. This topic is useful and their results are new, and also appropriate for the scope of this journal. They also provide a much needed estimation of how uncertainty in the emissions estimates propagate into uncertainties in PM2.5 estimates, which is a source of error not often well quantified in health impact studies. That being said, the manuscript good use more attention to previous works, especially in the introduction. These and some additional comments are highlighted below, which include requests for more information about the fidelity of the modeling estimates used here, and the impact of a few assumptions in its application that are made but not evaluated either through their own work presented here or references to literature (i.e. assuming PM2.5 responds linearly to emissions changes, or that anthropogenic SOA is negligible). Addressing these concerns constitutes major revisions, after which point this manuscript will be suitable for publication in ACP.

Major:

**1.35:** I wonder if the authors considered including some more recent estimates e.g. from the Global Burden of Disease project on estimated numbers of premature mortalities from ambient PM2.5 exposure, such as Cohen et al., The Lancet, 2017.

Ok, I see that relevant works be e.g. Lelieveld (2015), Silva (2016) or Cohen (2017) are finally discussed on page 11. Such works however should be discussed as part of the introduction and background information, in order to more clearly articulate the role of the present work.
In general the introduction was lacking in some detail with regards to previous works that have considered sector-specific health impacts, the role of model uncertainty vs emissions uncertainties or uncertainties in concentration-response parameterizations on estimates of PM2.5 health impacts.

As suggested by the Reviewer, the following sentences have been added to the introduction:

"Exposure to and impact from aerosols on humans can be estimated by a variety of approaches, ranging from epidemiological studies to pure modelling approaches. The Burnett et al. (2014)

risk-response methodology is often used in models to estimate premature deaths/mortality (PD) due to air pollution exposure, e.g. in Lelieveld et al. (2015) and Silva et al. (2016), who report a global mortality in 2010 due to air quality issues induced by anthropogenic emissions of 2.5 and 2.2 million people, respectively. A higher global mortality is found in a more recent work by Cohen et al. (2017) accounting for 3.9 million premature deaths/year due to different model assumptions. In Europe, Brant et al. (2013) estimate 680 thousand premature deaths, which is twice as high as the numbers reported for the CAFE (Clean Air for Europe) study (Watkiss et al., 2005). Recently, using the same emission database as in this study, Im et al. (2017) report a multi-model mean estimate of PD of 414.000 (range 230-570 thousand) for Europe and 160 thousand PDs for the USA. At the global scale, models, in some cases using satellite information (Brauer et al., 2015;Van Donkelaar et al., 2016), are the most practical source of information of exposure to air pollution. However, model calculations are subject to a range of uncertainties related with incomplete understanding of transport, chemical transformation, removal processes, and not the least, emission information."

**2.1:** Suggest adding references to any number of studies that have estimated the human health impacts of sector-specific policies for PM2.5 reduction.

As suggested by the reviewer we added in the manuscript the following some references related with studies on human health impacts of sector-specific PM2.5 contributions:

"These policies are usually implemented under national legislation (Henneman et al., 2017; Morgan, 2012), while in Europe transboundary air pollution is also addressed by the regional protocol under the UNECE Convention on Long-Range Transport of Air Pollution (CLRTAP). At city/local level, several studies have been developed to assess the contribution of sector specific emissions to PM2.5 concentrations with the aim of designing air quality plans at local and regional level (Karagulian et al., 2015; Thunis et al., 2016)."

Equation 1: This equation is an approximation, not an equals sign. This should be clearly indicated, and the error associated with ignoring second-order terms should be discussed, either using evidence from the own authors work or from reference to many previous studies in the literature that have explored the nonlinear response of PM2.5 to emissions perturbations.

Equation 1 represents how PM concentrations can be estimated using the 20% perturbation which is the basis of the TM5-FASST methodology. So the equal sign is correct, although this equation represents an approximation due to errors both of the chemistry and transport modeling and to the emissions. We refer the Reviewer to the paper by van Dingenen et al (submitted, 2018) for details about the errors due the chemistry and transport, while in this work we address mainly the errors due to emissions. Below additional details about the TM5-FASST methodology:

The reduced-form model TM5-FASST is computing the concentration resulting from an arbitrary emission scenario $E_s$ using a perturbation approach, i.e. the difference between $E_s$ and $E_{ref}$ ($dE_s$) is considered as a perturbation on $E_{ref}$ and the resulting concentration is evaluated as a perturbation $dPM$ on the reference concentration, hence:

$$PM(E_s) = PM(E_{ref} + dE_s) = PM_{ref} + dPM = PM_{ref} + SRC \cdot dE_s \qquad \text{(a)}$$

Where $dE_s = E_s - E_{ref}$ and $E_{ref}$ is the RCP reference scenario from which the *SRC* have been computed.

The contribution of a single sector $j$ is calculated as the difference between the concentration including all sectors, and the concentration from the emissions excluding the single sector $j$

$$PM\,(E_{s,j}) = PM(E_s) - PM(E_s - E_{s,j}) = SRC \cdot [dE_s - d(E_s - E_{s,j})] = SRC \cdot E_{s,j}$$

If the linearity holds, the sum of $PM(E_{s,j})$ over all sectors j should be equal to $PM(E_s)$, or:

$$\sum_j PM(E_{s,j}) = PM_{ref} + SRC \cdot \left(E_s - E_{ref}\right)$$

The TM5-FASST runs were performed for different scenarios, comparing the reference HTAP_v2.2 emissions with a scenario where emissions from one single sector were subtracted from the total emissions. Then comparing the reference case and each scenario (REF-sector$_i$), the contribution of each sector to PM2.5 concentrations is estimated. This approach is based on the assumption that the individual sector contributions add up linearly to total PM2.5, as mentioned in the paper. The paper by Van Dingenen et al. describing the whole TM5-FASST methodology has just been submitted to ACP (van Dingenen et al., submitted, 2018) Equation 1 represents the basis of the TM5-FASST method, since it describes how a variation in the emissions (delta emissions) determines a delta in PM2.5 based on the source receptor relationships.

The following discussion on how to apply the "perturbation approach" on the sector and source attribution will be also included in the paper by van Dingenen et al. (submitted, 2018):

Equation (2) expresses the 'perturbation' approach applied in the linearized TM5-FASST model, i.e. an arbitrary emission scenario is evaluated as a deviation from the base emission scenario, and the resulting pollutant concentration is obtained as the sum of the base concentration and a delta term, the latter proportional to the emission deviation from the base case (Figure 1).

A particular application of TM5-FASST is the attribution of the (anthropogenic) pollutant concentration to individual source regions or sectors. Due to the fixed contribution of the base concentration which does not contain information on the originating sources, Eq. (2) is not immediately suitable for such an analysis. Instead, we calculate for each individual source the contributing part by first evaluating all sources together ('total' simulation'), and subsequently subtracting the individual source emissions ($E_s$) from the total, evaluating the resulting pollutant concentration ($C_{minus\_s}$), and making the difference with the 'total simulation' to obtain the single source contribution ($C_s$).

$$C_{j,tot}(y) = C_{j,base}(y) + \sum_{n_x} \sum_{n_i} A_{ij}[x,y] \cdot \left[E_{i,tot}(x) - E_{i,base}(x)\right] \qquad \text{(2)}$$

$$C_{j,minus\_s}(y) = C_{j,base}(y) + \sum_{n_x} \sum_{n_i} A_{ij}[x,y] \cdot \left[E_{i,tot}(x) - E_{i,s}(x) - E_{i,base}(x)\right] \qquad (4)$$

$$C_{j,s}^*(y) = C_{j,tot}(y) - C_{j,minus\_s}(y) = \sum_{n_x} \sum_{n_i} A_{ij}[x,y] \cdot E_{i,s}(x) \qquad (5)$$

We can now reconstruct $C_{j,tot}^*$ as the sum of the individual source contributions:

$$C_{j,tot}^*(y) = \sum_{n_s} C_{j,s}^*(y) \qquad (6)$$

$C_{j,tot}^*(y)$ is equivalent to $C_{j,tot}(y)$ in Eq. 2 only if

$C_{j,base}(y) = \sum_{n_x} \sum_{n_i} A_{ij}[x,y] \cdot E_{i,base}(x)$, in other words if the emission-concentration relation is perfectly linear and passes through the origin.

In reality there is some degree of non-linearity in most emission-pollutant relation as illustrated in Figs. 3 and 6. Figure A shows for each of the FASST regions the total $PM_{2.5}$ concentration obtained by Eq. 6 versus the TM5 base simulation result, illustrating the non-linearity error resulting from the application of Eq. 6. For 43 out of 56 regions, the deviation from the base simulation is less than 30%, only 3 regions (former Soviet Union, New Zealand and Pacific) deviate more than 50% from the TM5 model result. Consistency with the 'perturbation approach' is restored by simply rescaling the individual source contributions:

$$C_{j,s}(y) = \frac{C_{j,tot}(y)}{C_{j,tot}^*(y)} \; C_{s,j}^*(y) \qquad (7)$$

This approach is valid for evaluating the attribution by sector as well as by source region.

[Figure]

Figure A: Scatter plot of regionally averaged PM2.5 concentration (including all anthropogenic components) obtained as the sum of individual source region contribution by linear scaling of

their respective emissions with TM5-FASST source-receptor coefficients (Eq. 6), versus the regional average obtained by the full TM5 model.

**Section 2.2:** Some essential details of the TM5-FASST model are missing. What is the accuracy of the baseline PM2.5 (total, and speciated) concentrations estimated by TM5 fast compared to in situ measurements in different parts of the world? In locations where such data is not available, how do the model estimates compare to those from other models, or from remote-sensing derived products? How much error is expected owing to the coarse model resolution when estimating population-weighted concentrations, given the relatively high-resolution variability in population densities?

In the work by van Dingenen et al. (submitted on the 31$^{st}$ of January 2018 to the ACP HTAP special issue) details about the comparison between the linearized TM5-FASST model and the full TM5 runs are provided in Section 3.1 "Validation against the full TM5 model: additivity and linearity". They also report the linearity and additivity issues for PM2.5 and its speciation in Figure 3 of their paper, as shown below.

[Figure]

[Figure]

*Figure 1. Additivity and linearity test of perturbations using TM5 outcome for regional population-weighted mean secondary inorganic PM2.5 concentrations for 3 perturbation magnitudes (green: +100%, red: -20%, blue: -80% relative to base simulation emissions). X-axis: simultaneous perturbation of SO2 and NOx emissions Left column Y-axis: sum of TM5 concentration response to two individual SO2 and NOx perturbations. Right column Y-axis: sum of linearly extrapolated individual 20% perturbations (FASST approach). Each point corresponds to the population-weighted mean concentrations over a receptor region (same regions as in Fig. 2).*

Ok - while I do see that there is a single paragraph addressing this in very broad strokes, on page 7 (21-28), this evaluation is incommensurate with the scales of the paper. Given the regional, sectoral and species specificity of the source attribution results, the authors need to examine model fidelity on the same scales.

In addition, van Dingenen et al. (submitted, 2018) report in Fig.7 of their publication the comparison of modeled PM2.5 concentrations between the full TM5 runs and the TM5-FASST ones, as shown below.

[Figure]

*Figure 2. (a) PM2.5 concentration obtained with TM5-FASST versus TM5-CTM for high (FLE, red dots) and low (MIT, green dots) emission scenarios (see text). Each point represents the population-weighted mean over a TM5-FASST receptor region. Black line: 1:1 relation. breakdown for (b) primary (BC+POM+other primary PM2.5) and (c) secondary (SO4+NO3+NH4) PM components (same axis definitions as left plot).*

In addition, van Dingenen et al. validated their modeled PM2.5 concentrations against in situ measurements and satellite derived data, as discussed in the following.

Figure 3 shows the comparison between the PM2.5 concentrations modeled by TM5-FASST and the measured ones reported in the WHO database for different world regions (i.e. EUR=Europe, NAM=North America, China, S-ASIA=Southern Asia, LAM=Latin America, AFR=Africa). This includes measurement points as well as PM2.5 concentration estimates based on a fraction of PM10 measurements (e.g. almost all points for the comparison in China are based on this second method). Quite good agreement is observed for Europe, North America and partly China where measurements have been performed over longer time compared to developing countries and they are based on quite consolidated methods. The comparison for Latin America and Africa is much less robust and the scatter possibly highlights a non-optimal modeling of large scale biomass burning for the TM5-FASST model. Figure 4 reports the comparison of WHO regional average of urban stations against the FASST population weighted average of grid cells. Similarly to the findings of Figure 1, the comparison for industrialized countries is very good, while for other developing regions the agreement is less satisfactory both due to less accurate measurements (e.g. reported by WHO) and lower quality modeling of specific sources by TM5-FASST (e.g. large scale biomass burning).

TM5-FASST modeled PM2.5 concentrations have been also validated against satellite products (see Figure 5) which are based on aerosol optical depth measurements together with chemical transport model information to retrieve from the total column the information of PM

concentrations in the lowest layer of the atmosphere (Boys et al., 2014; van Donkelaar et al., 2010, 2014).

[Figure]

Figure 3 - TM5-FASST grid-cell mean (with urban increment parameterisation) versus individual monitoring stations (WHO consolidated database, including both measured and estimated PM2.5).

[Figure]

Figure 4 - WHO regional average of urban stations (+/- 1 stdev) and FASST population weighted average of grid cells.

| EUROPE | USA |
|---|---|
| CHINA | INDIA |
| BRASIL | |

[Figure]

Figure 5 - Comparison between TM5-FASST and satellite products for world regions (Boys et al., 2014; van Donkelaar et al., 2010, 2014).

While for the full details and discussion with refer to Van Dingenen et al. (2018), we summarize their results in our manuscript as following:

In section 2.1 we added the following sentence:

"The TM5-FASST model is extensively documented in a companion publication in this special issue. Van Dingenen et al., (2018) provide an extensive evaluation of the model, model assumptions and performance with regard to linearity and additivity of concentration response to different size emission perturbations and future emission scenarios. The validation of TM5-FASST against the full TM5 model runs is extensively discussed by van Dingenen et al. (2018), as well as the validity of the assumptions of linearity and additivity behind this reduced form-model. Below we summarize the most important features of relevance for this work, and refer for more detail to Van Dingenen et al., (2018)."

We added in the manuscript the following discussion in section 3.3:

"The TM5-FASST model developed by van Dingenen et al. (2018) has been validated against concentration estimates derived from the WHO database and satellite-based measurements (van Donkelaar et al., 2010, 2014). General good agreement is found between the PM2.5 concentrations modeled by TM5-FASST and the measured ones reported in the WHO database for Europe (within 20% deviation), North America (within 5% deviation) and partly China due to the higher accuracy of the measurements. The comparison for Latin America and Africa is much less robust (40-60% deviation from the 1:1 line) and the scatter possibly highlights a non-optimal modeling of specific sources relevant for these regions by TM5-FASST (e.g. large scale biomass burning) by the TM5-FASST model. Similar results are also found comparing regional averages of urban stations from WHO against the FASST population weighted average of grid cells. The TM5-FASST modeled PM2.5 concentrations have been compared to satellite products which are based on aerosol optical depth measurements together with chemical transport model information to retrieve from the total column the information of PM concentrations in the lowest layer of the atmosphere (Boys et al., 2014; van Donkelaar et al., 2010, 2014). The regional comparison shows consistent results with the ground based measurements comparison (e.g. good

agreement for EU and USA within 10% deviation, while lower agreement for developing and emerging countries)."

**11.31:** Cohen et al. (2017) also report a range for the total estimated global premature deaths from ambient PM2.5 - which should be repeated here. This is interesting to consider, as the source of the uncertainty in the Cohen paper is from uncertainty in the concentration-response relationships (IERs), not from uncertainties in the exposure estimates that may be owing to uncertainties (in part) from emissions. However, the range of values cited here (+/- 1.1 million) indicates that this uncertainty associated with emissions estimates is a factor, which hasn't been much considered previously. This is an import results of the present work which I believe could be highlighted more (i.e. by comparing the magnitude of the emissions-driven uncertainties to the magnitude of other types of uncertainties considered in different studies). Quantitative summary of this (similar to the final sentence of the manuscript) would be nice to see in the abstract as well.

The following sentences have been added:
"In our work we only evaluate how the uncertainty of emission inventories influences the health impact estimates focusing on the interregional aspects (we do not evaluate effects of misallocation of sources within regions) and not all the other sources of uncertainties often included in literature studies, such as the uncertainty of concentration-response estimates, of air quality models used to estimate particulate matter concentrations, etc. An overview of the propagation of the uncertainty associated with an ensemble of air quality models to health and crop impacts is provided by Solazzo et al. (2018, submitted). Solazzo et al. find in their analysis over the European countries a mean number of PDs due to exposure to PM2.5 and ozone of approximately 370 thousands (inter-quantile range between 260 and 415 thousand). Moreover, they estimate that a reduction in the uncertainty of the modelled ozone by 61% - 80% (depending on the aggregation metric used) and by 46% for PM2.5, produces a reduction in the uncertainty in premature mortality and crop loss of more than 60%. However, we show that the often neglected emission inventories' uncertainty provides a range of premature deaths of ±1.1 million at the global scale, which is in the same order of magnitude of the uncertainty of air quality models and concentration-response functions (Cohen et al., 2017)."

Minor:

**2.10-2.14:** What fraction of secondary PM2.5 long-range transport is owing to transport of the gas-phase precursors vs the transport of the secondarily formed PM2.5 itself?

To answer this question, which was not explicitly studied in this publication, but included in the model calculations, one has to consider 4 aspects: chemical lifetime of the precursor gases, atmospheric transport, transport distance, and removal processes of both precursors and aerosols. Lifetimes of precursor gases range from hours (NH3), hours-to-days (NOx) and several days (SO2). A back-of-the-envelope calculation assuming a lifetime 0.1 hour and a wind speed of 1 m/s, would indicate a transport distance of ca 8 km, and clearly most of the precursor would be oxidized before leaving the ca. 100x100 km TM5 gridbox. On the other hand a lifetime of 7 days

and a wind speed of 10 m/s would imply that this precursor could travel thousands of km before 2/3 of it would be oxidized.

We propose to include the following phrase:

"Although primary PM$_{2.5}$ (particulate matter with a diameter less than 2.5 µm) and intermediately lived (days-to-weeks) precursor gases can travel over long distances, the transboundary components of anthropogenic PM are mainly associated with secondary aerosols which are formed in the atmosphere through complex chemical reactions and gas-to-aerosol transformation, transport and removal processes, of gaseous precursors transported out of source regions (Maas and Grennfelt, 2016)."

**2.27:** Clarify here that this inventory, and the prescribed emissions for these experiments, pertain only to anthropogenic emissions.
Done

**3.14:** Can the authors comment on the validity of this assumption, as backed up by their own investigations or those in previous studies in the literature?

We assume that individual sector contributions add up linearly to total PM2.5. The figure below shows the very good agreement between total PM2.5 concentrations and the sum of sector-specific concentrations for each receptor region. Additional details can be found in van Dingenen et al. (submitted, 2018).

[Figure]

Figure 1 – Comparison between the total modeled PM2.5 concentration and the sum of the sectors.

**3.23:** The source-receptor modeling was based around a single year that didn't alight with the year of the emissions considered. To what extent does this misalignment potentially impact results? Or to what extent is the meteorology in this particular year representative of a climatological average? I guess I'm just wondering if the authors have checked if 2001 was for any reason particularly extreme with regards to temperature, precipitation, transport, or sources of natural PM2.5 such as biomass burning?

Anthropogenic emissions in general do not greatly vary from year to year and a large co-variation with specific meteorological conditions is considered not very important. Indeed such co-variation can be an important issue for natural emission. Biomass burning, sea salt and mineral dust are dependent among other factors on meteorological conditions. For the natural emissions of dust, sea salt and biomass burning we included the recommended gridded datasets made for AEROCOM phase 1 for the year 2000- indeed not aligning with the meteorological year 2001 used in the TM5 CTM in this study. There are three considerations of relevance for this paper. If the goal is to have the most accurate estimate of natural emissions, the use of a community endorsed dataset is probably a safe one, since model generated emissions would carry their own uncertainties. While, especially for mineral dust and biomass burning, there are large inter-annual variations, these variations- at least at larger scales- are probably smaller than the emission uncertainties themselves. And finally, the use of 'constant' emission, allows factoring out their uncertainties, since the scope of the work is considering mostly anthropogenic emissions.

**3.27:** To what extent does not including anthropogenic SOA influence conclusions about the role of different sectors?

Unfortunately we do not have estimates of the contribution of anthropogenic SOA, as the gas phase chemical degradation scheme didn't include emissions of the relevant precursor gases. The importance of anthropogenic SOA ranges regionally widely, as demonstrated by a recent study by Farina et al. (2010) indicating a global source of 1.6 Tg, or ca. 5.5 % of the overall SOA formation. The relatively importance, however, may dependent regionally, and is deemed higher in regions with less VOC emission controls. Overall, we feel that the uncertainty stemming from our knowledge in SOA formation is higher than the omission of anthropogenic SOA. We would also like to mention that the development of the volatility-based SOA formation approach, means that the boundaries between 'primary' and 'secondary' SOA are disappearing, making it difficult to attribute organic aerosol to either primary, secondary (or natural-anthropogenic), as they strongly interact. Nevertheless, we speculate that the inclusion of SOA would possibly lead to a higher role of the transboundary pollution mainly for those sectors emitting PM and VOCs (e.g. residential, and to some extent transport and industry).

Therefore we added the following sentences to the manuscript:

"The importance of anthropogenic SOA ranges regionally widely, as demonstrated by a recent study by Farina et al. (2010) indicating a global source of 1.6 Tg, or ca. 5.5 % of the overall SOA formation. The relatively importance, however, may dependent regionally, and is deemed higher in regions with less VOC emission controls. We speculate that the inclusion of SOA would

possibly lead to a higher role of the transboundary pollution mainly for those sectors emitting PM and VOCs (e.g. residential, and to some extent transport and industry).”

**4.4:** It seems that rather than aggregation the authors could consider some metrics that are normalized with regards to the country size or population.

In this work we decided to aggregate the 56 FASST regions into 10 world regions based on the geographical location and as much as possible the degree of development and emissions (of course African countries do not have all the same degree of development etc., but for us it made more sense to group them together instead of putting some African countries with Russian or Latin America countries because of similar size or population). Moreover, the population information is taken into account when calculating the population weighted PM concentrations for the aggregated regions. Population data are presented in Table S2. However, in order to make mortality results more comparable among countries we included the normalized PD metric in Table 4.

**Table 4 – Absolute and population size normalized number of premature deaths/year due to anthropogenic PM$_{2.5}$ air pollution in world regions and corresponding uncertainty range.**

|  | PD (thousand deaths/year) | Normalized PD (deaths/year/million people) |
|---|---|---|
| China+ | 670 (350 - 100) | 669 |
| India+ | 610 (270 - 960) | 609 |
| Europe | 260 (140 - 480) | 405 |
| SE Asia | 150 (83 - 250) | 50 |
| Russia | 110 (67 - 240) | 449 |
| North America | 100 (55 - 170) | 306 |
| Africa | 74 (34 - 160) | 90 |
| Middle East | 56 (32 - 97) | 237 |
| Latin America | 26 (14 - 53) | 49 |
| Oceania | 0.055 (0.034 - 0.12) | 2 |

**4.20:** Here and elsewhere the Janssens-Maenhout (2017, submitted) paper is cited, although it's hard to evaluate what information is contained therein.

We clarified line 20 at page 4 as following:

“Uncertainty values of the activity data by sector and country are obtained from Table 2 of Janssens-Maenhout et al. (2017, submitted) and Olivier et al. (2016). Using this approach, the uncertainty in the global total anthropogenic CO2 emissions is estimated to range from -9% to +9% (95% confidence interval), which is the result from larger uncertainties of about +/-15% for non-Annex I countries, whereas uncertainties of less than +/-5% are obtained for the 24OECD90 countries for the time series from 1990 (Olivier et al, 2016) reported to UNFCCC.”

**About Figure 1:** It's not clear – are the % contributions to the average PM2.5 in each region, or to the population-weighted average PM2.5 in each region?

Percentages represent the contributions to the population-weighted average PM2.5 in each region. Figure caption has been modified accordingly.

**7.34:** I think the impacts of the residential sector on indoor air quality are well known and have been documented in many previous studies that could be cited.

The following papers are now cited in the text:

The residential sector is one of the most significant sources of PM all over the world, potentially also affecting indoor air quality (e.g. Ezzati, 2008; Lim et al., 2013; Chafe et al., 2014 ).

**7.39:** Similarly, the role of the agricultural sector or NH3 in particularly has been noted in several previous and recent studies. The authors continue to cite only Maas and Grennfelt, 2016, despite the broader literature available for comparison.

The following papers are now cited in the text: Pozzer et al. (2017), Tsimpidi et al. (2007), Zhang et al. (2008), Backes et al. (2016) and Erisman et al. (2004).

"Interestingly, the agricultural sector is affecting pollution in Asia as well as in Europe (Backes et al., 2016; Erisman et al., 2004) and North America, confirming the findings of the UNECE Scientific Assessment Report and several other scientific publications (Maas and Grennfelt, 2016;Pozzer et al., 2017;Tsimpidi et al., 2007;Zhang et al., 2008)."

**8.9-11:** Can the authors explain why primary emissions play such a large role in the uncertainty analysis, compared to their contribution to absolute PM2.5 concentration?

Primary PM emissions are mainly emitted from the residential, transport and to a smaller extent industrial sectors and they are characterized by the largest values of uncertainty. With the exception of the countries where the contribution of the power generation sector is relevant (which mainly leads to the formation of secondary inorganic components of PM), the other countries are dominated by the remaining sources highly emitting primary PM which are therefore strongly contributing to the final PM2.5 concentration.

**9.20:** Given that this work doesn't include anthropogenic SOA, what is the role of NMVOCs in PM2.5 formation? I guess I was just surprised to see these mentioned here.

In section 3.4.2 we rank the sector specific contribution to emission uncertainties for each of the pollutant provided by the HTAP_v2.2 inventory. As the Reviewer pointed out, TM5-FASST does not model SOA formation from anthropogenic VOCs. However, in order to provide a complete overview on the sector contribution to emission inventories' uncertainty we reported this information also for anthropogenic NMVOCs. This analysis wants to assess the emission

inventories uncertainty and it is independent from the model or source-receptor model we use to estimate PM concentrations.

**11.34:** What is the "urban increment subgrid adjustment"?

As extensively discussed in van Dingenen et al. (submitted, 2018), to better represent the actual mean population exposure within a grid cell some adjustments are included in the TM5-FASST tool. A first adjustment is performed based on the assumption that the spatial distribution of primary emitted PM2.5 correlates with population density; then information on urban and rural population grids is included and further assumptions are also applied (e.g. primary PM2.5 from the residential and the surface transport sectors are contributing to the local (urban) increment, while other aerosol precursor components and other sectors are assumed to be homogenously distributed over the grid cell). Secondary PM2.5 is formed over longer time scales and therefore more homogeneously distributed at the regional scale.

The following sentence has been therefore added into the manuscript:

"When comparing mortality estimates we need to take into account that several elements affect the results, like the resolution of the model, the urban increment subgrid adjustment (including information on urban and rural population, refer to van Dingenen et al. (submitted, 2018)), the inclusion or not of natural components, the impact threshold value used, and RR functions."

**11.33 - 35:** I strongly agree that these factors are critical towards making these comparisons, as are sources of information such as population densities and baseline mortality rates. For those precise reasons, the authors should provide details on these aspects as used in their study, as have been provided in the cited works, in order to make such comparisons possible and meaningful.

The manuscript has been rephrased as following:

"When comparing mortality estimates we need to take into account that several elements affect the results, like the resolution of the model, the urban increment subgrid adjustment (including information on urban and rural population, refer to van Dingenen et al. (submitted, 2018)), the inclusion or not of natural components, the impact threshold value used, and RR functions. In this study we used pollution the population weighted PM2.5 concentration (excluding natural components) at 1x1 degree resolution as metric for estimating health effects due to air, with a threshold value of 5.8 µg/m3, no urban increment adjustment, and relative risk functions accordingly with Burnett et al. (2014).

**12.10-12:** What it is about these regions that given them such relatively large extra-regional contributions to PM2.5 health impacts?

As shown in Fig.3, Hungary, Czech Republic, Mongolia and the Gulf region are characterized by a very high fraction of transported pollution and therefore the corresponding extra-regional contribution to the health impacts is high.

The manuscript has been rephrased as following:

"However, there are marked exceptions, such as the Gulf region, Hungary, Czech Republic, Mongolia, etc., where the extra-regional and within-region contributions to mortality are at least comparable. In fact Hungary and Czech Republic are strongly influenced by polluted regions in Poland (mainly); likewise Mongolia is suffering from the vicinity of source in China. The Gulf region produces a lot of its own pollution, but is also influenced by transport from Africa and Eurasia as reported by Lelieveld et al. (2009)."

Editorial:

**2.23:** "not to the least" change to "not the least"
Done

**2.35:** "at sector" change to "at the sector"
Done

**2.36:** "on the potential" change to "of the potential"
Done

**3.19:** Some of this sentence seems to be missing.
The sentence has been corrected as following:
"In order to calculate $PM_{2.5}$ concentrations from the HTAP_v2.2 emissions, we deployed the gridded TM5-FASST version 1.4b (Van Dingenen et al., 2017, in preparation)."

**4.16:** "as following" change to "as follows"
Done

**6.16:** "across" change to "an across"
Done

**8.23:** "Europe the" change to "Europe, the"

Done

**References**

Backes, A. M., Aulinger, A., Bieser, J., Matthias, V., and Quante, M.: Ammonia emissions in Europe, part II: How ammonia emission abatement strategies affect secondary aerosols, Atmospheric Environment, 126, 153-161, https://doi.org/10.1016/j.atmosenv.2015.11.039, 2016.

Brandt, J., Silver, J. D., Christensen, J. H., Andersen, M. S., Bønløkke, J. H., Sigsgaard, T., Geels, C., Gross, A., Hansen, A. B., Hansen, K. M., Hedegaard, G. B., Kaas, E., and Frohn, L. M.: Assessment of past, present and future health-cost externalities of air pollution in Europe and

the contribution from international ship traffic using the EVA model system, Atmos. Chem. Phys., 13, 7747-7764, 10.5194/acp-13-7747-2013, 2013.

Brauer, M., Freedman, G., Frostad, J., Van Donkelaar, A., Martin, R. V., Dentener, F., Dingenen, R. v., Estep, K., Amini, H., and Apte, J. S.: Ambient air pollution exposure estimation for the global burden of disease 2013, Environmental Science & Technology, 50, 79-88, 2015.

Burnett, R. T., C. A. Pope, M. Ezzati, C. Olives, S. S. Lim, S. Mehta, H. H. Shin, Hwashin H., Singh, G., Hubbell, B., Brauer, M., Anderson, H. R., Smith, K. R., Balmes, J. R., Bruce, N. G., Kan, H., Laden, F., Prüss-Ustün, A., Turner, M. C., Gapstur, S. M., Diver, W. R., Cohen, A.: An Integrated Risk Function for Estimating the Global Burden of Disease Attributable to Ambient Fine Particulate Matter Exposure, Environmental Health Perspectives, 122(4),397-403 doi:10.1289/ehp.1307049 2014.

Cohen, A. J., Brauer, M., Burnett, R., Anderson, H. R., Frostad, J., Estep, K., Balakrishnan, K., Brunekreef, B., Dandona, L., Dandona, R., Valery, F., Greg, F., Bryan, H., Amelia, J., Haidong, K., Luke, K., Yang, L., Randall, M., Lidia, M., C Arden, P. I., Hwashin, S., Kurt, S., Gavin, S., Matthew, T., Rita, v. D., Aaron, v. D., Theo, V., Christopher J L, M., and Mohammad H, F.: Estimates and 25-year trends of the global burden of disease attributable to ambient air pollution: an analysis of data from the Global Burden of Diseases Study 2015, The Lancet, 389, 1907-1918, 2017.

Chafe, Z. A., Brauer, M., Klimont, Z., Van Dingenen, R., Mehta, S., Rao, S., Riahi, K., Dentener, F., and Smith, K. R.: Household Cooking with Solid Fuels Contributes to Ambient PM(2.5) Air Pollution and the Burden of Disease, Environmental Health Perspectives, 122, 1314-1320, 10.1289/ehp.1206340, 2014.

Ezzati, M.: Indoor air pollution and health in developing countries, The Lancet, 366, 104-106, 10.1016/s0140-6736(05)66845-6.

Erisman, J. W., and Schaap, M.: The need for ammonia abatement with respect to secondary PM reductions in Europe, Environmental Pollution, 129, 159-163, https://doi.org/10.1016/j.envpol.2003.08.042, 2004.

Farina, S. C., Adams, P. J., and Pandis, S. N.: Modeling global secondary organic aerosol formation and processing with the volatility basis set: Implications for anthropogenic secondary organic aerosol, Journal of Geophysical Research: Atmospheres, 115, 10.1029/2009jd013046, 2010.

Henneman et al., 2017. Evaluating the effectiveness of air quality regulations: A review of accountability studies and frameworks. Journal of the Air & Waste Management Association Volume 67, 2017 - Issue 2.

Im, U., Brandt, J., Geels, C., Hansen, K. M., Christensen, J. H., Andersen, M. S., Solazzo, E., Kioutsioukis, I., Alyuz, U., Balzarini, A., Baro, R., Bellasio, R., Bianconi, R., Bieser, J., Colette, A., Curci, G., Farrow, A., Flemming, J., Fraser, A., Jimenez-Guerrero, P., Kitwiroon, N., Liang,

C. K., Pirovano, G., Pozzoli, L., Prank, M., Rose, R., Sokhi, R., Tuccella, P., Unal, A., Vivanco, M. G., West, J., Yarwood, G., Hogrefe, C., and Galmarini, S.: Assessment and economic valuation of air pollution impacts on human health over Europe and the United States as calculated by a multi-model ensemble in the frame work of AQMEII3, Atmos. Chem. Phys. Discuss., 2017, 1-34, 10.5194/acp-2017-751, 2017.

Karagulian, F., Belis, C. A., Dora, C. F. C., Prüss-Ustün, A. M., Bonjour, S., Adair-Rohani, H., and Amann, M.: Contributions to cities' ambient particulate matter (PM): A systematic review of local source contributions at global level, Atmospheric Environment, 120, 475-483, https://doi.org/10.1016/j.atmosenv.2015.08.087, 2015.

Lelieveld, J., Hoor, P., Jöckel, P., Pozzer, A., Hadjinicolaou, P., Cammas, J. P., and Beirle, S.: Severe ozone air pollution in the Persian Gulf region, Atmos. Chem. Phys., 9, 1393-1406, 10.5194/acp-9-1393-2009, 2009.

Lelieveld, J., Evans, J. S., Fnais, M., Giannadaki, D., and Pozzer, A.: The contribution of outdoor air pollution sources to premature mortality on a global scale, Nature, 525, 367-371, 10.1038/nature15371, 2015.

Lim, S. S., Vos, T., Flaxman, A. D., Danaei, G., Shibuya, K., Adair-Rohani, H., AlMazroa, M. A., Amann, M., Anderson, H. R., Andrews, K. G., Aryee, M., Atkinson, C., Bacchus, L. J., Bahalim, A. N., Balakrishnan, K., Balmes, J., Barker-Collo, S., Baxter, A., Bell, M. L., Blore, J. D., Blyth, F., Bonner, C., Borges, G., Bourne, R., Boussinesq, M., Brauer, M., Brooks, P., Bruce, N. G., Brunekreef, B., Bryan-Hancock, C., Bucello, C., Buchbinder, R., Bull, F., Burnett, R. T., Byers, T. E., Calabria, B., Carapetis, J., Carnahan, E., Chafe, Z., Charlson, F., Chen, H., Chen, J. S., Cheng, A. T.-A., Child, J. C., Cohen, A., Colson, K. E., Cowie, B. C., Darby, S., Darling, S., Davis, A., Degenhardt, L., Dentener, F., Des Jarlais, D. C., Devries, K., Dherani, M., Ding, E. L., Dorsey, E. R., Driscoll, T., Edmond, K., Ali, S. E., Engell, R. E., Erwin, P. J., Fahimi, S., Falder, G., Farzadfar, F., Ferrari, A., Finucane, M. M., Flaxman, S., Fowkes, F. G. R., Freedman, G., Freeman, M. K., Gakidou, E., Ghosh, S., Giovannucci, E., Gmel, G., Graham, K., Grainger, R., Grant, B., Gunnell, D., Gutierrez, H. R., Hall, W., Hoek, H. W., Hogan, A., Hosgood, H. D., III, Hoy, D., Hu, H., Hubbell, B. J., Hutchings, S. J., Ibeanusi, S. E., Jacklyn, G. L., Jasrasaria, R., Jonas, J. B., Kan, H., Kanis, J. A., Kassebaum, N., Kawakami, N., Khang, Y.-H., Khatibzadeh, S., Khoo, J.-P., Kok, C., Laden, F., Lalloo, R., Lan, Q., Lathlean, T., Leasher, J. L., Leigh, J., Li, Y., Lin, J. K., Lipshultz, S. E., London, S., Lozano, R., Lu, Y., Mak, J., Malekzadeh, R., Mallinger, L., Marcenes, W., March, L., Marks, R., Martin, R., McGale, P., McGrath, J., Mehta, S., Memish, Z. A., Mensah, G. A., Merriman, T. R., Micha, R., Michaud, C., Mishra, V., Hanafiah, K. M., Mokdad, A. A., Morawska, L., Mozaffarian, D., Murphy, T., Naghavi, M., Neal, B., Nelson, P. K., Nolla, J. M., Norman, R., Olives, C., Omer, S. B., Orchard, J., Osborne, R., Ostro, B., Page, A., Pandey, K. D., Parry, C. D. H., Passmore, E., Patra, J., Pearce, N., Pelizzari, P. M., Petzold, M., Phillips, M. R., Pope, D., Pope, C. A., III, Powles, J., Rao, M., Razavi, H., Rehfuess, E. A., Rehm, J. T., Ritz, B., Rivara, F. P., Roberts, T., Robinson, C., Rodriguez-Portales, J. A., Romieu, I., Room, R., Rosenfeld, L. C., Roy, A., Rushton, L., Salomon, J. A., Sampson, U., Sanchez-Riera, L., Sanman, E., Sapkota, A., Seedat, S., Shi, P., Shield, K., Shivakoti, R., Singh, G. M., Sleet, D. A., Smith, E., Smith, K. R., Stapelberg, N. J. C., Steenland, K., Stöckl, H., Stovner, L. J., Straif, K., Straney, L., Thurston, G. D., Tran, J. H.,

Van Dingenen, R., van Donkelaar, A., Veerman, J. L., Vijayakumar, L., Weintraub, R., Weissman, M. M., White, R. A., Whiteford, H., Wiersma, S. T., Wilkinson, J. D., Williams, H. C., Williams, W., Wilson, N., Woolf, A. D., Yip, P., Zielinski, J. M., Lopez, A. D., Murray, C. J. L., and Ezzati, M.: A comparative risk assessment of burden of disease and injury attributable to 67 risk factors and risk factor clusters in 21 regions, 1990–2010: a systematic analysis for the Global Burden of Disease Study 2010, The Lancet, 380, 2224-2260, 10.1016/s0140-6736(12)61766-8, 2013.

Morgan, R.K., 2012. Environmental Impact assessment: the state of the art. Impact Assessment and Project Appraisal 30, 5-14.

Olivier, J.G.J., Janssens-Maenhout, G., Muntean, M., Peters, J.A.H.W: Trends in global CO2 emissions: 2016 report, JRC 103425, 2016.

Pozzer, A., Tsimpidi, A. P., Karydis, V. A., de Meij, A., and Lelieveld, J.: Impact of agricultural emission reductions on fine particulate matter and public health, Atmos. Chem. Phys. Discuss., 2017, 1-19, 10.5194/acp-2017-390, 2017.

Silva, R. A., Adelman, Z., Fry, M. M., and West, J. J.: The impact of individual anthropogenic emissions sectors on the global burden of human mortality due to ambient air pollution, Environmental Health Perspectives, 124, 1776, 2016.

Solazzo, E., Riccio, A., van Dingenen, R., Valentini, L., Galmarini, S.: Evaluation and uncertainty estimation of air quality impact on crop yields and premature deaths using a multi-model ensemble, Science of the Total Environment, submitted, 2018.

Thunis, P., Degraeuwe, B., Pisoni, E., Ferrari, F., and Clappier, A.: On the design and assessment of regional air quality plans: The SHERPA approach, Journal of Environmental Management, 183, 952-958, https://doi.org/10.1016/j.jenvman.2016.09.049, 2016.

Tsimpidi, A. P., Karydis, V. A., and Pandis, S. N.: Response of inorganic fine particulate matter to emission changes of sulfur dioxide and ammonia: The eastern United States as a case study, J. Am. Waste Manage. Assoc., 57, 1489–1498, 2007.

Van Dingenen, R., Dentener, F., Crippa, M., Leitao-Alexandre, J., Marmer, E., Rao, S., Solazzo, E., and Valentini, L.: TM5-FASST: a global atmospheric source-receptor model for rapid impact analysis of emission changes on air quality and short-lived climate pollutants, 2018, submitted.

Van Donkelaar, A., Martin, R. V., Brauer, M., Hsu, N. C., Kahn, R. A., Levy, R. C., Lyapustin, A., Sayer, A. M., and Winker, D. M.: Global estimates of fine particulate matter using a combined geophysical-statistical method with information from satellites, models, and monitors, Environmental Science & Technology, 50, 3762-3772, 2016.

Watkiss, P., Pye, S. and Holland, M.: CAFE CBA: Baseline Analysis 2000 to 2020. Service Contract for Carrying out Cost-Benefit Analysis of Air Quality Related Issues, in particular in

the clean Air for Europe (CAFE) Programme, April 2005, http://www. cafe-cba.org/assets/baseline analysis 2000-2020 05-05.pdf (last access: 25 February 2013), 2005.

Zhang, Y., Wu, S.-Y., Krishnan, S., Wang, K., Queen, A., Aneja, V. P., and Arya, S. P.: Modeling agricultural air quality: Current status, major challenges, and outlook, Atmos. Environ., 42, 3218–3237, 2008.

---

## Author Comment (AC4) · 16 Feb 2018

The authors are grateful to Referee#2 for the helpful comments that helped improve the manuscript. Due to the strict link between this publication and the work recently submitted by van Dingenen et al. (submitted, 2018) about the TM5-FASST methodology, we offered the possibility to the Editor and the Reviewer to access the work of van Dingenen et al. (submitted, 2018) although not yet published in ACPD. We feel that we have been able to address all concerns, as outlined below. Thanks to the Reviewer's comments, we also realized that some methodological aspects of the TM5-FASST tool could have been further developed also in the publication of van Dingenen et al. (submitted, 2018). Therefore, discussions on the comparison between PM2.5 modeled concentrations vs. the measured ones, as well as further details about the extension of the "perturbation approach" to the attribution of sectors and sources will be included in the review phase of the paper by van Dingenen et al. (submitted, 2018). We feel that we have been able to address all concerns, as outlined below.

**Anonymous Referee #2**

The authors coupled the HTAPv2.2 global air pollutant emission inventory with the global source receptor model TM5-FASST to evaluate the relative contribution of the major anthropogenic emission sources to air quality and health in 2010. They focused on PM2.5 due to its negative impacts on human health. The objective of this paper is to evaluate the emissions uncertainties at sector and regional levels, and their propagation in modeled PM2.5 concentrations and associated impacts on health. Although the authors state that they have two objectives, I do not quite understand the difference between the two. I find that what the paper is trying to do is important but there are some major problems that need to be addressed before this can be published in ACP.

First, if the objective is to understand the health impacts of PM2.5, I believe that the authors need to make sure that their model simulations match with the observations. I do not find the existing comparison in the paper (p. 7, l. 21-28) very convincing. The authors could have at least compared with the recent WHO database of annual PM2.5 concentrations at various cities (http://www.who.int/phe/health_topics/outdoorair/databases/cities/en/). For the US, there is much better database that could be used (https://www.epa.gov/outdoor-air-quality-data/interactive-mapair- quality-monitors). The authors seem to allude that it is ok to not include the natural emissions but I disagree and think that the natural emissions need to be included in the model.

We acknowledge the suggestion of the Reviewer about the comparison of the PM2.5 concentrations estimated by TM5-FASST with other databases in addition to what already provided in our manuscript.
The EPA air quality statistics for USA for the year 2010 (https://www.epa.gov/outdoor-air-quality-data/air-quality-statistics-report) report an annual concentration of PM2.5 of 12.0 ug/m3 which is higher compared to our estimate (7.8 ug/m3) because measured PM2.5 concentrations include all sources of PM (e.g. large scale biomass burning and SOA from anthropogenic sources which are not accounted in our study).
However, the TM5-FASST model developed by van Dingenen et al. (submitted) has been validated against concentration estimates derived from the WHO database and satetellite-based

measurements (excluding dust and sea salt). We report below some details about these comparisons which will be included in the submitted manuscript by van Dingenen et al.

Figure 1 shows the comparison between the PM2.5 concentrations modeled by TM5-FASST and the measured ones reported in the WHO database for different world regions (i.e. EUR=Europe, NAM=North America, China, S-ASIA=Southern Asia, LAM=Latin America, AFR=Africa). This includes measurement points as well as PM2.5 concentration estimates based on a fraction of PM10 measurements (e.g. almost all points for the comparison in China are based on this second method). Quite good agreement is observed for Europe, North America and partly China where measurements have been performed over longer time compared to developing countries and they are based on quite consolidated methods. The comparison for Latin America and Africa is much less robust and the scatter possibly highlights a non-optimal modeling of large scale biomass burning for the TM5-FASST model. Figure 2 reports the comparison of WHO regional average of urban stations against the FASST population weighted average of grid cells. Similarly to the findings of Figure 1, the comparison for industrialized countries is very good, while for other developing regions the agreement is less satisfactory both due to less accurate measurements (e.g. reported by WHO) and lower quality modeling of specific sources by TM5-FASST (e.g. large scale biomass burning).

TM5-FASST modeled PM2.5 concentrations have been also validated against satellite products (see Figure 3) which are based on aerosol optical depth measurements together with chemical transport model information to retrieve from the total column the information of PM concentrations in the lowest layer of the atmosphere (Boys et al., 2014; van Donkelaar et al., 2010, 2014).

[Figure]

[Figure]

Figure 1 - TM5-FASST grid-cell mean (with urban increment parameterisation) versus individual monitoring stations (WHO consolidated database, including both measured and estimated PM2.5).

[Figure]

Figure 2 - WHO regional average of urban stations (+/- 1 stdev) and FASST population weighted average of grid cells.

| EUROPE | USA |
|---|---|

[Figure]

Figure 3 - Comparison between TM5-FASST and satellite products for world regions (Boys et al., 2014; van Donkelaar et al., 2010, 2014).

Therefore we added in the manuscript the following discussion in section 3.3:

"The TM5-FASST model developed by van Dingenen et al. (2018) has been validated against concentration estimates derived from the WHO database and satellite-based measurements (van

Donkelaar et al., 2010, 2014). General good agreement is found between the PM2.5 concentrations modeled by TM5-FASST and the measured ones reported in the WHO database for Europe (within 20% deviation), North America (within 5% deviation) and partly China due to the higher accuracy of the measurements. The comparison for Latin America and Africa is much less robust (40-60% deviation from the 1:1 line) and the scatter possibly highlights a non-optimal modeling of specific sources relevant for these regions by TM5-FASST (e.g. large scale biomass burning) by the TM5-FASST model. Similar results are also found comparing regional averages of urban stations from WHO against the FASST population weighted average of grid cells. The TM5-FASST modeled PM2.5 concentrations have been compared to satellite products which are based on aerosol optical depth measurements together with chemical transport model information to retrieve from the total column the information of PM concentrations in the lowest layer of the atmosphere (Boys et al., 2014; van Donkelaar et al., 2010, 2014). The regional comparison shows consistent results with the ground based measurements comparison (e.g. good agreement for EU and USA within 10% deviation, while lower agreement for developing and emerging countries)."

Second, I find that the emissions uncertainty estimate seems a little simplistic to only assess within the HTAP inventory, considering the existing differences among various inventories. Also, if the RCP emissions for the year 2000 are used as a baseline, to me it makes more sense to use RCP 2010 in their analysis, rather than switching to HTAP v2.2. Or if the HTAP is to be used, the uncertainty analysis should include the differences in estimates between RCP 2010 and HTAP v2.2. Also, it might be a good idea to compare with some other estimates in existing studies that have estimated emissions uncertainties for certain countries.

We would like to stress that the aim of this work is not to compare different emission inventories since this has already been done in other publications (specifically regarding the HTAP_v2 inventory, e.g. Janssens-Maenhout et al., 2015, Crippa et al., 2016), but we aim at addressing the uncertainty of sector specific emissions from this inventory in a quantitative way as well as the differences we observe from one region to the other, based on the uncertainty of activity data and emission factors. There are several reasons to use HTAP_v2.2 and not e.g. the RCP2000 as the basis for our assessment of emission propagation. The TF HTAP aims at bringing policy relevant information, and to this end, it has compiled a policy relevant emission inventory (HTAP_v2.2) for the most recently available year. While the RCP2000 was at the basis of the FASST calculations, and presented the best community emissions effort at the time, we feel that it is now superseded by the more accurate HTAP_v2.2. Given our focus on regional (and not so much gridded) results, we feel that this choice is justified.

Therefore we added the following explanation in Section 2.1 of the manuscript:

"The aim of this work is to address the uncertainty of sector specific emissions from this inventory in a quantitative way as well as the differences we observe from one region to the other, based on the uncertainty of activity data and emission factors. As discussed in the next section, the reason to use HTAP_v2.2 and not e.g. the RCP2000 as the basis for our assessment of emission propagation is that the TF HTAP aims at bringing policy relevant information, and to this end, it has compiled a policy relevant emission inventory (HTAP_v2.2) for the most recently available year. While the RCP2000 was at the basis of the FASST calculations, and presented the

best community emissions effort at the time, the HTAP_v2.2 inventory is now day much more accurate in particular given the focus on regional (and not so much gridded) emission analysis of our work."

Differently from CO2 for which emission uncertainties are much better know, literature studies dealing with uncertainty of emission inventories of all air pollutants show a lack of information on the corresponding uncertainties (while intercomparisons among different inventories are often shown). In addition, literature studies often make use of region- and sector-specific emission inventories and they do not provide a global view on all pollutants, sectors and regions (Hoesly et al., 2017).

However, we took into account the Reviewer's comment including some references with literature studies on emission inventory uncertainties.

Page 9, line 16: Smith et al. (2011) report a range of regional uncertainty for SO2 up to 30% while our estimates are slightly higher (up to 50%).

Page 9, line 24: "Among all air pollutants, represent one of the most uncertain pollutant due to very different combustion conditions, different fuel qualities and lack of control measures (Klimont et al., 2017)."

Third, I also find it problematic that important details and assumptions of TM5-FASST methodology are described in the paper that is still under preparation. I am assuming that the $\Delta PM_{ref}$ and $\Delta E_{ref}$ in Eq. 1 refer to the difference between the TM5-FASST simulation results for PM2.5 (and also PM10 as well?) using the RCP baseline and the perturbation (-20%) and the emissions themselves, respectively. However, I find it troublesome that these stay constant when the emissions change for all regions and sectors. We know that PM2.5 formation is a non-linear process and I do not believe it would work in a linear form for every region for every sector. If it does, maybe that is because simulation uses too coarse of a resolution and the result does not seem realistic.

The paper by van Dingenen et al. (submitted) has now been submitted to ACP. It contains a detailed description on the methodology and documents the validity of the linearity assumption for PM2.5 (the simulations were done only for PM2.5 and not PM10). Unfortunately anthropogenic SOA is not explicitly modeled in TM5 but treated as a pseudo-emission. In the manuscript we clarified the concept of dE and dPM as following:

"The reduced-form model TM5-FASST is computing the concentration resulting from an arbitrary emission scenario $E_i$ using a perturbation approach, i.e. the difference between $E_i$ and $E_{i,ref}$ *(dE)* is considered as a perturbation on $E_{ref}$ and the resulting concentration is evaluated as a perturbation *dPM* on the reference concentration."

Also, it seems problematic that no explicit treatment of anthropogenic SOA is considered.

Unfortunately we do not have estimates of the contribution of anthropogenic SOA, as the gas phase chemical degradation scheme didn't include emissions of the relevant precursor gases. The importance of anthropogenic SOA ranges regionally widely, as demonstrated by a recent study by Farina et al. (2010) indicating a global source of 1.6 Tg, or ca. 5.5 % of the overall SOA formation. The relatively importance, however, may dependent regionally, and is deemed higher in regions with less VOC emission controls. Overall, we feel that the uncertainty stemming from our knowledge in SOA formation is higher than the omission of anthropogenic SOA. We would also like to mention that the development of the volatility-based SOA formation approach, means that the boundaries between 'primary' and 'secondary' SOA are disappearing, making it difficult to attribute organic aerosol to either primary, secondary (or natural-anthropogenic), as they strongly interact. Nevertheless, we speculate that the inclusion of SOA would possibly lead to a higher role of the transboundary pollution mainly for those sectors emitting PM and VOCs (e.g. residential, and to some extent transport and industry).

Therefore we added the following sentences to the manuscript:

"The importance of anthropogenic SOA ranges regionally widely, as demonstrated by a recent study by Farina et al. (2010) indicating a global source of 1.6 Tg, or ca. 5.5 % of the overall SOA formation. The relatively importance, however, may dependent regionally, and is deemed higher in regions with less VOC emission controls. We speculate that the inclusion of SOA would possibly lead to a higher role of the transboundary pollution mainly for those sectors emitting PM and VOCs (e.g. residential, and to some extent transport and industry)."

Is it correct that TM5-FASST simulations were run for each sector separately and also for all the sectors combined? That is how it looks like from Figure 4. If so, can the authors confirm that the sum of concentrations from each of the sectors run separately are similar to the values when the simulation was done including all the sector emissions together? It would be a nice test to check the linearity in the model. If the simulations were done in this way, then what was the reason equation 1 had to be used? The authors could have easily calculated the impact of each sector using these simulations instead?

In general, the reduced-form model TM5-FASST is computing the concentration resulting from an arbitrary emission scenario $E_s$ using a perturbation approach, i.e. the difference between $E_s$ and $E_{ref}$ ($dE_s$) is considered as a perturbation on $E_{ref}$ and the resulting concentration is evaluated as a perturbation $dPM$ on the reference concentration, hence:

$$PM(E_s) = PM(E_{ref} + dE_s) = PM_{ref} + dPM = PM_{ref} + SRC \cdot dE_s \qquad (a)$$

Where $dE_s = E_s - E_{ref}$ and $E_{ref}$ is the RCP reference scenario from which the $SRC$ have been computed.

The contribution of a single sector $j$ is calculated as the difference between the concentration including all sectors, and the concentration from the emissions excluding the single sector $j$

$$PM(E_{s,j}) = PM(E_s) - PM(E_s - E_{s,j}) = SRC \cdot [dE_s - d(E_s - E_{s,j})] = SRC \cdot E_{s,j}$$

If the linearity holds, the sum of *PM(Es,j)* over all sectors j should be equal to *PM(Es)*, or:

$$\sum_j PM(E_{s,j}) = PM_{ref} + SRC \cdot (E_s - E_{ref})$$

In Figure 4 we compare both sides of the equation to demonstrate that indeed the linearity assumption holds sufficiently well.

A caveat of TM5-FASST is that no sector-specific SRC have been computed (except for international shipping which was evaluated separately), and consequently our single sector analysis implicitly assumes that the spatial distribution of pollutant emissions at the resolution considered here (1°x1°) is similar for all sectors within each source region. Taking into account that

(1) the spatial distribution of primary anthropogenic emissions is commonly generated using population density as the major proxy (except for large scale biomass burning) – e.g. domestic burning, transport, industry

(2) in many cases, the emission of secondary pollutant precursors is dominated by a single sector (e.g. $NH_3$ mainly from agriculture, $NO_x$ mainly from transport, $SO_2$ mainly form energy production)

we deem that the spatial distribution of the individual sectors can be estimated sufficiently accurately for the present analysis, as shown in Figure 4 which has been obtained from the 'total' SRC, applied on single-sector emissions. A similar approach has been recently implemented by Liang et al. (2018) based on the HTAP2 source receptors.

The TM5-FASST runs were performed for different scenarios, comparing the reference HTAP_v2.2 emissions with a scenario where emissions from one single sector were subtracted from the total emissions. Then comparing the reference case and each scenario (REF-sectori), the contribution of each sector to PM2.5 concentrations is estimated. This approach is based on the assumption that the individual sector contributions add up linearly to total PM2.5, as mentioned in the paper. The paper by Van Dingenen et al. describing the whole TM5-FASST methodology has just been submitted to ACP (van Dingenen et al., submitted) Equation 1 represents the basis of the TM5-FASST method, since it describes how a variation in the emissions (delta emissions) determines a delta in PM2.5 based on the source receptor relationships.

[Figure]

Figure 4 – Comparison between the total modeled PM2.5 concentration and the sum of the sectors.

The following discussion on how to apply the "perturbation approach" on the sector and source attribution will be also included in the paper by van Dingenen et al. (submitted, 2018):

Equation (2) expresses the 'perturbation' approach applied in the linearized TM5-FASST model, i.e. an arbitrary emission scenario is evaluated as a deviation from the base emission scenario, and the resulting pollutant concentration is obtained as the sum of the base concentration and a delta term, the latter proportional to the emission deviation from the base case (Figure 1).

A particular application of TM5-FASST is the attribution of the (anthropogenic) pollutant concentration to individual source regions or sectors. Due to the fixed contribution of the base concentration which does not contain information on the originating sources, Eq. (2) is not immediately suitable for such an analysis. Instead, we calculate for each individual source the contributing part by first evaluating all sources together ('total' simulation'), and subsequently subtracting the individual source emissions ($E_s$) from the total, evaluating the resulting pollutant concentration ($C_{minus\_s}$), and making the difference with the 'total simulation' to obtain the single source contribution ($C_s$).

$$C_{j,tot}(y) = C_{j,base}(y) + \sum_{n_x} \sum_{n_i} A_{ij}[x,y] \cdot [E_{i,tot}(x) - E_{i,base}(x)] \qquad (2)$$

$$C_{j,minus\_s}(y) = C_{j,base}(y) + \sum_{n_x} \sum_{n_i} A_{ij}[x,y] \cdot [E_{i,tot}(x) - E_{i,s}(x) - E_{i,base}(x)] \qquad (4)$$

$$C_{j,s}^*(y) = C_{j,tot}(y) - C_{j,minus\_s}(y) = \sum_{n_x} \sum_{n_i} A_{ij}[x,y] \cdot E_{i,s}(x) \qquad (5)$$

We can now reconstruct $C_{j,tot}^*$ as the sum of the individual source contributions:

$$C_{j,tot}^*(y) = \sum_{n_s} C_{j,s}^*(y) \qquad (6)$$

$C_{j,tot}^*(y)$ is equivalent to $C_{j,tot}(y)$ in Eq. 2 only if

$C_{j,base}(y) = \sum_{n_x} \sum_{n_i} A_{ij}[x,y] \cdot E_{i,base}(x)$, in other words if the emission-concentration relation is perfectly linear and passes through the origin.

In reality there is some degree of non-linearity in most emission-pollutant relation as illustrated in Figs. 3 and 6. Figure A shows for each of the FASST regions the total PM$_{2.5}$ concentration obtained by Eq. 6 versus the TM5 base simulation result, illustrating the non-linearity error resulting from the application of Eq. 6. For 43 out of 56 regions, the deviation from the base simulation is less than 30%, only 3 regions (former Soviet Union, New Zealand and Pacific) deviate more than 50% from the TM5 model result. Consistency with the 'perturbation approach' is restored by simply rescaling the individual source contributions:

$$C_{j,s}(y) = \frac{C_{j,tot}(y)}{C_{j,tot}^*(y)}\ C_{s,j}^*(y) \qquad (7)$$

This approach is valid for evaluating the attribution by sector as well as by source region.

[Figure]

Figure A: Scatter plot of regionally averaged PM2.5 concentration (including all anthropogenic components) obtained as the sum of individual source region contribution by linear scaling of their respective emissions with TM5-FASST source-receptor coefficients (Eq. 6), versus the regional average obtained by the full TM5 model.

I have a hard time understanding the sentence on p. 9 l. 5-8. How do the authors determine the relative contribution to total emission inventory uncertainty? Are the authors using the

uncertainty for a specific sector over the total uncertainty for a specific pollutant as the "average sector relative contribution to total emission inventory"? If so, this does not necessarily take the magnitude of emissions into account and so maybe just looking at this value and deciding which sector to focus on might be a little too simplistic?

As discussed in Section 2.3, "uncertainties have been estimated for each emission sector for every country/region and pollutant. Then an overall uncertainty has been estimated using equation 5 (shown below) from the EMEP/EEA, 2013 Guidebook and which accounts for the weighted contribution of each sector to the overall uncertainty. Then the contribution of each sector to the overall uncertainty is given by the weight of each term of the equation compared to the others, so it does not correspond to the "average sector relative contribution to total emission inventory".

We rephrased as following:

"The complete overview of allTM5-FASST regions is provided in Fig. S2, where the share of each term of the sum of Eq.5 $\left( \sigma_{EMI\ i,c,p} * \frac{EMI_{i,c,p}}{EMI_{tot,c,p}} \right)^2$, representing the sector contribution to the uncertainty of each pollutant in each region, is reported."

Are the upper and the lower boundaries of PM2.5 concentrations (Table 2 and Figure 5) calculated based on the linear relationship between emissions per region? In other words, are they simply calculated from emissions, rather than running the simulation again in a chemical transport model?

To calculate the upper and lower boundaries of PM2.5 concentrations we used the TM5-FASST model and so they are based on the linear relationship between emissions per region. However, new emission datasets including the upper and lower range of uncertainty have been given as input for new TM5-FASST runs which gave us the upper and lower range of PM2.5 concentrations.

We added a sentence at the end of paragraph 2.3 to clarify our approach:

"Based on the upper and lower emission range per region, new TM5-FASST model runs have been performed per source region to retrieve the corresponding range of concentrations in receptor regions (therefore the total number of computations is 56*2 for the uncertainty analysis)."

**Minor comments:**
  1. I would like to see a figure that shows the 10 aggregated receptor regions, as it is unclear, for example, what China+ region includes. Does it just include Mongolia? Or also Korea and Japan?

     Table S2 of the Supplementary material already includes this information for all aggregated regions. China+ includes China and Mongolia+North Korea. We do not aim at having another Figure in the supplementary material about the regions aggregation, in order to avoid

repeating information already provided in a Table and to avoid misunderstandings with the map about the 56 TM5-FASST regions used for the model runs.

2. Why are some European countries lumped together in Figure 2 (Austria and Slovenia, for example), whereas others are not?

The following explanation has been added in the Supplementary Material (S1) to explain the TM5-FASST regions aggregation.

"The 56 TM5-FASST regions were chosen to obtain an optimal match with integrated assessment models such as IMAGE (Eickhout et al., 2004; van Vuuren et al., 2007), MESSAGE (Riahi et al., 2007), GAINS (Höglund-Isaksson and Mechler, 2005) as well as the POLES model (Russ et al., 2007; Van Aardenne et al., 2007). The grouping of small countries was motivated by (a) finding a compromise between spatial resolution and computational effort required to obtain the set of source-receptor matrices for TM5-FASST and (b) avoiding inaccurate mapping of small individual countries that are represented by only a few 1°x1° grid cells.
Most European countries are defined as individual source regions, except for the smallest countries, which have been aggregated."

3. Why are there more countries in Figure 3 than in Figure 1?

Figure 1 represents the global view using the 10 aggregated world regions, while figure 3 shows a disaggregated view making use of the original 56 TM5-FASST regions. The reason behind the aggregation to 10 regions is explained at page 4 of the manuscript: "In order to make smaller regions (e.g. European countries) comparable with larger regions (like USA, China and India), in this work an aggregation procedure to 10 world regions (refer to Table S2) has been applied (China+, India+, SE Asia, North America, Europe, Oceania, Latin America, Africa, Russia and Middle East)."

p. 2. l. 30-34: The sentence is too long and difficult to understand. Please rephrase the sentence.

The sentence has been rephrased as following:

"The objective of this study is to evaluate the relevance of uncertainties in regional sectorial emission inventories (power generation, industry, ground transport, residential, agriculture and international shipping), and its propagation in modeled $PM_{2.5}$ concentrations and associated impacts on health. We also investigate the uncertainties in $PM_{2.5}$ from within the region to extra-regional contributions."

p. 2 l. 36-37: The authors state that a second objective of the analysis is to "inform local, regional hemispheric air quality policy makers on the potential impacts of less known emission sectors or regions" but they are focusing on the "6 major anthropogenic emission sectors (l. 6-7, p. 3)." What do they mean by "less known emission sectors" then?

Less known emission sectors (and less regulated ones in terms of emissions) are the residential and agricultural sectors, so the sentence has been rephrased as following:

"A second objective of this analysis is to evaluate the importance of emission uncertainties at sector and regional level on PM2.5, to better inform local, regional and hemispheric air quality policy makers on the potential impacts of sectors with larger uncertainties less known emission sectors (e.g. residential and agriculture) or regions (e.g. developing and emerging countries).

p. 3. l. 19-20. This sentence is not finished.

The sentence has been corrected as following:

"In order to calculate PM2.5 concentrations from the HTAP_v2.2 emissions, we use the native 1°x1° resolution source-receptor gridmaps obtained for TM5-FASST_v0 (Van Dingenen et al., 2018, submitted)".

p. 3. L. 22. Why was such a coarse resolution used, when HTAPv2.2 is much finer?

At the time of creating the TM5-FASST Source receptor relationships (ca. 2007-2010), 1x1 degree global resolution was still of unprecedented high resolution (given hundreds of simulations) and more common was resolutions around 2 to 3 degrees (T42). Only since recently more global models are running on 1x1 degree or somewhat finer, but it is still difficult to make 100s of SR calculations. The 0.1x0.1 HTAP_v2 resolution is employed only in full by regional model studies that used global model results as boundary conditions.

The following sentence has been added for clarity in the manuscript:

"TM5-FASST uses aggregated regional emissions (i.e. one annual emission value per pollutant or precursor for each of the 56 regions + shipping), with an implicit underlying 1°x1° resolution emission spatial distribution from RCP year 2000 which was partly based EDGAR methodology and gridmaps."

p. 3 l. 30 relativey ->relatively
correction done

p. 7 l. 37-39 Perhaps a reference to Bauer et al. (2016) would be appropriate here.

Some changes have been made in that section, adding also more references:

"In order to understand the origin of global PM2.5 concentrations, we look at sector specific maps (Fig. 4). The power and industrial sectors are mainly contributing to PM concentrations in countries having emerging economies and fast development (e.g. Middle East, China and India), while the ground transport sector is a more important source of PM concentrations in industrialised countries (e.g. North America and Europe) and in developing Asian countries. The residential sector is one of the most significant sources of PM all over the world, potentially also affecting indoor air quality (Ezzati, 2008; Lim et al., 2013; Chafe et al., 2014)."

p. 11 l. 36. It is unclear to me where this value (7% for the global non accidental mortalities) is coming from. Can you clarify or cite the source?

We cited the source of our estimates as following:

"We also estimate that 7 % of the global non accidental mortalities from the Global Burden of Disease (http://vizhub.healthdata.org/gbd-compare; Forouzanfar et al. (2015)) are attributable to air pollution in 2010;"

p. 12 l. 10 such the Gulf -> such as the Gulf

correction done

p. 19 Table 2. How do you quantify the uncertainty for a certain pollutant for a region?

The methodology behind the uncertainty estimates for a certain pollutant and region is described in Sect. 2.3 of the manuscript and with the equations 3 and 4.
Table S3 of the Supplementary material also provides region- and pollutant- specific emission uncertainties.

Reference:
Bauer, S.E., K. Tsigaridis, and R.L. Miller, 2016: Significant atmospheric aerosol pollution caused by world food cultivation. Geophys. Res. Lett., 43, no. 10, 5394-5400, doi:10.1002/2016GL068354.

Boys, B. L., Martin, R. V., van Donkelaar, A., MacDonell, R. J., Hsu, N. C., Cooper, M. J., Yantosca, R. M., Lu, Z., Streets, D. G., Zhang, Q. and Wang, S. W.: Fifteen-Year Global Time Series of Satellite-Derived Fine Particulate Matter, Environ. Sci. Technol., 48(19), 11109–11118, doi:10.1021/es502113p, 2014.

Crippa, M., Janssens-Maenhout, G., Dentener, F., Guizzardi, D., Sindelarova, K., Muntean, M., Van Dingenen, R., and Granier, C.: Forty years of improvements in European air quality: regional policy-industry interactions with global impacts, Atmospheric Chemistry and Physics, 16, 3825-3841, 2016.

Eickhout, B., Den Elzen, M. G. J. and Kreileman, G. J. J.: The Atmosphere-Ocean System of IMAGE 2.2. A global model approach for atmospheric concentrations, and climate and sea level projections, RIVM, Bilthoven, The Netherlands. [online] Available from: http://rivm.openrepository.com/rivm/handle/10029/8936 (Accessed 10 January 2017), 2004.

Forouzanfar, M. H., Alexander, L., Anderson, H. R., Bachman, V. F., Biryukov, S., Brauer, M., Burnett, R., Casey, D., Coates, M. M., Cohen, A., Delwiche, K., Estep, K., Frostad, J. J., Kc, A., Kyu, H. H., Moradi-Lakeh, M., Ng, M., Slepak, E. L., Thomas, B. A., Wagner, J., Aasvang, G. M., Abbafati, C., Ozgoren, A. A., Abd-Allah, F., Abera, S. F., Aboyans, V., Abraham, B., Abraham, J. P., Abubakar, I., Abu-Rmeileh, N. M. E., Aburto, T. C., Achoki, T., Adelekan, A., Adofo, K., Adou, A. K., Adsuar, J. C., Afshin, A., Agardh, E. E., Al Khabouri, M. J., Al Lami,

F. H., Alam, S. S., Alasfoor, D., Albittar, M. I., Alegretti, M. A., Aleman, A. V., Alemu, Z. A., Alfonso-Cristancho, R., Alhabib, S., Ali, R., Ali, M. K., Alla, F., Allebeck, P., Allen, P. J., Alsharif, U., Alvarez, E., Alvis-Guzman, N., Amankwaa, A. A., Amare, A. T., Ameh, E. A., Ameli, O., Amini, H., Ammar, W., Anderson, B. O., Antonio, C. A. T., Anwari, P., Cunningham, S. A., Arnlöv, J., Arsenijevic, V. S. A., Artaman, A., Asghar, R. J., Assadi, R., Atkins, L. S., Atkinson, C., Avila, M. A., Awuah, B., Badawi, A., Bahit, M. C., Bakfalouni, T., Balakrishnan, K., Balalla, S., Balu, R. K., Banerjee, A., Barber, R. M., Barker-Collo, S. L., Barquera, S., Barregard, L., Barrero, L. H., Barrientos-Gutierrez, T., Basto-Abreu, A. C., Basu, A., Basu, S., Basulaiman, M. O., Ruvalcaba, C. B., Beardsley, J., Bedi, N., Bekele, T., Bell, M. L., Benjet, C., Bennett, D. A., Benzian, H., Bernabé, E., Beyene, T. J., Bhala, N., Bhalla, A., Bhutta, Z. A., Bikbov, B., Abdulhak, A. A. B., Blore, J. D., Blyth, F. M., Bohensky, M. A., Başara, B. B., Borges, G., Bornstein, N. M., Bose, D., Boufous, S., Bourne, R. R., Brainin, M., Brazinova, A., Breitborde, N. J., Brenner, H., Briggs, A. D. M., Broday, D. M., Brooks, P. M., Bruce, N. G., Brugha, T. S., Brunekreef, B., Buchbinder, R., Bui, L. N., Bukhman, G., Bulloch, A. G., Burch, M., Burney, P. G. J., Campos-Nonato, I. R., Campuzano, J. C., Cantoral, A. J., Caravanos, J., Cárdenas, R., Cardis, E., Carpenter, D. O., Caso, V., Castañeda-Orjuela, C. A., Castro, R. E., Catalá-López, F., Cavalleri, F., Çavlin, A., Chadha, V. K., Chang, J.-c., Charlson, F. J., Chen, H., Chen, W., Chen, Z., Chiang, P. P., Chimed-Ochir, O., Chowdhury, R., Christophi, C. A., Chuang, T.-W., Chugh, S. S., Cirillo, M., Claßen, T. K. D., Colistro, V., Colomar, M., Colquhoun, S. M., Contreras, A. G., Cooper, C., Cooperrider, K., Cooper, L. T., Coresh, J., Courville, K. J., Criqui, M. H., Cuevas-Nasu, L., Damsere-Derry, J., Danawi, H., Dandona, L., Dandona, R., Dargan, P. I., Davis, A., Davitoiu, D. V., Dayama, A., de Castro, E. F., De la Cruz-Góngora, V., De Leo, D., de Lima, G., Degenhardt, L., del Pozo-Cruz, B., Dellavalle, R. P., Deribe, K., Derrett, S., Jarlais, D. C. D., Dessalegn, M., deVeber, G. A., Devries, K. M., Dharmaratne, S. D., Dherani, M. K., Dicker, D., Ding, E. L., Dokova, K., Dorsey, E. R., Driscoll, T. R., Duan, L., Durrani, A. M., Ebel, B. E., Ellenbogen, R. G., Elshrek, Y. M., Endres, M., Ermakov, S. P., Erskine, H. E., Eshrati, B., Esteghamati, A., Fahimi, S., Faraon, E. J. A., Farzadfar, F., Fay, D. F. J., Feigin, V. L., Feigl, A. B., Fereshtehnejad, S.-M., Ferrari, A. J., Ferri, C. P., Flaxman, A. D., Fleming, T. D., Foigt, N., Foreman, K. J., Paleo, U. F., Franklin, R. C., Gabbe, B., Gaffikin, L., Gakidou, E., Gamkrelidze, A., Gankpé, F. G., Gansevoort, R. T., García-Guerra, F. A., Gasana, E., Geleijnse, J. M., Gessner, B. D., Gething, P., Gibney, K. B., Gillum, R. F., Ginawi, I. A. M., Giroud, M., Giussani, G., Goenka, S., Goginashvili, K., Dantes, H. G., Gona, P., de Cosio, T. G., González-Castell, D., Gotay, C. C., Goto, A., Gouda, H. N., Guerrant, R. L., Gugnani, H. C., Guillemin, F., Gunnell, D., Gupta, R., Gupta, R., Gutiérrez, R. A., Hafezi-Nejad, N., Hagan, H., Hagstromer, M., Halasa, Y. A., Hamadeh, R. R., Hammami, M., Hankey, G. J., Hao, Y., Harb, H. L., Haregu, T. N., Haro, J. M., Havmoeller, R., Hay, S. I., Hedayati, M. T., Heredia-Pi, I. B., Hernandez, L., Heuton, K. R., Heydarpour, P., Hijar, M., Hoek, H. W., Hoffman, H. J., Hornberger, J. C., Hosgood, H. D., Hoy, D. G., Hsairi, M., Hu, G., Hu, H., Huang, C., Huang, J. J., Hubbell, B. J., Huiart, L., Husseini, A., Iannarone, M. L., Iburg, K. M., Idrisov, B. T., Ikeda, N., Innos, K., Inoue, M., Islami, F., Ismayilova, S., Jacobsen, K. H., Jansen, H. A., Jarvis, D. L., Jassal, S. K., Jauregui, A., Jayaraman, S., Jeemon, P., Jensen, P. N., Jha, V., Jiang, F., Jiang, G., Jiang, Y., Jonas, J. B., Juel, K., Kan, H., Roseline, S. S. K., Karam, N. E., Karch, A., Karema, C. K., Karthikeyan, G., Kaul, A., Kawakami, N., Kazi, D. S., Kemp, A. H., Kengne, A. P., Keren, A., Khader, Y. S., Khalifa, S. E. A. H., Khan, E. A., Khang, Y.-H., Khatibzadeh, S., Khonelidze, I., Kieling, C., Kim, D., Kim, S., Kim, Y., Kimokoti, R. W., Kinfu, Y., Kinge, J. M., Kissela, B. M., Kivipelto,

M., Knibbs, L. D., Knudsen, A. K., Kokubo, Y., Kose, M. R., Kosen, S., Kraemer, A., Kravchenko, M., Krishnaswami, S., Kromhout, H., Ku, T., Defo, B. K., Bicer, B. K., Kuipers, E. J., Kulkarni, C., Kulkarni, V. S., Kumar, G. A., Kwan, G. F., Lai, T., Balaji, A. L., Lalloo, R., Lallukka, T., Lam, H., Lan, Q., Lansingh, V. C., Larson, H. J., Larsson, A., Laryea, D. O., Lavados, P. M., Lawrynowicz, A. E., Leasher, J. L., Lee, J.-T., Leigh, J., Leung, R., Levi, M., Li, Y., Li, Y., Liang, J., Liang, X., Lim, S. S., Lindsay, M. P., Lipshultz, S. E., Liu, S., Liu, Y., Lloyd, B. K., Logroscino, G., London, S. J., Lopez, N., Lortet-Tieulent, J., Lotufo, P. A., Lozano, R., Lunevicius, R., Ma, J., Ma, S., Machado, V. M. P., MacIntyre, M. F., Magis-Rodriguez, C., Mahdi, A. A., Majdan, M., Malekzadeh, R., Mangalam, S., Mapoma, C. C., Marape, M., Marcenes, W., Margolis, D. J., Margono, C., Marks, G. B., Martin, R. V., Marzan, M. B., Mashal, M. T., Masiye, F., Mason-Jones, A. J., Matsushita, K., Matzopoulos, R., Mayosi, B. M., Mazorodze, T. T., McKay, A. C., McKee, M., McLain, A., Meaney, P. A., Medina, C., Mehndiratta, M. M., Mejia-Rodriguez, F., Mekonnen, W., Melaku, Y. A., Meltzer, M., Memish, Z. A., Mendoza, W., Mensah, G. A., Meretoja, A., Mhimbira, F. A., Micha, R., Miller, T. R., Mills, E. J., Misganaw, A., Mishra, S., Ibrahim, N. M., Mohammad, K. A., Mokdad, A. H., Mola, G. L., Monasta, L., Hernandez, J. C. M., Montico, M., Moore, A. R., Morawska, L., Mori, R., Moschandreas, J., Moturi, W. N., Mozaffarian, D., Mueller, U. O., Mukaigawara, M., Mullany, E. C., Murthy, K. S., Naghavi, M., Nahas, Z., Naheed, A., Naidoo, K. S., Naldi, L., Nand, D., Nangia, V., Narayan, K. M. V., Nash, D., Neal, B., Nejjari, C., Neupane, S. P., Newton, C. R., Ngalesoni, F. N., de Dieu Ngirabega, J., Nguyen, G., Nguyen, N. T., Nieuwenhuijsen, M. J., Nisar, M. I., Nogueira, J. R., Nolla, J. M., Nolte, S., Norheim, O. F., Norman, R. E., Norrving, B., Nyakarahuka, L., Oh, I.-H., Ohkubo, T., Olusanya, B. O., Omer, S. B., Opio, J. N., Orozco, R., Pagcatipunan, R. S., Jr., Pain, A. W., Pandian, J. D., Panelo, C. I. A., Papachristou, C., Park, E.-K., Parry, C. D., Caicedo, A. J. P., Patten, S. B., Paul, V. K., Pavlin, B. I., Pearce, N., Pedraza, L. S., Pedroza, A., Stokic, L. P., Pekericli, A., Pereira, D. M., Perez-Padilla, R., Perez-Ruiz, F., Perico, N., Perry, S. A. L., Pervaiz, A., Pesudovs, K., Peterson, C. B., Petzold, M., Phillips, M. R., Phua, H. P., Plass, D., Poenaru, D., Polanczyk, G. V., Polinder, S., Pond, C. D., Pope, C. A., Pope, D., Popova, S., Pourmalek, F., Powles, J., Prabhakaran, D., Prasad, N. M., Qato, D. M., Quezada, A. D., Quistberg, D. A. A., Racapé, L., Rafay, A., Rahimi, K., Rahimi-Movaghar, V., Rahman, S. U., Raju, M., Rakovac, I., Rana, S. M., Rao, M., Razavi, H., Reddy, K. S., Refaat, A. H., Rehm, J., Remuzzi, G., Ribeiro, A. L., Riccio, P. M., Richardson, L., Riederer, A., Robinson, M., Roca, A., Rodriguez, A., Rojas-Rueda, D., Romieu, I., Ronfani, L., Room, R., Roy, N., Ruhago, G. M., Rushton, L., Sabin, N., Sacco, R. L., Saha, S., Sahathevan, R., Sahraian, M. A., Salomon, J. A., Salvo, D., Sampson, U. K., Sanabria, J. R., Sanchez, L. M., Sánchez-Pimienta, T. G., Sanchez-Riera, L., Sandar, L., Santos, I. S., Sapkota, A., Satpathy, M., Saunders, J. E., Sawhney, M., Saylan, M. I., Scarborough, P., Schmidt, J. C., Schneider, I. J. C., Schöttker, B., Schwebel, D. C., Scott, J. G., Seedat, S., Sepanlou, S. G., Serdar, B., Servan-Mori, E. E., Shaddick, G., Shahraz, S., Levy, T. S., Shangguan, S., She, J., Sheikhbahaei, S., Shibuya, K., Shin, H. H., Shinohara, Y., Shiri, R., Shishani, K., Shiue, I., Sigfusdottir, I. D., Silberberg, D. H., Simard, E. P., Sindi, S., Singh, A., Singh, G. M., Singh, J. A., Skirbekk, V., Sliwa, K., Soljak, M., Soneji, S., Søreide, K., Soshnikov, S., Sposato, L. A., Sreeramareddy, C. T., Stapelberg, N. J. C., Stathopoulou, V., Steckling, N., Stein, D. J., Stein, M. B., Stephens, N., Stöckl, H., Straif, K., Stroumpoulis, K., Sturua, L., Sunguya, B. F., Swaminathan, S., Swaroop, M., Sykes, B. L., Tabb, K. M., Takahashi, K., Talongwa, R. T., Tandon, N., Tanne, D., Tanner, M., Tavakkoli, M., Te Ao, B. J., Teixeira, C. M., Téllez Rojo, M. M., Terkawi, A. S., Texcalac-Sangrador, J. L., Thackway, S. V., Thomson, B., Thorne-Lyman,

A. L., Thrift, A. G., Thurston, G. D., Tillmann, T., Tobollik, M., Tonelli, M., Topouzis, F., Towbin, J. A., Toyoshima, H., Traebert, J., Tran, B. X., Trasande, L., Trillini, M., Trujillo, U., Dimbuene, Z. T., Tsilimbaris, M., Tuzcu, E. M., Uchendu, U. S., Ukwaja, K. N., Uzun, S. B., van de Vijver, S., Van Dingenen, R., van Gool, C. H., van Os, J., Varakin, Y. Y., Vasankari, T. J., Vasconcelos, A. M. N., Vavilala, M. S., Veerman, L. J., Velasquez-Melendez, G., Venketasubramanian, N., Vijayakumar, L., Villalpando, S., Violante, F. S., Vlassov, V. V., Vollset, S. E., Wagner, G. R., Waller, S. G., Wallin, M. T., Wan, X., Wang, H., Wang, J., Wang, L., Wang, W., Wang, Y., Warouw, T. S., Watts, C. H., Weichenthal, S., Weiderpass, E., Weintraub, R. G., Werdecker, A., Wessells, K. R., Westerman, R., Whiteford, H. A., Wilkinson, J. D., Williams, H. C., Williams, T. N., Woldeyohannes, S. M., Wolfe, C. D. A., Wong, J. Q., Woolf, A. D., Wright, J. L., Wurtz, B., Xu, G., Yan, L. L., Yang, G., Yano, Y., Ye, P., Yenesew, M., Yentür, G. K., Yip, P., Yonemoto, N., Yoon, S.-J., Younis, M. Z., Younoussi, Z., Yu, C., Zaki, M. E., Zhao, Y., Zheng, Y., Zhou, M., Zhu, J., Zhu, S., Zou, X., Zunt, J. R., Lopez, A. D., Vos, T., and Murray, C. J.: Global, regional, and national comparative risk assessment of 79 behavioural, environmental and occupational, and metabolic risks or clusters of risks in 188 countries, 1990–2013: a systematic analysis for the Global Burden of Disease Study 2013, The Lancet, 386, 2287-2323, 10.1016/s0140-6736(15)00128-2, 2015.

Hoesly, R. M., Smith, S. J., Feng, L., Klimont, Z., Janssens-Maenhout, G., Pitkanen, T., Seibert, J. J., Vu, L., Andres, R. J., Bolt, R. M., Bond, T. C., Dawidowski, L., Kholod, N., Kurokawa, J. I., Li, M., Liu, L., Lu, Z., Moura, M. C. P., O'Rourke, P. R., and Zhang, Q.: Historical (1750–2014) anthropogenic emissions of reactive gases and aerosols from the Community Emission Data System (CEDS), Geosci. Model Dev. Discuss., 2017, 1-41, 10.5194/gmd-2017-43, 2017.

Höglund-Isaksson, L. and Mechler, R.: The GAINS model for greenhouse gases-version 1.0: Methane (CH4), IIASA Interim Report, International Institute for Applied Systems Analysis, Laxenburg, Austria. [online] Available from: http://pure.iiasa.ac.at/7784/ (Accessed 10 January 2017), 2005.

Liang, C. K., West, J. J., Silva, R. A., Bian, H., Chin, M., Dentener, F. J., Davila, Y., Emmons, L., Folberth, G., Flemming, J., Henze, D., Im, U., Jonson, J. E., Kucsera, T., Keating, T. J., Lund, M. T., Lenzen, A., Lin, M., Pierce, R. B., Park, R. J., Pan, X., Sekiya, T., Sudo, K., and Takemura, T.: HTAP2 multi-model estimates of premature human mortality due to intercontinental transport of air pollution, Atmos. Chem. Phys. Discuss., 2018, 1-35, 10.5194/acp-2017-1221, 2018.

Janssens-Maenhout, G., Crippa, M., Guizzardi, D., Dentener, F., Muntean, M., Pouliot, G., Keating, T., Zhang, Q., Kurokawa, J., and Wankmüller, R.: HTAP_v2. 2: a mosaic of regional and global emission grid maps for 2008 and 2010 to study hemispheric transport of air pollution, Atmospheric Chemistry and Physics, 15, 11411-11432, 2015.

Klimont, Z., Kupiainen, K., Heyes, C., Purohit, P., Cofala, J., Rafaj, P., Borken-Kleefeld, J., and Schöpp, W.: Global anthropogenic emissions of particulate matter including black carbon, Atmos. Chem. Phys., 17, 8681-8723, 10.5194/acp-17-8681-2017, 2017.

Riahi, K., Grübler, A. and Nakicenovic, N.: Scenarios of long-term socio-economic and environmental development under climate stabilization, Technol. Forecast. Soc. Change, 74(7), 887–935, doi:10.1016/j.techfore.2006.05.026, 2007.

Russ, P., Wiesenthal, T., Van Regemorter, D. and Ciscar, J.: Global Climate Policy Scenarios for 2030 and beyond. Analysis of Greenhouse Gas Emission Reduction Pathway Scenarios with the POLES and GEM-E3 models., European Commission, Joint Research Centre, IPTS, Seville, Spain., 2007.

Smith, S. J., van Aardenne, J., Klimont, Z., Andres, R. J., Volke, A., and Delgado Arias, S.: Anthropogenic sulfur dioxide emissions: 1850–2005, Atmos. Chem. Phys., 11, 1101-1116, 10.5194/acp-11-1101-2011, 2011.

Van Aardenne, J., Dentener, F., Van Dingenen, R., Maenhout, G., Marmer, E., Vignati, E., Russ, P., Szabo, L. and Raes, F.: Climate and air quality impacts of combined climate change and air pollution policy scenarios, JRC Scientific and Technical Reports, Ispra, Italy, 2007.

Van Dingenen, R., Dentener, F., Crippa, M., Leitao-Alexandre, J., Marmer, E., Rao, S., Solazzo, E., and Valentini, L.: TM5-FASST: a global atmospheric source-receptor model for rapid impact analysis of emission changes on air quality and short-lived climate pollutants, 2018, submitted.

van Donkelaar, A., Martin, R. V., Brauer, M., Kahn, R., Levy, R., Verduzco, C. and Villeneuve, P. J.: Global Estimates of Ambient Fine Particulate Matter Concentrations from Satellite-Based Aerosol Optical Depth: Development and Application, Environ. Health Perspect., 118(6), 847–855, doi:10.1289/ehp.0901623, 2010.

van Donkelaar, A., Martin, R. V., Brauer, M. and Boys, B. L.: Use of Satellite Observations for Long-Term Exposure Assessment of Global Concentrations of Fine Particulate Matter, Environ. Health Perspect., doi:10.1289/ehp.1408646, 2014.

van Vuuren, D. P., Elzen, M. G. J. den, Lucas, P. L., Eickhout, B., Strengers, B. J., Ruijven, B. van, Wonink, S. and Houdt, R. van: Stabilizing greenhouse gas concentrations at low levels: an assessment of reduction strategies and costs, Clim. Change, 81(2), 119–159, doi:10.1007/s10584-006-9172-9, 2007.

---

## Author Comment (AC5) · 19 Feb 2018

The comment was uploaded in the form of a supplement:
https://www.atmos-chem-phys-discuss.net/acp-2017-779/acp-2017-779-AC5-
supplement.pdf

---

## Author Comment (AC6) · 19 Feb 2018

The comment was uploaded in the form of a supplement:
https://www.atmos-chem-phys-discuss.net/acp-2017-779/acp-2017-779-AC6-supplement.pdf

---

## Author Comment (AC7) · 19 Feb 2018

[revised manuscript text omitted]
. Similar results are also found comparing regional averages of urban stations from WHO against the FASST population weighted average of grid cells. The TM5-FASST modeled PM$_{2.5}$ concentrations have been compared to satellite products which are based on aerosol optical depth measurements together with chemical transport model information to retrieve from the total column the information of PM concentrations in the lowest layer of the atmosphere (Boys et al., 2014; van Donkelaar et al., 2010, 2014). The regional comparison shows consistent results with the ground based measurements comparison (e.g. good agreement for EU and USA within 10% 
[revised manuscript text omitted]

---

## Author Comment (AC8) · 19 Feb 2018

The comment was uploaded in the form of a supplement:
https://www.atmos-chem-phys-discuss.net/acp-2017-779/acp-2017-779-AC8-supplement.pdf

---

## Referee Report (RR1)

The authors coupled the HTAP_v2.2 global air pollutant emission inventory with the global source receptor model TM5-FASST to evaluate the relative contribution of the major anthropogenic emission sources to air quality and health in 2010. As I noted in my previous review, I find that what the paper is trying to do is important. However, I still find that the objective of the paper is unclear throughout the paper and I am not sure if this paper should be stand-alone or should be combined together with the Van Dingenen et al. (2018) paper that is currently under review for ACP. Most importantly, I do not understand the rationale behind quantifying health impacts from sectorial emissions, given that the uncertainty is so high.

First, the biggest problem I have with this paper is that there are significant underestimations of $PM_{2.5}$ concentrations in many countries and to me, the linearity estimation for $PM_{2.5}$ is not satisfactory. I am not convinced that there is new science in the paper and as one of the reviewers was suggesting, maybe this paper, combined with the Van Dingenen et al. (2018) paper should probably be moved to GMD to discuss potential of the new tool for assessing air quality and health impacts.

Second, the objective needs to be better defined. As for the two objectives of the study, there are two sentences in the manuscript:
P. 2, l. 46 "The objective of this study is to evaluate the relevance of uncertainties in regional sectorial emission inventories, and their propagation in modelled $PM_{2.5}$ concentrations and associated impacts on health."
P. 3, l. 15 "A second objective of this analysis is to evaluate the importance of emission uncertainties at sector and regional level on $PM_{2.5}$, to better inform local, regional and hemispheric air quality policy makers on the potential impacts of sectors with larger uncertainties or regions."
The two are very similar and I am not sure if the second objective is necessary. On p. 4, l. 12, the aim of this work is explained "to address the uncertainty of sector specific emissions from this inventory in a quantitative way as well as the differences we observe from one region to the other, based on the uncertainty of activity data and emission factors." Furthermore, later in the text on p. 13 l. 38, the authors state, "[i]n our work we only evaluate how the uncertainty of emission inventories influences the health impact estimates focusing on the interregional aspects and not all the other sources of uncertainties." The authors should be consistent in what the objective and the aim of this work is throughout the paper.

Third, the paper should have all the methodologies related to the objective in the paper. For example, if the objective of this paper is indeed on quantifying health impacts, I think the premature mortality calculation methodology should move from the Van Dingenen et al. (2018) to this paper and the crop damage should be taken out from this paper.

Fourth, the writing could be improved, as it is often difficult to follow, as described in minor comments below.

Minor comments:
1. P. 1, l. 29 Not sure what the authors mean by "improve emission inventories knowledge and air quality"
2. P. 2, l. 9 Not sure what the authors mean by "improve globally air quality and possibly human health"

3. P. 2, l. 13-20 I am unsure what the authors mean in the two sentences.
4. P. 2, l. 35 414.000 → 414,000 or 414 thousand
5. P. 4, l. 28 "can be also applied also" → delete the second "also"
6. P. 5 l. 19 "now day much more" → "now much more?"
7. P. 6 l. 10 Not sure what the authors mean by "24OECD90 countries"
8. P. 14 l. 25 How did the authors come up with a threshold value of 5.8 $\mu g/m^3$?

---

## Referee Report (RR2)

This is my third review of this paper. I understand that the Van Dingenen et al. (2018) paper is already published but I still wonder if there is a new science in this paper by itself, as I had questioned in the previous reviews. If the key objective of the paper is "to evaluate the relevance of uncertainties in regional sectorial emissions inventories and their propagation in modelled $PM_{2.5}$ concentrations and associated impacts on health," I feel that more needs to be done in the paper. For example, I find it troubling that there is a single number listed as a fraction of extra-regional pollution contribution per country in section 3.1 (Hungary 75%, etc.). The same is true for the sectorial contributions to $PM_{2.5}$ concentrations in section 3.2 (30% by shipping emissions in the Mediterranean). Probably most troubling is the health effect quantification (e.g., 32.4% of total mortality related with agriculture). When there are large uncertainties as the authors have already acknowledged, I find it necessary to clearly describe these ranges in each step and also in tables and figures as well. Considering the objective of the paper, I do not see the point of sections 3.1-3.3. It is probably better to expand section 3.4 that discusses the impact of uncertainties from emissions.

There are also quite a few editorial issues that need to be addressed. I cannot point them all but below are a few:

1. The explanation of $PM_{2.5}$ appears on l. 13 on p. 2 when $PM_{2.5}$ is already mentioned on l. 5.
2. I believe it should be written as "improve global air quality" instead on l. 9, p. 2.
3. The first "and" should be deleted on l. 33, p. 3.
4. $CH_4$ is not mentioned in l. 28-29 on p. 5 but I believe HTAP_v2.2 includes that?
5. I think the second "the" should be taken out from l. 40, p. 5
6. Chili → Chile on l. 15, p. 11

---

## Author Response (AR2)

The authors are grateful to the Referee for these additional comments that helped in improving the manuscript. The paper by Van Dingenen et al. (submitted, 2018) about the TM5-FASST methodology is now accepted for final publication, therefore we hope that with the additional information provided below as well as the changes done in the manuscript will help in solving all the concerns.

The authors coupled the HTAP_v2.2 global air pollutant emission inventory with the global source receptor model TM5-FASST to evaluate the relative contribution of the major anthropogenic emission sources to air quality and health in 2010. As I noted in my previous review, I find that what the paper is trying to do is important. However, I still find that the objective of the paper is unclear throughout the paper and I am not sure if this paper should be stand-alone or should be combined together with the Van Dingenen et al. (2018) paper that is currently under review for ACP. Most importantly, I do not understand the rationale behind quantifying health impacts from sectorial emissions, given that the uncertainty is so high.

First, the biggest problem I have with this paper is that there are significant underestimations of PM2.5 concentrations in many countries and to me, the linearity estimation for PM2.5 is not satisfactory. I am not convinced that there is new science in the paper and as one of the reviewers was suggesting, maybe this paper, combined with the Van Dingenen et al. (2018) paper should probably be moved to GMD to discuss potential of the new tool for assessing air quality and health impacts.

The paper by van Dingenen et al. (2018) is accepted for final publication in ACP. In our manuscript we clarify the role of our paper being an application of the TM5-FASST methodology.

"This work is an application of the TM5-FASST model, which is extensively documented in a companion publication in this special issue. Van Dingenen et al., (2018) provide an extensive evaluation of the model, model assumptions and performance with regard to linearity and additivity of concentration response to different size of emission perturbations and future emission scenarios."

Moreover, in the supplementary material we provide additional information on the assumptions of linearity as reported below:

**S1.2 – Sector and source region attribution using the TM5-FASST source-receptor relationships**

**S1.2.1 - Attribution by sector**

The TM5-FASST methodology uses a local perturbation approach in the vicinity of a reference simulation, where the total concentration of component (or metric) $j$ in receptor region $y$, resulting from emissions of all $n_i$ precursors $i$ in all $n_x$ source regions $x$, is obtained as a

perturbation on the base-simulation concentration (Van Dingenen et al., 2018). Hence, the $PM_{2.5}$ concentration in region $y$ for an emission scenario different from the reference scenario is obtained as:

$$PM(y) = PM_{ref}(y) + \Delta PM(y) \tag{1}$$

The perturbation term $\Delta PM(y)$ is obtained from the linear scaling of the difference between scenario and reference emission (i.e. the emission perturbation):

$$\Delta PM(y) = \sum_{j=1}^{n_j} \sum_{k=1}^{n_x} \sum_{i=1}^{n_i} A_{ij}[x_k, y] \cdot \left[ E_i(x_k) - E_{i,ref}(x_k) \right] \tag{2}$$

where the summation runs over $n_i$ precursor species, $n_j$ $PM_{2.5}$ components and $n_x$ source regions, and $A_{ij}[x_k, y]$ is the source-receptor coefficient, expressing the emission-concentration response sensitivity in the vicinity of the reference conditions, evaluated from a 20% emission perturbation (see Van Dingenen et al., 2018):

$$A_{ij}[x, y] = \frac{\Delta PM_{ref}^{j}(y)}{\Delta E_{i,ref}(x)} \tag{3}$$

with $\Delta E_{i,ref}(x) = 0.2 E_{i,ref}(x)$ and $\Delta PM_{ref}^{j}(y)$ the corresponding $PM_{2.5}$ component $j$ response.

Eq. (2) can also be applied to attribute individual sector contributions to the pollutant concentration by setting the "emission perturbation" equal to the emission contribution of a single sector. The $PM_{2.5}$ contribution from the single sector S equals

$$\Delta PM'_S(y) = \sum_{j=1}^{n_j} \sum_{k=1}^{n_x} \sum_{i=1}^{n_i} A_{ij}[x_k, y] \cdot \left[ E_{S,i}(x_k) \right] \tag{4}$$

Having obtained the marginal $PM_{2.5}$ contributions from the individual sectors, the total $PM_{2.5}$ can be re-composed as the sum from all $n_S$ sectors S:

$$PM'(y) = \sum_{s=1}^{n_S} \Delta PM'_s(y) \tag{5}$$

However, due to non-linearities in emission-concentration responses, the sum of all individual sector contributions may not exactly match the total $PM_{2.5}$ obtained from Eqs. (1) and (2) where we write $E_i(x_k)$ as the sum of the emissions by sector:

$$PM(y) = PM_{base}(y) + \sum_{j=1}^{n_j} \sum_{k=1}^{n_x} \sum_{i=1}^{n_i} A_{ij}[x_k, y] \cdot \left[ \sum_{s=1}^{n_S} E_{s,i}(x_k) - E_{i,ref}(x_k) \right] \tag{6}$$

$PM'(y)$ from Eq. 5 and $PM(y)$ from Eq. 6 are equivalent if

$$PM_{ref}(y) = \sum_{j=1}^{n_j} \sum_{k=1}^{n_x} \sum_{i=1}^{n_i} A_{ij}[x_k, y] \cdot E_{i,ref}(x_k) \tag{7}$$

Using Eq. 3 this is equivalent to the condition that

$$PM_{ref}(y) = \sum_{j=1}^{n_j} \sum_{k=1}^{n_x} \sum_{i=1}^{n_i} A_{ij}[x_k, y] \frac{\Delta PM(y)}{0.2 E_{i,\text{ref}}(x_k)} E_{i,ref}(x_k) \tag{8}$$

or

$$PM_{ref}(y) = \sum_{j=1}^{n_j} \sum_{k=1}^{n_x} \sum_{i=1}^{n_i} 5.\Delta PM(y) \tag{9}$$

In other words, total $PM_{2.5}$ will be correctly reproduced as the sum of the individual sector contributions if and only if the $PM_{2.5}$ base concentration can be approached by 5 times the 20% perturbation response, implying a perfectly linear emission-concentration response for all precursors. Figure A1.1 shows the correspondence between regionally aggregated $\sum_{j=1}^{n_j} \sum_{k=1}^{n_x} \sum_{i=1}^{n_i} 5.\Delta PM$ and $PM_{ref}$. The agreement is satisfactory although not perfect. In order to restore the closure between the total $PM_{2.5}$ and the sum of the sectors, we therefore rescale the sector contributions such that their sum corresponds to the total $PM_{2.5}$ obtained from the local perturbation calculation, i.e. we use the relative contribution by sector resulting from Eq. 5 and apply them onto the total $PM_{2.5}$ obtained from Eq. 6.

$$\Delta PM_S(y) = \frac{\Delta PM\prime_S(y)}{\sum_{s=1}^{n_S} \Delta PM\prime_S(y)} PM(y) \tag{10}$$

**S1.2.2 Attribution by source region**

The marginal contribution of an individual source regions ($x$) to the total $PM_{2.5}$ concentration in a given receptor region ($y$) is obtained (via Eq. 2) from

$$\Delta PM\prime_x(y) = \sum_{j=1}^{n_j} \sum_{i=1}^{n_i} A_{ij}[x, y] \cdot E_i(x) \tag{11}$$

Similar as for the sector break-down, the emission perturbation has been replaced by an extrapolation of the SR coefficient over the total emission magnitude in a given source region, and non-linearities may lead to non-closure between the sum of all $\Delta PM\prime_x(y)$ and total $PM_{2.5}$ obtained from the local perturbation as in Eqs. (1) and (2). In order to restore the closure we apply the same scaling procedure as in Eq. 10:

$$\Delta PM_x(y) = \frac{\Delta PM\prime_x(y)}{\sum_{k=1}^{n_k} \Delta PM\prime_{x_k}(y)} PM(y) \tag{12}$$

[Figure]

**Figure S1.2 - Scatter plot of regionally aggregated PM$_{2.5}$ concentrations. Y-axis: FASST linearized extrapolation of a 20% emission perturbation towards 100%, versus the full TM5 computation, for the FASST reference emission scenario (RCP year 2000, se Van Dingenen et al., 2018). The Figure evaluates the validity of Eq. 9.**

Second, the objective needs to be better defined. As for the two objectives of the study, there are two sentences in the manuscript: P. 2, l. 46 "The objective of this study is to evaluate the relevance of uncertainties in regional sectorial emission inventories, and their propagation in modelled PM2.5 concentrations and associated impacts on health."

We thank the Reviewer for the comment regarding the objectives of the paper and we clarified the objectives of the paper as described below.

We kept the sentence in the introduction (reported below) since it represents the key objective of the uncertainty analysis.

"The objective of this study is to evaluate the relevance of uncertainties in regional sectorial emission inventories, and their propagation in modelled PM2.5 concentrations and associated impacts on health."

P. 3, l. 15 "A second objective of this analysis is to evaluate the importance of emission uncertainties at sector and regional level on PM2.5, to better inform local, regional and hemispheric air quality policy makers on the potential impacts of sectors with larger uncertainties or regions." The two are very similar and I am not sure if the second objective is necessary.

We rephrased as following:

Based on our analysis on the importance of emission uncertainties at sector and regional level on PM$_{2.5}$, we aim at informing local, regional and hemispheric air quality policy makers on the potential impacts of sectors with larger uncertainties (e.g. residential and agriculture) or regions (e.g. developing and emerging countries).

On p. 4, l. 12, the aim of this work is explained "to address the uncertainty of sector specific emissions from this inventory in a quantitative way as well as the differences we observe from one region to the other, based on the uncertainty of activity data and emission factors." Furthermore, later in the text on p. 13 l. 38, the authors state, "[i]n our work we only evaluate how the uncertainty of emission inventories influences the health impact estimates focusing on the interregional aspects and not all the other sources of uncertainties." The authors should be consistent in what the objective and the aim of this work is throughout the paper.

We rephrased as following:

In the following, we will address the uncertainty of sector specific emissions from this inventory in a quantitative way as well as the differences we observe from one region to the other, based on the uncertainty of activity data and emission factors.

Third, the paper should have all the methodologies related to the objective in the paper. For example, if the objective of this paper is indeed on quantifying health impacts, I think the premature mortality calculation methodology should move from the Van Dingenen et al. (2018) to this paper and the crop damage should be taken out from this paper.

We agree with the Reviewer about the need of knowing the details of the methodology applied to estimate the health effects in TM5-FASST, as well as other methodological assumptions. In our manuscript we have summarized the key features the reader need to know in order to understand the results discussed in out manuscript. However, we cannot report all the details about the TM5-FASST methodologies which are extensively described both in the main text and in the supplementary material of the work by Van Dingenen et al. (2018). To help the reader in linking our manuscript with the paper by Van Dingenen et al. (2018), we added the information about the sections of the paper by Van Dingenen et al. (2018) where to find these methodological information. For example we now report that:

As described in Sect 2.5 and S5 of the paper by Van Dingenen et al. (2018), the mortality estimation in TM5-FASST is based on the integrated exposure-response functions defined by Burnett et al. (2014).

Fourth, the writing could be improved, as it is often difficult to follow, as described in minor comments below.

Minor comments:

1. P. 1, l. 29 Not sure what the authors mean by "improve emission inventories knowledge and air quality"

We corrected as following: "improve emission inventories knowledge and air quality modeling".

2. P. 2, l. 9 Not sure what the authors mean by "improve globally air quality and possibly human health"
   The sentence now reads:
   Local, regional and international coordination is therefore needed to define air pollution policies to improve globally air quality and possibly human health.

3. P. 2, l. 13-20 I am unsure what the authors mean in the two sentences.
   This paragraph aims at giving the context of air quality issues which are not only happing locally but also at regional and global scale. Then we focus on a short description of particulate matter composition and formation, being the compound we look at in this publication.

   "Local, regional and international coordination is therefore needed to define air pollution policies to improve globally air quality and possibly human health. The CLRTAP's Task Force on Hemispheric Transport of Air Pollution looks at the long-range transport of air pollutants in the Northern Hemisphere aiming to identify promising mitigation measures to reduce background pollution levels and its contribution to pollution in rural as well as urban regions. Although primary PM2.5 (particulate matter with a diameter less than 2.5 μm) and intermediately lived (days-to-weeks) precursor gases can travel over long distances, the transboundary components of anthropogenic PM are mainly associated with secondary aerosols which are formed in the atmosphere through complex chemical reactions and gas-to-aerosol transformation, transport and removal processes, of gaseous precursors transported out of source regions (Maas and Grennfelt, 2016). However, the most extreme episodes of exposure often occur under extended periods of low wind speeds and atmospheric stability, favoring formation of secondary aerosols close to the source regions. Secondary aerosol from anthropogenic sources consists of both inorganic -mainly ammonium nitrate and ammonium sulfate and ammonium bisulfate and associated water, formed from emissions of sulphur dioxide (SO2), nitrogen oxides (NOx) and ammonia (NH3), and organic compounds involving thousands of compounds and often poorly known reactions (Hallquist et al., 2009)."

4. P. 2, l. 35 414.000 à 414,000 or 414 thousand
   Changed to 414 thousand
5. P. 4, l. 28 "can be also applied also" à delete the second "also"

Change done as requested

6. P. 5 l. 19 "now day much more" à "now much more?"
   Change done as requested

7. P. 6 l. 10 Not sure what the authors mean by "24OECD90 countries"

A footnote has been inserted to identify the OECD countries in 1990:

OECD countries in 1990: Australia, Austria, Belgium, Canada, Denmark, Finland, France, Germany, Greece, Iceland, Ireland, Italy, Japan, Luxembourg, Netherlands, New Zealand, Norway, Portugal, Spain, Sweden, Switzerland, Turkey, United Kingdom, United States.

8. P. 14 l. 25 How did the authors come up with a threshold value of 5.8 µg/m3?

The threshold value of 5.8 µg/m3 comes from literature (Anenberg et al., 2010) and it is fully described in the work by Van Dingenen et al. (2018).

References

Anenberg, S. C., Horowitz, L. W., Tong, D. Q. and West, J. J.: An estimate of the global burden of anthropogenic ozone and fine particulate matter on premature human mortality using atmospheric modeling, Environ. Health Perspect., 118(9), 1189–1195, doi:10.1289/ehp.0901220, 2010.

Van Dingenen, R., Dentener, F., Crippa, M., Leitao, J., Marmer, E., Rao, S., Solazzo, E., and Valentini, L.: TM5-FASST: a global atmospheric source-receptor model for rapid impact analysis of emission changes on air quality and short-lived climate pollutants, Atmos. Chem. Phys. Discuss., 2018, 1-55, 10.5194/acp-2018-112, 2018.

---

## Author Response (AR3)

**Reviewer 1:**

This is my third review of this paper. I understand that the Van Dingenen et al. (2018) paper is already published but I still wonder if there is a new science in this paper by itself, as I had questioned in the previous reviews.

We thank Reviewer 1 for the insightful comments. Indeed the objectives and novelties of this study are beyond what was suggested by the original title of our publication. In order to highlight all the objectives of this study, we have rephrased the title of this paper:

"Contribution and uncertainty of sectorial and regional emissions to regional and global  $\rm PM_{2.5}$  health impacts"

The objectives and novelties of this study, not covered by the Van Dingenen et al. paper are now listed in the introduction, and include the evaluation of i) the relative contribution of anthropogenic emission sources to PM2.5 concentrations at global scale, ii) identification of the emission sectors and emission regions for which pollution reduction measures would lead to the largest improvement on air quality and iii) the relevance of uncertainties in regional sectorial emission inventories (power generation, industry, ground transport, residential, agriculture and international shipping), and their propagation in modelled PM2.5 concentrations and associated impacts on health.

If the key objective of the paper is "to evaluate the relevance of uncertainties in regional sectorial emissions inventories and their propagation in modelled PM2.5 concentrations and associated impacts on health," I feel that more needs to be done in the paper.

The Reviewer's comments stem from the expectation that each single subsection is providing uncertainty information. We make it clear now that the earlier sections (3.1, 3.2 and 3.3) rather focus on providing 'central' estimates of regional, sectorial and gridded contributions, whereas Section 3.4 is providing the corresponding uncertainty estimates. We have therefore added the following sentence at the beginning of section 3:

"In this section, we first provide 'central' estimates of regional (Sect. 3.1), sectorial (Sect. 3.2) and gridded (Sect. 3.3) contributions, whereas the corresponding uncertainty estimates are discussed from Sect. 3.4 onward."

For example, I find it troubling that there is a single number listed as a fraction of extraregional pollution contribution per country in section 3.1 (Hungary 75%, etc.). The same is true for the sectorial contributions to PM2.5 concentrations in section 3.2 (30% by shipping emissions in the Mediterranean). Probably most troubling is the health effect quantification (e.g., 32.4% of total mortality related with agriculture). When there are large uncertainties as the authors have already acknowledged, I find it necessary to clearly describe these ranges in each step and also in tables and figures as well.

The uncertainties mentioned by the Reviewer are provided in Table 2, Fig.5 and Table 4 where key metrics and the corresponding range of uncertainty by region are reported. In particular Table 2 reports annual average PM2.5 concentrations and the corresponding uncertainty range for each TM5-FASST region, Table 4 contains the numbers of premature deaths/year due to anthropogenic PM2.5 air pollution in world regions and corresponding uncertainty range, and Figure 5 graphically reports within-region and extra-regional anthropogenic PM2.5 concentrations and their uncertainty for all TM5-FASST regions. We tried not to repeat the information already provided in all figures and tables also in the text to avoid redundancy and ensure readability. As already mentioned, our discussion starts with describing the 'central' source-receptor estimates and from there on calculating the uncertainties. To follow the Reviewer's suggestion, when information about uncertainty was missing we modified the manuscript accordingly, in particular in the conclusion, to highlight the range of our estimates due to the uncertainty of the emissions.

Considering the objective of the paper, I do not see the point of sections 3.1-3.3. It is probably better to expand section 3.4 that discusses the impact of uncertainties from emissions.

As we now better explain in the introduction, uncertainty of emission inventories is only one objective of this work, since we aim also at addressing the sector specific regional contribution estimates to PM2.5 concentrations, as presented in sections 3.1-3.3 are needed. Therefore, we have modified the text of the introduction accordingly: "The objectives and novelties of this study are the evaluation of i) the relative contribution of anthropogenic emission sources to PM2.5 concentrations at global scale, ii) the emission sectors and emission regions in which pollution reduction measures would lead to the largest improvement on the overall air quality and iii) the relevance of uncertainties in regional sectorial emission inventories (power generation, industry, ground transport, residential, agriculture and international shipping), and their propagation in modelled PM2.5 concentrations and associated impacts on health."

There are also quite a few editorial issues that need to be addressed. I cannot point them all but below are a few:

1. The explanation of PM2.5 appears on l. 13 on p. 2 when PM2.5 is already mentioned on l. 5.

The explanation of PM2.5 has now been introduced at its earliest appearance.

2. I believe it should be written as "improve global air quality" instead on 1. 9, p. 2.

The change has been done accordingly with the Reviewer's comment.

3. The first "and" should be deleted on l. 33, p. 3.

The change has been done accordingly with the Reviewer's comment.

4. CH4 is not mentioned in l. 28-29 on p. 5 but I believe HTAP\_v2.2 includes that?

The HTAP\_v2.2 inventory does not include methane emissions, as documented by Janssens-Maenhout et al. (2015) and at the following link: http://edgar.jrc.ec.europa.eu/htap\_v2/index.php

5. I think the second "the" should be taken out from 1. 40, p. 5

We disagree with the reviewer's comment since the sentence reads as following:

"....a set of emission perturbation scenarios has been created by subtracting from the reference dataset the emissions of each sector."

6. Chili: Chile on l. 15, p. 11

The change has been done accordingly with the Reviewer's comment.

**References:**

Janssens-Maenhout, G., Crippa, M., Guizzardi, D., Dentener, F., Muntean, M., Pouliot, G., Keating, T., Zhang, Q., Kurokawa, J., and Wankmüller, R.: HTAP\_v2. 2: a mosaic of regional and global emission grid maps for 2008 and 2010 to study hemispheric transport of air pollution, Atmospheric Chemistry and Physics, 15, 11411-11432, 2015.

**Reviewer 2**

The authors have done a good job responding to reviewer comments. They have brought in a lot more material from their companion paper on the model description and evaluation, which address reviewer concerns in those regards. I have a few remaining comments, described below, which amount to only minor changes to manuscript text and this minor revisions, after which point the paper will be suitable for publication in ACP.

**We thank Reviewer 2 for the insightful comments.**

**Comments:**

Presentation of the main equations still comes across as a bit folksy. The authors refer to it as "perturbation approach" — their quotes, not mine, sometimes double and sometimes single — but more rigorously I think as scientists they can more specifically refer to this as a first order approximation that includes the first (linear) term of a Taylor expansion of PM as a function of emissions. Without the remaining higher order terms the expression is approximate. Further, the authors state "So the equal sign is correct, although this equation represents an approximation", which is an oxymoron. The authors confuse discussion of a computational equation implemented in their model (which may well be approximate) and noting whether or not that equation is exact (with an equals sign) or an expression of an approximation (with an approximation sign). In this case it is the latter, and the equation on paper needs to be fixed to show this. That being said, I appreciate the additional discussion added to the main text and the SI regarding the equations used for estimating PM responses owing to emissions perturbations, which have indeed helped make the manuscript stronger and more complete.

We have implemented in Sect. 2.1 the changes required by the Reviewer as following:

Specifically, the reduced-form model TM5-FASST is computing the concentration resulting from an arbitrary precursor emission strength  $E_i$  using a first order perturbation approach, i.e. for each PM component *j*, the change in concentration  $dPM_j$  resulting from a change in emission strength  $E_i(x)$  of precursor *i* in source region *x*, relative to a reference emission  $E_{i,ref}(x)$ , is approximated by the first linear term of a Taylor expansion of PM as a function of emissions:

$$dPM_{i}(y) \cong A_{ij}[x, y] [E_{i}(x) - E_{i,ref}(x)]$$
(Eq. 1)

Where

$$A_{ij}[x, y] = \frac{\Delta C_j(y)}{\Delta E_i(x)} \text{ with } \Delta E_i(x) = 0.2E_{i,ref}(x)$$

(Eq. 2)

 $A_{ij}[x, y]$  is a set of independently computed source-receptor matrices, expressing the linearized emission-concentration response between each relevant precursor (*i*) emission and PM component *j* concentration, for each pair of source (*x*) and receptor (*y*) regions (Van Dingenen et al., 2018).

In Sect. S1.2 we explain in detail how Eq. 1 can be also applied for evaluating the attribution by sector as well as by source region, based on the work by Van Dingenen et al. (2018).

The additional content on model accuracy, again drawing from the companion paper, is now more detailed, which is appreciated.

We are grateful to the Reviewer for agreeing with the changes we performed in the manuscript based on his first comments.

The response regarding other sources of uncertainty — I appreciate the added discussion regarding model errors from Solazzo 2018. However my comments were with regards to uncertainties in the concentration-response functions, which are typically the only ones considered.

**Fine with the Reviewer's comment.**

Further, the response of the authors in this regard could still be much stronger. The title of this paper includes "uncertainty analysis". However, the abstract only notes that the

uncertainty analysis for health impacts was performed (last sentence), and does not even state the results. This is a big loss for this work — the authors should do a better job of capturing these quite interesting results (up to 1 million premature deaths uncertainty associated with emissions uncertainties?) in the abstract and conclusions, specifically in comparison to the level of uncertainty normally associated with these types of studies.

The uncertainty analysis performed within this paper aims at rising the awareness on how the uncertainty of emission inventories affects PM concentrations and its impacts on human health. We agree with the suggestion of the Reviewer in stressing our findings on emission uncertainty propagation to impacts both in the abstract and conclusions as discussed in the following. In addition we rephrased the title to satisfy the additional requests of Reviewer 1: "Contribution and uncertainty of sectorial and regional emissions to regional and global PM2.5 health impacts".

The only other statement regarding uncertainty in the abstract (second to last sentence) is rather obvious and could be omitted, unless it is going to be quantified.

In the context of uncertainty, we rephrased the abstract as following:

"We investigate emission inventory uncertainties and their propagation to PM2.5 concentrations, in order to identify the most effective strategies to be implemented at sector and regional level to improve emission inventories knowledge and air quality modeling. We show that the uncertainty of PM concentrations depends not only on the uncertainty of local emission inventories, but also on that of the surrounding regions. Countries having high emission uncertainties are often impacted by the uncertainty of pollution coming from surrounding regions, highlighting the need of effective efforts in improving emission not only within a region but also from extra-regional sources. Finally, we propagate emission inventories uncertainty to PM concentrations and health impacts. We estimate 2.1 million premature deaths/year with an uncertainty of more than 1 million premature deaths/year due to the uncertainty associated only with the emissions."

Regarding SOA, given the pace at which our understanding of how SOA forms has evolved, I'm not sure a 2010 paper (Farina) is "recent" anymore. But still, the discussion here is appreciated.

We added the following two references in addition to the work of Farina et al. 2010:

**Shiraiwa et al. (2017) and Peng et al. (2016).**

In response to my question about previous line 12.10 - 15 ("why do these regions have large extra-regional contributions"), the response (health impacts are large because pollution is large) is a bit lacking. Why is it larger here, say, than other parts of the world? Is the long-range transport here particularly strong or efficient? I also wonder if the answer may have to do with underlying baseline mortalities being higher in some regions.

In the paper we now refer to section 3.1 to clarify this concept. PDs attributed to internal/external emissions are directly linked (proportional) to the internal/external PM2.5 contributions discussed in section 3.1. However, the GULF region has higher internal than external contribution, so we removed it from the exceptions.

"As explained in Sect. 3.1, PDs attributed to internal/external emissions are directly linked (proportional) to the internal/external PM2.5 contributions. For most of the TM5-FASST regions, PDs due to anthropogenic emissions within the source region are higher than the extra-regional contributions. However, there are marked exceptions, such as the Gulf region, Hungary, Czech Republic, Mongolia, etc., where the extra-regional and within-region contributions to mortality are at least comparable. In factFor instance, Hungary and Czech Republic are strongly influenced by polluted regions in Poland (mainly); likewise Mongolia is suffering fromaffected by the vicinity of sources in China. The Gulf region produces a lot of its own pollution, but is also influenced by transport from Africa and Eurasia as reported by Lelieveld et al. (2009)."

   Dentener1

[revised manuscript text omitted]
| 42 | (Eq. 2)                                                                                                                                                                                 |                 |

 $dPM_i(y) \cong A_{ii}[x, y] [E_i(x) - E_{i,ref}(x)]$

| { | Formatted: Font: Italic     |
|---|-----------------------------|
| { | Formatted: Font: Italic     |
| { | Formatted: Subscript        |
| 1 | Formatted: Font: Italic     |
| Ì | Formatted: Font: Italic     |
| 1 | Formatted: Font: Not Italic |
| { | Formatted: Font: Not Italic |

(Eq. 1)

1 where the summation is made over all primary emitted components and precursors (*i*) for 2 secondary components, and  $A_{ij}[x, y]SRC_t[x, y]$  is a set of independently computed Ssource-3 Rreceptor Cmatrices, officients describing expressing the linearized relationship emission-4 concentration response between each relevant precursor (*j*) emission of specific components 
[revised manuscript text omitted]
    | 6.1 (3.1 - 12.5)                                        | 84%                                                        |
| ddle         | Middle East             | 9.2 (5.4 - 17.8)                                        | 58%                                                        |
| ſ/ Mi        | Turkey                  | 8.7 (4.9 - 17.1)                                        | 67%                                                        |
| Gul          | Gulf region             | 7.8 (4.7 - 14.5)                                        | 57%                                                        |
|              | Brazil                  | 1.6 (1.1 - 2.6)                                         | 85%                                                        |
| ca           | Mexico                  | 4.2 (2.1 - 9.2)                                         | 62%                                                        |
| meri         | Rest of Central America | 2.0 (1.0 - 4.0)                                         | 78%                                                        |
| itin A       | Chile                   | 13.7 (7.3 - 29)                                         | 70%                                                        |
| La           | Argentina+Uruguay       | 1.1 (0.7 - 1.9)                                         | 77%                                                        |
|              | Rest of South America   | 2.4 (1.6 - 3.9)                                         | 69%                                                        |
| ¥            | Canada                  | 4.3 (2.4 - 8.3)                                         | 66%                                                        |
| N            | USA                     | 7.8 (4.4 - 14.4)                                        | 71%                                                        |
|              | Kazakhstan              | 4.9 (3.2 - 8.9)                                         | 62%                                                        |
| æ            | Former USSR Asia        | 7.5 (4.0 - 17.6)                                        | 49%                                                        |
| Russi        | Russia (EU)             | 3.3 (1.9 - 6.7)                                         | 57%                                                        |
| н            | Russia (Asia)           | 2.7 (1.7 - 5.1)                                         | 64%                                                        |
|              | Ukraine                 | 7.8 (4.2 - 15.9)                                        | 65%                                                        |
|              | Australia               | 1.1 (0.8 - 1.4)                                         | 84%                                                        |
| ceam         | New Zealand             | 0.3 (0.1 - 0.5)                                         | 60%                                                        |
| 0            | Pacific Islands         | 0.2 (0.1 - 0.4)                                         | 75%                                                        |

[revised manuscript text omitted]